# Coordination and Control: Limits in Standard Representations of Multi-Reservoir Operations in Hydrological Modeling

Charles Rougé[1,2], Patrick M. Reed[2], Danielle S. Grogan[3], Shan Zuidema[3], Alexander Prusevich[3], Stanley Glidden[3], Jonathan R. Lamontagne[4], and Richard B. Lammers[3]

[1]Department of Civil and Structural Engineering, University of Sheffield, Sheffield, United Kingdom
[2]Department of Civil and Environmental Engineering, Cornell University, Ithaca, NY, United States
[3]Water Systems Analysis Group, University of New Hampshire, Durham, NH, United states
[4]Department of Civil and Environmental Engineering, Tufts University, Medford, MA, United States

**Correspondence:** Charles Rougé (c.rouge@sheffield.ac.uk)

**Abstract.** Major multi-reservoir cascades represent a primary mechanism for dealing with hydrologic variability and extremes within institutionally complex river basins world-wide. These coordinated management processes fundamentally reshape water balance dynamics. Yet, multi-reservoir coordination processes have been largely ignored in the increasingly sophisticated representations of reservoir operations within large-scale hydrological models. The aim of this paper is twofold, (i) provide evidence that the common modeling practice of parameterizing each reservoir in a cascade independently from the others is a significant approximation, and (ii) demonstrate potential unintended consequences of this independence approximation when simulating the dynamics of hydrological extremes in complex reservoir cascades. We explore these questions using the Water Balance Model, which features detailed representations of the human infrastructure coupled to the natural processes that shape water balance dynamics. It is applied to the Upper Snake River Basin in the Western U.S., and its heavily regulated multi-reservoir cascade. We employ a time-varying sensitivity analysis that utilizes Method of Morris factor screening to explicitly track how the dominant release rule parameters evolve both along the cascade, and in time according to seasonal high- and low-flow events. This enables us to address (i) by demonstrating how the progressive and cumulative dominance of upstream releases significantly dampens the ability of downstream reservoir rules' parameters to influence flow conditions. We address (ii) by comparing simulation results with observed reservoir operations during critical low-flow and high-flow events in the basin. Our time-varying parameter sensitivity analysis with the Method of Morris clarifies how independent single-reservoir parameterizations and their tacit assumption of independence leads to reservoir release behaviors that generate artificial water shortages and flooding, whereas the observed coordinated cascade operations avoided these outcomes for the same events. To further explore the role of (non-)coordination in the large deviations from the observed operations, we use an offline multi-reservoir water balance model in which adding basic coordination mechanisms drawn from the observed emergency operations is sufficient to correct the deficiencies of the independently parameterized reservoir rules from the hydrological model. These results demonstrate the importance of understanding the state-space context in which reservoir releases occur and where operational coordination plays a crucial role in avoiding or mitigating water-related extremes. Understanding how major infrastructure is coordinated and controlled in major river basins is essential to properly assessing future flood and drought hazards in a changing world.

# 1 Introduction

The cumulative impacts of reservoir cascades on river flows has been recognized and demonstrated worldwide by early global hydrological models (Dynesius and Nilsson, 1994; Vörösmarty et al., 1997). Since then, these findings, frequently corroborated in the literature (e.g., Nilsson et al., 2005; Adam et al., 2007; Döll et al., 2009; Biemans et al., 2011; Grill et al., 2019), have taken a new significance with the planned or ongoing construction of more than 3,700 major dams, most of them in the global South (Zarfl et al., 2015). This new wave of dam construction cements the role of man-made reservoirs as key actors on the global hydrological cycle. A striking illustration of this fact is the cumulative consequences of building multiple dams on river flow regimes, ecosystem benefits, or sediment transport in previously relatively undammed major river basins such as the Amazon (Latrubesse et al., 2017; Timpe and Kaplan, 2017) or the Mekong (Schmitt et al., 2018).

In parallel, and as a response to evolving flood and drought risks in a changing world, communities involved in large-scale hydrological modeling aim to address the challenges posed by representing, monitoring, and forecasting these risks at fine resolutions in both space and time (Wood et al., 2011; Bierkens, 2015). For high-resolution modeling of multiple reservoir systems, reservoirs should not be lumped together, but rather, their individual impacts on system dynamics should be carefully accounted for (Shin et al., 2019). In this context, better representations of how human societies (mis-)manage their water resources needs to be integrated in these models (Wada et al., 2017), especially since currently state-of-the-art models yield mixed results for the modeling of monthly extremes (Zaherpour et al., 2018). There remains opportunities for research to determine which aspects of human management are most urgent to integrate in standard reservoir representations. One such aspect is coordination between reservoirs, long-recognized as a key aspect of water management (e.g., Loucks and van Beek, 2005; Marques and Tilmant, 2013; Jeuland et al., 2014; Quinn et al., 2019). Multi-reservoir coordination implies that release decisions at each reservoir in the basin are explicitly influenced by the current and future state of other reservoirs. So far such behavior has not been implemented in release rules for large-scale hydrological models. It is not clear to which extent this can be related to results from a recent intermodel comparison by Masaki et al. (2017), who found discrepancies between models when representing flows across large reservoir cascades. This echoes an earlier study that found deteriorating goodness-of-fit of monthly releases along such cascades (Adam et al., 2007).

The present study uses a diagnostic analysis of the implication of noncoordinated parametrizations for reservoir release decisions to (i) provide evidence that the common modeling practice of parameterizing each reservoir in a cascade independently from the others is a significant approximation, and (ii) demonstrate potential unintended consequences of this independence approximation when simulating the dynamics of hydrological extremes in complex reservoir cascades. We focus on a highly resolved model of the Upper Snake River Basin (USRB) – 30 arc seconds spatial resolution for an average grid cell of about 0.6 square kilometer, and a daily time step – that encompasses a total of 128 reservoirs in the Western U.S. Our model-based representation of the USRB exploits the Water Balance Model (WBM; e.g. Wisser et al., 2010), which is well-suited for regional or global scale hydrological assessments (e.g. Wisser et al., 2008; Grogan et al., 2017) and includes a representation of human impacts on the water cycle. The remainder of this introduction reviews reservoir representations in hydrological models, including their use for flood and drought modeling, and key aspects of our contributed diagnostic assessment.

Early attempts at representing man-made reservoirs modeled them as natural reservoirs (i.e., lakes; Meigh et al., 1999; Coe, 2000; Döll et al., 2003). In 2006, representations proposed separately by Haddeland et al. and Hanasaki et al. introduce the idea that man-made reservoirs should have a distinct parametrization that reflect the reservoir's purpose, leading to two different kinds of reservoir representations (Nazemi and Wheater, 2015). Haddeland et al. (2006) optimize release for the upcoming year assuming future inflows are known, and following a management objective in line with the reservoir's primary purpose. This optimization-based scheme has been extended in several studies, most notably van Beek et al. (2011) who replaced perfect foresight of the next year's inflows with an uncertain forecast (for other improvements, see also Adam et al., 2007; Wada et al., 2014). Alternatively, Hanasaki et al. (2006) propose a parametrization that simulates releases based on a set of site-specific parameters such as long-term average inflow, reservoir capacity and beginning-of-year storage, and downstream water demands. There exist several refinements of this rule, by changing the definition of what constitutes downstream demand (Döll et al., 2009), by considering more reservoir purposes (Biemans et al., 2011; Yoshikawa et al., 2014), by allowing the reservoir's primary purpose to vary seasonally (Voisin et al., 2013a), or by proposing a general rule differentiating refill and drawdown seasons for large multipurpose reservoirs (Wisser et al., 2010).

This first generation of reservoir representation has led to improved simulations of historically observed discharge at the monthly time scale (Pokhrel et al., 2012; Li et al., 2015; Veldkamp et al., 2018). It has been integrated into increasingly complex modeling frameworks. For instance, the rule first proposed for a global flow routing model by Hanasaki et al. (2006) has been integrated as part of the global hydrological model H08 (Hanasaki et al., 2008), which then has been integrated into a land surface model that models the carbon, energy and water cycles (Pokhrel et al., 2012). Similarly, the rule proposed by Voisin et al. (2013a) has been incorporated into increasingly complex modeling frameworks accounting for regional-scale feedbacks between climate, socio-economic systems and heavily managed water, energy and food systems (Voisin et al., 2013b; Kraucunas et al., 2015). As the models including these reservoir representations have grown more complex, so have the questions asked of them. Applications typically include assessments of past and present water withdrawals, human impacts on hydrology, water stress and scarcity (e.g., Biemans et al., 2011; Wada et al., 2011, 2014; Yoshikawa et al., 2014; Hanasaki et al., 2018; Liu et al., 2019; Meza et al., 2019). Recently, modelling frameworks have been extended to include water quality (Wanders et al., 2019) or economic appraisals of the consequences of scarcity (Bierkens et al., 2019). These models are also increasingly being used for appraisals of future water scarcity under integrated socio-economics and climatic scenarios (e.g., Hanasaki et al., 2013; Hejazi et al., 2015; Jägermeyr et al., 2016; Herbert and Döll, 2019).

Reservoir management is also critical for understanding flooding, where simulations must resolve much finer timescales (i.e., daily or shorter). Reservoir rules like those of Hanasaki et al. (2006) can be modified to be applied with a daily time step to investigate the potential of reservoir management to alleviate flooding (Mateo et al., 2014), or be modified to better model the periods when reservoirs are nearly full (Shin et al., 2019). Large-scale or global flooding assessments are made more complex by the fact that hydrologic routing by itself is insufficient for floodplain modeling (e.g., Sampson et al., 2015; Schumann et al., 2016). In this context, a good first approximation to account for reservoirs is to allocate flood storage capacity following an extreme precipitation event, especially since this alone can dramatically alter a flood's outcome (Metin et al., 2018). Yet, subtle changes in flood management by reservoirs can have decisive impacts, both in theory (Najibi et al., 2017), and in observed

catastrophic flooding events like in Kerala (Southern India) in 2018 (Mishra et al., 2018). A finer assessment of the capacity of reservoirs for flood management involves explicit consideration of the multipurpose nature of reservoirs, as they often are assigned flood control duties on top of other uses. To achieve this, the representation proposed in LISFLOOD by Burek et al. (2013) partitions storage into different compartments; Zajac et al. (2017) demonstrated the merits of this formulation for flood impact assessment at the global scale.

Similar to Burek et al. (2013), several representations of varying complexity have been proposed to divide active storage capacity into several compartments, both to obtain sensible operations at submonthly time steps and to account for the fact that most reservoirs are inherently multipurpose installations (Wu and Chen, 2012; Zhao et al., 2016; Wang et al., 2019; Yassin et al., 2019). Another way to account for the complex nature of operations at a daily time step has been to directly emulate observed operations using machine learning techniques (Ehsani et al., 2016; Coerver et al., 2018). Both types of approaches have also been implemented in search for representations that can adapt to evolving climate conditions (Ehsani et al., 2016; Zhao et al., 2016). Thus, Ehsani et al. (2017) demonstrated the role of reservoir storage in alleviating the impacts of both floods and droughts under a changing climate in the Northeastern U.S..

It is worth noting, however, that all of the reservoir representations discussed above do not account for coordination within multi-reservoir systems. In other words, consequences of a release decision on downstream reservoir levels (and management objectives) are not considered. To date, there has not been a carefully designed diagnostic model evaluation of the implications of errors in representing actual human coordination and controls in high-impact, complex river basin contexts. This study links observed operations for recent high- and low-flow events in the USRB's reservoir cascade to clarify how standard representations of release rules capture the underlying coupled human-natural processes that are critical to model-based assessments of our vulnerabilities to extremes. The diagnostic model evaluation approach used in this work employs time-varying sensitivity analysis (e.g., Reusser and Zehe, 2011; Herman et al., 2013b; Guse et al., 2014; Pianosi and Wagener, 2016; Lamontagne et al., 2019; Quinn et al., 2019). Building on prior successful diagnostic model evaluation studies, our sensitivity analysis is based on the factor screening capabilities of the Method of Morris (Morris, 1991; Campolongo et al., 2007), which requires significantly less computation time than other methods while providing high-fidelity measures of model controls (Herman et al., 2013a; Iooss and Lemaître, 2015). We explicitly map how reservoir rule parameterizations relate to the impacts of model behavior across the successive reservoirs within the USRB cascade at a daily time-scale. To isolate the impacts of (not) including coordination in reservoir release rules, we complete the analysis with simple offline water balance models in which we add simple coordination mechanisms similar to the ones we observed in recent real-world operations of the USRB's multi-reservoir cascade.

This diagnostic assessment exploits the Water Balance Model with a simulation-based parametric release rule introduced by Proussevitch et al. (2013) and incorporated to WBM in several recent large-scale assessments (Grogan et al., 2015, 2017; Zaveri et al., 2016; Liu et al., 2017). This representation is state-of-the-art in its ability to reproduce the climatological daily water balance of a single reservoir over the year with high accuracy. The possibility to use different parametrizations depending on the reservoir's perceived use and behavior, and the fact that release behavior structurally depends on storage level, are all features that capture the advanced reservoir representations currently in use in other models. Note that we do not seek to validate this

release rule, but rather, to use it as a typical example of release rule abstractions in large-scale hydrological models, in that it does not feature direct coordination between reservoirs' release decisions.

The rest of this work is structured as follows. Section 2 presents the study area and model used for the analysis, including a detailed explanation of the reservoir rule. Section 3 introduces and justifies the methodological aspects of the analysis. Section 4 presents the results from the diagnostic approach. Section 5 and 6 discuss the implications of our findings as well as our overall concluding remarks.

## 2    Study area and model

### 2.1    The Upper Snake River Basin

The Snake River originates east of the Teton Range in western Wyoming, then crosses the mountains into the Snake River Plain in southern Idaho. After flowing west through the entirety of that plain, it flows north to join the Columbia River. This work focuses on the Upper Snake River Basin (USRB; Figure 1), which has a drainage area of about 92,000 sq. km and is characterized by a snow-dominated, semi-arid climate. To ensure water availability for the whole agricultural season, the U.S. Bureau of Reclamation has built and operated a network of reservoirs, canals and lateral distribution ditches since the early twentieth century (U.S. Bureau of Reclamations, 2012). Since then, a diverse array of demands, including hydropower, irrigation, ecological conservation, and downstream water allocation, has increasingly required the USRB to be extensively managed with a network of dams of a broad range of sizes: 128 reservoirs of over $10\ hm^3$ (10 million $m^3$) throughout the basin, for a total volume of $6.93\ km^3$. Its waters are over-allocated across the USRB's competing demands (McGuire et al., 2006). The over-allocation is at least partially the result of historical perceptions of water availability where the twentieth century was wet compared with previous centuries (Wise, 2010). In fact, water availability is decreasing (Ahmadalipour and Moradkhani, 2017), forcing farmers to adapt to drier conditions (Hoekema and Sridhar, 2011). These drying trends are expected to worsen with climate change, especially as this will be accompanied by an increasing mismatch between seasonal patterns of water availability and use (Hamlet and Lettenmaier, 1999; Rauscher et al., 2008; Wise, 2012).

The USRB is also vulnerable to rain on snow events that can lead to extreme flooding. These are a common occurrence in the wider U.S. Northwest and are expected to get worse in the future (Musselman et al., 2018). In the USRB, a historically significant flood that caused widespread damage occurred in February 1962, with rainfall on frozen ground following a particularly cold spell (Thomas and Lamke, 1962). This was despite the recent completion of the Palisades Dam giving the Minidoka Project a significant ability to coordinate storage capacity for both water supply and flood control. Following this event, the USRB was also the site of the Teton Dam failure in 1976 (Independent Panel To Review Cause of Teton Dam Failure, 1976). All of these characteristics – heavy reliance on institutionally coordinated reservoir management in a drought- and flood-prone area that has experienced the consequences of dam failure, and where water extremes are expected to get worse with climate change – make the USRB a particularly relevant basin to study the representation of reservoirs within large-scale hydrological models. Any unintended consequences from modeling non-coordinated operations would be a note of caution for large-scale studies featuring water infrastructure balancing protection against water extremes with other competing uses.

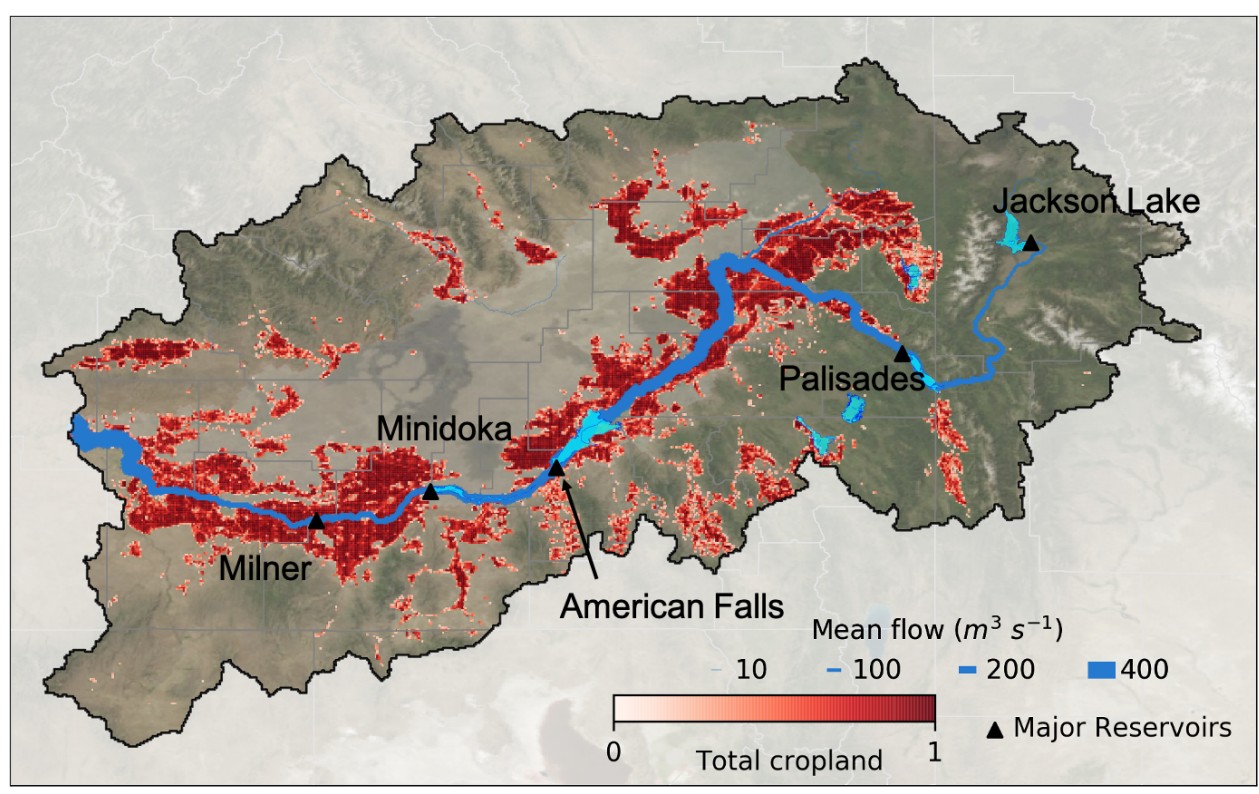

**Figure 1.** Upper Snake River basin (USRB) with the five reservoirs on the main stem of the Snake River.

## 2.2 The Water Balance Model

The University of New Hampshire Water Balance Model (WBM) is a process-based, modular, gridded hydrologic model that simulates spatially and temporally varying water volume and water quality across a wide range of spatial domains from global half-degree grid cell resolution (e.g., Grogan et al., 2017) to local 120 m grid cells (Stewart et al., 2011). WBM represents all major land surface components of the hydrological cycle, and tracks fluxes and balances between the atmosphere, above-ground water storage (e.g., snowpack, glaciers), soil, vegetation, groundwater, and local runoff. A digitized river network connects grid cells enabling simulation of flow through the river and groundwater systems and for simulating water temperature (Stewart et al., 2013). Direct human influences on the water cycle include domestic, industrial, and agricultural (irrigation and livestock) water demand and use, and the impacts of impervious surfaces. WBM accounts for the operation of dams and reservoirs (Wisser et al., 2010), inter-basin hydrological transfers (Zaveri et al. 2016), and agricultural water use from irrigation (Wisser et al., 2010; Grogan et al., 2015; Grogan, 2016; Zaveri et al., 2016; Grogan et al., 2017; Zuidema et al., 2020). Additionally, new WBM modules have been developed recently to include the use of sub-grid elevation band distributions derived from a high-resolution elevation dataset to improve handling of snowpack in mountainous regions.

## 2.3 WBM representation of the USRB

A drainage network of the USRB that covered an area of 92,900 $km^2$ (compared to USGS's estimate of 92,700 $km^2$) was developed at a spatial resolution of 30-arcseconds (approximately 780 $m$) based on HydroSHEDS (Lehner et al., 2008) corrected to better represent drainage as mapped by the US Geological Survey's National Hydrography Data (nhd.usgs.gov). Reservoir data were derived from the National Inventory of Dams (nid.usace.army.mil). We manually added dams and updated reservoir capacities, locations, and upstream drainage areas. WBM simulations used gridMET (Abatzoglou, 2013) for contemporary precipitation and temperature, and Modern Era Retrospective-Analysis for Research and Applications, version 2 (MERRA2; Gelaro et al., 2017) for open water evaporation, windspeed, relative humidity, leaf area index, and albedo to calculate potential evapotranspiration following Monteith (1965). Snow accumulation and melt followed the temperature-index based formulation of Willmott et al. (1985). Human population density, which controls both domestic and industrial water demand, came from the Gridded Population of the World (GPW) dataset (Center For International Earth Science Information Network-CIESIN-Columbia University, 2016). WBM used Food and Agricultural Organization (FAO) estimates of livestock density for cattle (Steinfeld et al., 2006) at 5 minute resolution following Wisser et al. (2010). These data compared favourably with the U.S. Department of Agriculture's (USDA) National Agricultural Summary Statistics (NASS) for 2005, but exhibit more realistic spatial variability than county-level averages. USDA Soil SURvey GeOgraphic (SSURGO) data parameterized available water capacity for the USRB.

WBM uses an adaptation of FAO's Irrigation and Drainage paper (Allen et al., 1998) to estimate crop water requirements based on potential evapotranspiration, soil moisture, and a crop coefficient ($k_c$) defining a particular crops' water use efficiency. Details regarding the crop water demand calculations are provided in previous works (Grogan et al., 2017; Wisser et al., 2010). This study used the US Department of Agriculture's Crop Data Layer (CDL) estimates of crop types (and land cover) at 30 m resolution (Han et al., 2012), aggregated by surface area averaging and remapped to a consistent group of crops as monthly irrigated and rainfed crop areas (MIRCA) crops (Portmann et al., 2010). We applied $k_c$, planting dates, and crop depletion factors from MIRCA to the CDL crop fractions. Open water, impervious area, and wetland data also came from CDL data. Process based representation of irrigation technology was recently introduced to WBM following key aspects of the formulation of Jägermeyr et al. (2015). Irrigation technologies used in the USRB varied by county following Maupin et al. (2014) and Dieter et al. (2018). Additional details regarding the specific implementation of irrigation technologies will be reported in a separate paper.

## 2.4 Reservoir representation within WBM

WBM's release rule for managed reservoirs expresses daily release as a fraction of long-term (five years or more) mean release at the reservoir as illustrated in Figure 3. This is a refined convention for release rules within hydrological models (Hanasaki et al., 2006; Wisser et al., 2010) to be primarily controlled by instantaneous reservoir storage and purpose rather than statistics on the probability distribution of inflow rates. WBM's reservoir module operates on a hourly time step to closely follow storage variations and yield a daily release total. The general form of the reservoir rule was first presented by Proussevitch et al. (2013)

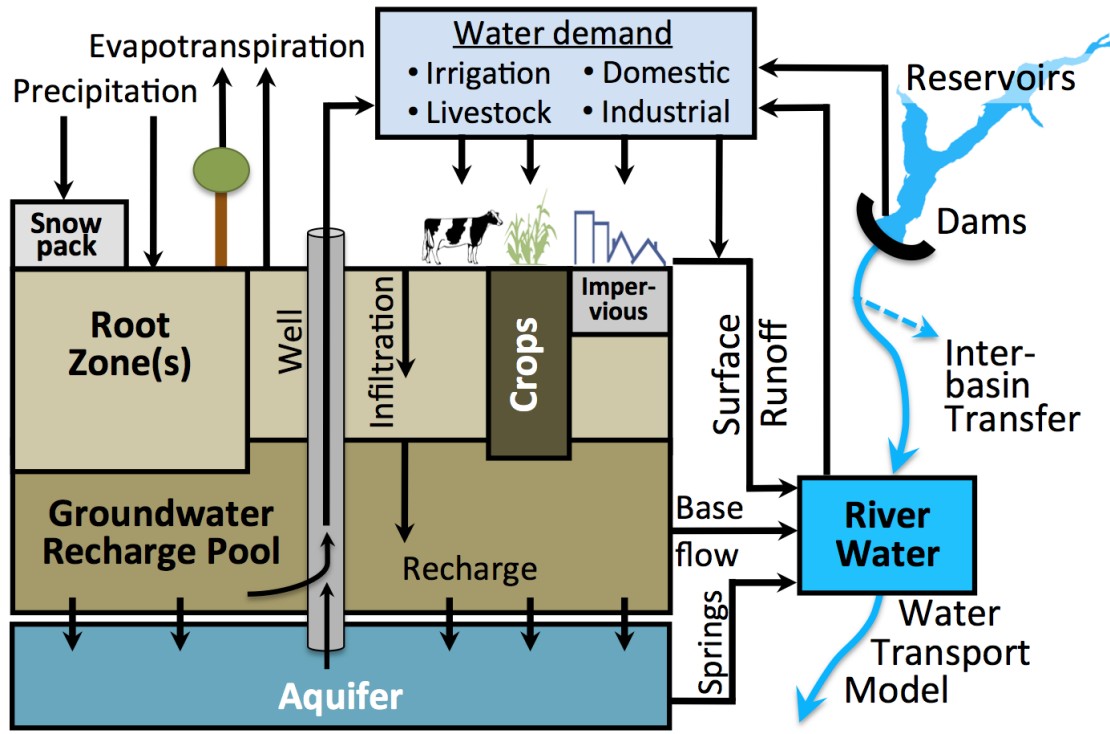

**Figure 2.** Conceptual representation of the Water Balance Model as used in this study.

and validated using the GRanD database (Lehner et al., 2011). Variants of this rule have been used with a daily time step on the Niger river basin (Oyerinde et al., 2016), and with large-scale assessments using WBM (Grogan et al., 2015, 2017; Zaveri et al., 2016; Liu et al., 2017). The fine-tuning of the parameters when establishing this version of the rule was made using a set of 22 large North-American and Eurasian reservoirs in offline mode, including the two largest reservoirs in the USRB

(Palisades and American Falls, daily release Nash-Sutcliffe eficiency (NSE) coefficient 0.70 and 0.60 respectively). Similar to what happens when a reservoir rule that classifies reservoirs by purpose is used in a large-scale model, we did not fine-tune the rule to each reservoir. This allows us to use the reservoir rule in conditions that are similar to what is done in most state-of-the-art hydrological models.

WBM's release rule, there are structurally different behaviors delimited by a reference storage $S_{ref}$, below which the priority

is to refill the dam and above which release levels increase rapidly as the reservoir gets nearly full. We call $R_{ref}$ the release at reference storage. For storage $S < S_{ref}$, the rule designed to favor filling the reservoir expresses release $R$ a logarithmic function of storage $S$:

$$R = R_{\min} + \frac{\ln(1 + P_R S)}{\ln(1 + P_R S_{ref})}(R_{ref} - R_{\min}) \tag{1}$$

where $R_{\min}$ is release at minimal storage, and $P_R$ is a shape parameter for the logarithmic part of the rule that controls the

propensity for release. Indeed, $P_R$ close to zero leads to an almost linear rule whereas the higher $P_R$, the more release gets

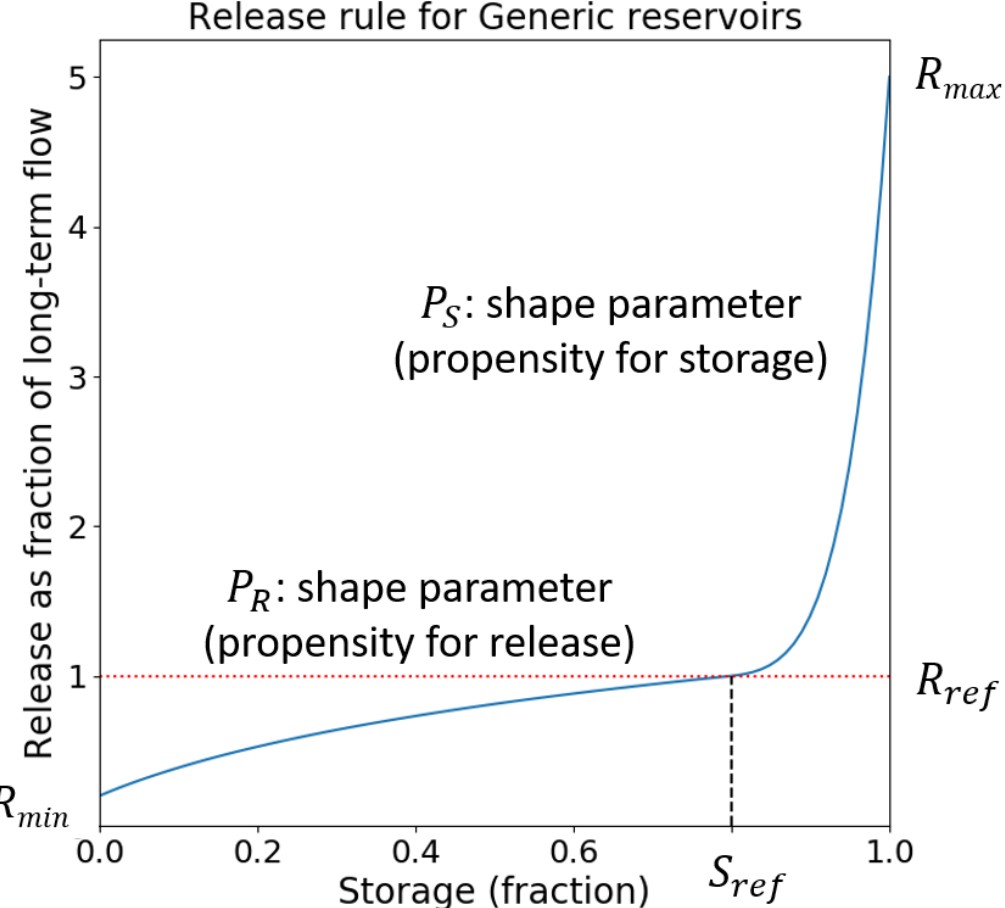

**Figure 3.** 6-parameter reservoir rule. Release is scaled by long-term annual inflow while storage is scaled by the active capacity: it is 0 at dead storage and 1 at full storage.

close to $R_{ref}$ even for near-empty storage. For $S \geq S_{ref}$, release $R$ varies exponentially with storage $S$:

$$R = R_{ref} + \frac{(S - S_{ref} + \Delta S)^{P_S} - \Delta S^{P_S}}{(S_{max} - S_{ref} + \Delta S)^{P_S} - \Delta S^{P_S}} (R_{max} - R_{ref}) \tag{2}$$

where $R_{max}$ is release at full storage and $\Delta S$ is computed from the other parameters to ensure that the transition between the logarithmic and exponential parts of the release rule is "smooth" (continuously differentiable). The exponential shape parameter $P_S$ is the propensity for storage, since it minimizes releases until storage is close to its maximal level.

Thus, there are six parameters for the reservoir rule in Figure 3: shape parameters $P_R$ and $P_S$ which represent respectively the propensity for release at low storage (getting releases closer to $R_{ref}$ faster) and for storage in near-full reservoirs (delaying

| Purpose | | $P_R$ | $P_S$ | $R_{\min}$ | $R_{\max}$ | $S_{ref}$ | $R_{ref}$ | $irrFreq$ | Number of reservoirs |
|---|---|---|---|---|---|---|---|---|---|
| Irrigation | (low) | 1 | 3 | 0.01 | 2 | 0.8 | 0.1 | range $[0,1]$ | 73 |
| | (high) | 297 | 3 | 0.292 | 0.885 | 0.949 | 1.44 | | |
| Generic | | 4 | 6 | 0.2 | 5 | 0.8 | 1 | N/A | 33 |
| Hydroelectric | | 200 | 3 | 0.2 | 1.25 | 0.9 | 1 | N/A | 12 |
| Water Supply | | 1 | 6 | 0.1 | 5 | 0.7 | 0.1 | N/A | 10 |

**Table 1.** Parameters for reservoirs in the USRB. The last column classifies the basin's 128 reservoirs by primary purpose.

releases for as long as possible); releases $R_{\min}$ and $R_{\max}$ at minimum and maximum storage; and the coordinates $S_{ref}$ and $R_{ref}$ of the (reference) inflection point. Note that this parameterization, similar to those of other state-of-the-art rules in large-scale hydrological models, do not account for possible coordination mechanisms in multi-actor, multi-reservoir systems. These parameters depend on the reservoir's primary purpose, as shown in Table 1.

"Irrigation" represents the dominant primary use: taken together, irrigation reservoirs represent a storage capacity of 6.27 $km^3$, or just over 90% of the USRB's total storage capacity. Note that "Irrigation" reservoirs require a seventh parameter to model the need to refill to store water for the irrigation season, and release it with the appropriate timing. This parameter, noted $irrFreq$, represents the relative frequency of water demand for irrigation throughout the year. It affects the release rule through each of the other parameters $p_i$ with $1 \le i \le 6$, according to:

$$p_i = p_i^{low} + irrFreq \cdot (p_i^{high} - p_i^{low}) \tag{3}$$

with $irrFreq$ between 0 and 1 and the low and high values of the parameters defined in Table 1. This results in three distinct release rules depending on time of year, as shown in Figure 4. Winter features a refill phase ($irrFreq = 0$) with low releases except for keeping a flood control compartment available, whereas peak irrigation season is a drawdown phase ($irrFreq = 1$) with high releases no matter the storage level. A shoulder season ($irrFreq = 0.5$) smooths out the transition between the two.

The reservoir rule for "Hydroelectric" primary use shows a near constant release except at very low storage levels, thanks to a very high propensity for release when $S < S_{ref}$. The primary function for the "Water supply" use is to keep releases minimal and storage maximal in order to maximize the quantity of water that can be drawn directly from the reservoir – except for circumstances that require flood control at near-full storage. Other reservoirs mix different uses, and they are represented by the "Generic" rule form that corresponds to Figure 3, and represents an implicit trade-off between uses that prioritize release

and those that prioritize storage.

Despite its relative simplicity, the release rule proposed here shares several important characteristics with other rules proposed in the literature. The logarithmic and exponential portions mirror the intuition that the release behavior is structurally different depending on storage levels, a trait emphasized by some recent release rules (Wu and Chen, 2012; Zhao et al., 2016; Wang et al., 2019; Yassin et al., 2019). Besides, the representation of reservoirs based on their primary purpose has been a re-

curring theme since the seminal release rules by Hanasaki et al. (2006) and Haddeland et al. (2006); the time-varying $irrFreq$ parameter also enables irrigation reservoir to have a flood control behavior in winter, similar to the improvement proposed by

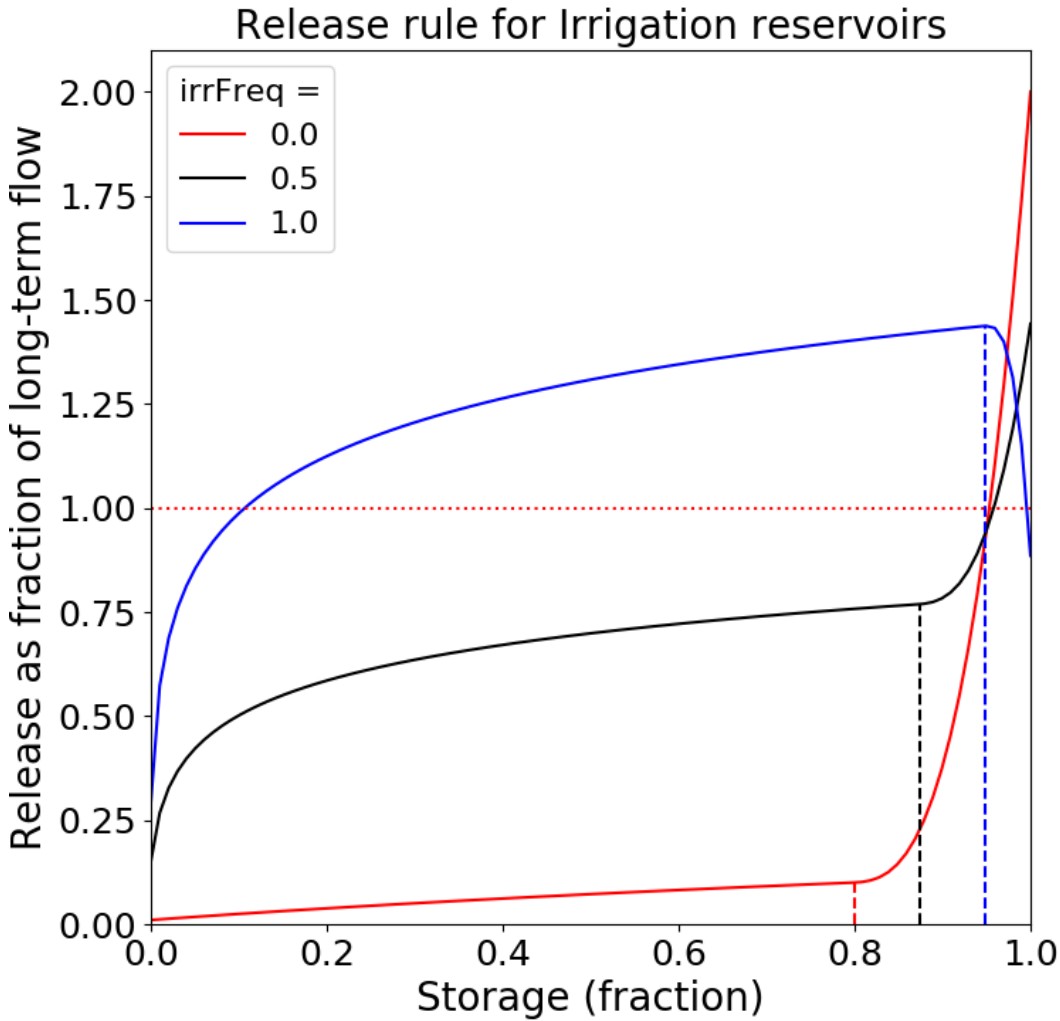

**Figure 4.** Impact of the seasonal shape parameter for reservoirs whose primary purpose is irrigation. Dashed line indicate the inflection point $S^*$. We have $irrFreq = 1$ between July 18 and August 30 included, 0 between October 12 and April 23 included, and 0.5 during the shoulder seasons.

Voisin et al. (2013a). Finally, there is the option to fine-tune individual reservoir's release rule parameters to better represent actual (generally multi-purpose) operations. Yet, adjusting parameters implies the assumption that the release rule is able to capture the main processes at play in operations of a multi-reservoir system. The following section introduces the experimental setup to diagnose this.

## 3 Methodology

### 3.1 General approach

This work endeavors (i) to provide evidence that the common modeling practice of parameterizing each reservoir in a cascade independently from the others is a significant approximation, and (ii) to demonstrate potential unintended consequences of this independence approximation when simulating the dynamics of hydrological extremes in complex reservoir cascades.

Our diagnostic global sensitivity analysis uses the Method of Morris to mathematically trace how downstream parameter choices or effects can be overwhelmed by upstream operational rules that are parameterized independently. The focus is not on which parameters in the release rule are most influential in regulating flows, but on clarifying how the set of dominant controls on water flows and reservoir storage levels evolves along a complex multi-reservoir cascade through time. Parametric sensitivities are then used alongside storage and release trajectories for the simulated ensemble to assess how sets of dominant controls in a point in time and at a given reservoir can be associated to high- or low-flow conditions. We detail the Method of Morris in Section 3.2 and the experiment we design with it in Section 3.3.

Tracking the set of dominant controls through space and time with the Method of Morris enables us to rigorously document the quantitative effects of independent parameterizations of the reservoir cascade, and highlight that downstream dynamics are not strongly controlled by independent downstream operational curve parameter specifications (contribution (i)). The unintended consequences of treating the reservoirs independently (contribution (ii)) are shown by comparing and constrasting the reservoir rule representation with real-world coordinated reservoir operations. For this, several steps are implemented in succession. First, a prerequisite is to observe dynamics in historical (i.e., observed) operations that cannot be accounted for by a reservoir's release rule simply by changing parameter values or integrating a near-term (less than a month) inflow forecast such as in some existing rules (e.g., Biemans et al., 2011). Other variables would be necessary to incorporate these behaviors within the hydrological models. These behaviors are hypothesized as preliminary indications of cooperation, and the events they highlight will be investigated further. Second, the comparison of historical and simulated operations is used to understand how reservoirs coordinated in the historical record, and contrast the impact of this with reservoir non-coordination in simulations. Differences observed in this comparison could be due to all sorts of errors in the model, and not just failing to represent coordination. Therefore, in a third and final step, the hypothesis that coordination is a key source of the release and storage discrepancies is further evaluated using offline reservoir water balance models that implement basic coordination on the simulated ensembles at times where coordination is hypothesized to be present and important in the real system's dynamic management of extreme events. These offline reservoir water balance models show that very basic coordination strategies alone are capable of correcting the documented differences between simulated and historical operations. Note that we cannot fully

quantify and explain all sources of errors: there is to our knowledge no study of a real operational context using a hydrological modeling experiment where one would absolutely know every source of error or potential confounding factor. However, the effectiveness of our basic representation of coordination in the water balance models does contribute a simple and quantitatively direct reduction of errors relative to the actual observed operational dynamics. The details of our water balance analysis can be referenced in Section 4.5.

We focus our application of this approach on the USRB in 2009-2016. The geographical extent of the USRB is small compared with that of areas traditionally considered in large-scale hydrological models. However, the regional focus on the USRB captures a highly dynamic and heavily controlled major reservoir cascade that is critical to managing floods and droughts. Conclusions from this study are fully transferable to larger scales, where critical representational errors in major infrastructure remain consequential. For this reason, we also aim to evaluate model outcomes in the same conditions as large-scale hydrological vulnerability assessments are carried out. For instance, we do not fine-tune reservoir rule parameters to individual basins, and instead use purpose-based parameterizations typical in large-scale studies (e.g., Biemans et al., 2011; Voisin et al., 2013a; Yoshikawa et al., 2014).

### 3.2 Time-varying sensitivity analysis: Method of Morris

The Method of Morris (Morris, 1991; Campolongo et al., 2007) has proven to be a successful tool for detailed diagnostic evaluations of large and complex hydrological models (e.g., Herman et al., 2013b; Zajac et al., 2017; Reinecke et al., 2019). This section presents the sampling technique used, the basic Morris sensitivity indices as well as a time-varying version of these indices. All sensitivity analyses were performed using the SALib toolbox written in Python by Herman and Usher (2017).

The Method of Morris samples points within the parametric spaces of interest by following so-called "trajectories". Two consecutive points of a trajectory share the same input values except for parameter input $i$ where they are separated by a distance $\Delta_i$. The value of input dimension $i$ is changed exactly once along a trajectory, and the order in which input dimensions are changed is random. If $D$ is the dimension of the parametric input space being sampled, then each trajectory comprises $D+1$ points. To ensure that the Morris sensitivity measures are as accurate as possible, sampling must cover the parametric input space as well as possible. This paper implements the method proposed by Ruano et al. (2012), which first generates a large number of trajectories, then selects a subset that provides near-optimal input space coverage using a computationally efficient optimization technique (as implemented here, $M = 50$ trajectories were selected out of a thousand).

To compute Morris indices from a set of $M$ input trajectories, one must run the model whose parametric input space is being sampled at each point $x$ of each trajectory. Therefore, there are $M \times (D + 1)$ model runs. For each trajectory $j$ ($1 \leq j \leq M$), model runs yield the so-called elementary effect along input dimension $i$ for each date $t$:

$$EE_i^j(t) = \frac{f(x_1, \ldots, x_i + \Delta_i, \ldots, x_M) - f(x)}{\Delta_i} \tag{4}$$

With $M$ trajectories being sampled, sensitivity index $\mu_i(t)$ for input dimension $i$ at date $t$ is the average over the elementary effects:

$$\overline{\mu_i}(t) = \frac{1}{M}\sum_{j=1}^{M} EE_i^j(t) \tag{5}$$

In this work, we are concerned with relative contributions to sensitivity across reservoir rule parameters ($1 \leq i \leq D$) and over a given time period ($t \in [t_1, t_2]$). Therefore, we compute the following normalized values for the Morris sensitivity index:

$$\mu_i(t) = \frac{\overline{\mu_i}(t)}{\mu_{\max}} \tag{6}$$

where $\mu_{\max}$ is the maximal value of $|\overline{\mu_i}(t)|$ over the input space and time frame of interest:

$$\mu_{\max} = \max_{i \in [1,n], t \in [t_1, t_2]} |\overline{\mu_i}(t)| \tag{7}$$

As a result, each $\mu_i(t)$ values will be between -1 and 1. Absolute values close to 1 representing inputs that have a dominant influence on outputs, not only compared with other inputs at that date, but also compared with inputs' impacts on outputs at other dates within $[t_1, t_2]$. Positive values denote that outputs values increase with input values, whereas the contrary holds for negative values.

### 3.3 Reservoir Parameters and Ranges

We conduct this diagnostic analysis with 7 groups of parameters. Each group contains one of the 7 parameters of the release rule for all 128 reservoirs in the USRB. This analysis uses a range of $\pm 10\%$ around base values for all parameters in Table 1. These modest 10% ranges would be conservative if our focus were calibration and not diagnosis. Results (Section 4) will demonstrate that our narrow sampling yields quite substantial effects when compounded across the reservoir cascade in periods where coordinated operations are significant. Besides, there are two reasons for choosing the same range across all parameters: 1) it accounts for the fact that each parameter does not have the same base value across all reservoirs, and 2) it facilitates comparisons between different parameters' sensitivity indices.

Our choice to explore 7 groups of parameters serves to reduce the computational burden of our diagnostic analyses, while facilitating a clear experimental mechanism to investigate the core parameterization assumptions used to capture multi-reservoir release and storage dynamics. It also meets the core objective of this study, which is to clarify the importance of multi-reservoir coordination and control to our model-based assessments of flood and drought vulnerabilities in complex river basin systems. Indeed, the chosen parameter set is necessary and sufficient to answer two key intermediary questions. First, we must understand how release rule parameters from a given reservoir influence its water balance (release and storage) through time. This makes it necessary to consider all 7 parameters of the release rule. Second, we must understand how the release rule parameters from upstream reservoirs influence subsequent at-site reservoir controls. This is key to understanding how the non-coordinated release rule affects the time-varying response to high- and low-flow extremes as we move down a reservoir cascade. Our experimental design highlights when parametric controls on reservoir releases are modified by upstream interferences. Indeed,

| Reservoir name | WBM primary usage | Capacity ($hm^3$) |
|---|:---:|:---:|
| Jackson Lake | Irrigation | 1,078 |
| Palisades | Irrigation | 1,503 |
| American Falls | Irrigation | 2,145 |
| Minidoka | Water supply | 123 |
| Milner | Water supply | 62 |

**Table 2.** Reservoir cascade on the main stem of the Upper Snake River, ordered from upstream to downstream.

the same parameters that increase release at a given reservoir also increase upstream releases. Both effects have opposite consequences for a reservoir's storage. Our analysis will track the instances in which upstream controls dominate at-site controls, and clarify the consequences of this for the reservoir cascade's response to hydrological extremes.

The $D = 7$ parameters and ranges thus defined are used to set up a method of Morris experiment with $M = 50$ trajectories. The ensemble size is $M \times (D + 1) = 400$. We ran this experiment on The Cube cluster at the Cornell Center for Advanced Computing Results. The Cube has 32 compute nodes with Dual 8-core E5-2680 CPUs at 2.7 GHz, with 128GB of RAM. A single run of the USRB WBM takes close to seven hours on average for the USRB, with an eight-year simulation period (2009-2016) preceded by a five-year spinup period. The ensemble of 400 members took almost 3,000 hours of compute time to get and analyze, using parallel runs exploiting Open Message Passing Interface version 1.6.5.

## 4  Results

Our results focus on the reservoir cascade on the main stem of the Upper Snake River (Table 2). The three upstream reservoirs in the table are the three largest reservoirs in the basin, and their capacity to store water for the irrigation season is crucial to the agricultural sector in the USRB. Consequently they are classified as "Irrigation" reservoirs. The two downstream reservoirs are smaller and must be maintained at high storage levels during the irrigation season so that canals can draw directly from them, leading to their classification as "Water supply" reservoirs. All but the most downstream reservoir (Milner) are part of, or associated to, the Minidoka Project, therefore their operations for water supply and flood protection are largely coordinated when deemed necessary. Using an ensemble of WBM simulations computed as specified in Section 3, we carry out a diagnostic evaluation of the parametric controls of the release rules in three steps. Initially, we focus on the upstream reservoir, Jackson Lake, where there are no interferences from other reservoirs upstream (Section 4.1). This is where imulation results enable us to quantify the main controls on a reservoir's release rule, a prerequisite to studying how these controls evolve with hydroclimatic events and along the reservoir cascade. This is also where we can find indications of coordination within the historical record, as defined in Section 3.1. Next, we quantify the upstream interference with downstream releases in the USRB's cascade (Section 4.2). Then, Sections 4.3 and 4.4 contrast actual observed operations with those from the simulated ensembles for recent USRB low and high flow events, respectively. Finally, Section 4.5 shows results from the offline water balance models devised by modifying simulation results with simple coordination mechanisms.

### 4.1 Upstream: controls on release and storage

#### 4.1.1 Dominant parametric controls in simulations

First, let us examine WBM's parametric reservoir rule's dynamic sensitivities through Jackson Lake, an upstream reservoir that is not influenced by inflows from any other reservoir in the USRB. Figure 5 provides a visualization of time-varying sensitivities at the daily time step. Simulated and observed hydrological time series overlay the sensitivities represented by the blue-to-red color scale. The left vertical axis represents the plotted reservoir state (release or storage). In Figure 5, shades of blue, red and white report the normalized Morris sensitivity index in a given time period (the horizontal axis); they are organized over seven lines, each corresponding to one of the seven reservoir rule input variables evaluated in our analysis and listed on the right vertical axis. As indicated by the colorbar right of the figure, shades of reds correspond to normalized Morris sensitivity values close to 1, indicating that the associated variable is dominant and that higher values of it correlate with higher values of the reservoir state. Conversely, shades of blue correspond to normalized Morris sensitivity values close to -1 and indicate that the associated variable is dominant but that higher values of it correlate with lower values of the reservoir state. Finally, whites indicate sensitivity is weak compared with that or other variables and / or dates.

In panel (a) of Figure 5, storage differences that emerge across the sampled parametrizations of the evaluated WBM ensemble members are the time integral of daily release differences, therefore they indicate the cumulative effects over time of how the parameters influence the release rule. Storage sensitivities present clear annual patterns for Jackson Lake, and are more broadly representative of the dominant controls for an "irrigation" reservoir within WBM that has an absence of interactions from other reservoirs. Panel (a) shows that the Jackson Lake storage sensitivities go to zero in periods where maximum storage is attained; this is expected because then, there is no variation in storage across the ensemble. In all other periods, there are three dominant parameters influencing storage: $S_{ref}$, $R_{ref}$ and $irrFreq$, with a remarkably consistent influence over the eight-year simulation period. The direction in which these parameters influence the release rule is consistent with the release rules of Figures 3 and 4. Indeed, higher values of $R_{ref}$ and $irrFreq$ directly increase release, therefore decreasing storage over time, whereas increasing $S_{ref}$ delays the transition between the logarithmic and exponential parts of the rule, and has the opposite effect – except if storage is very high during peak irrigation season, due to the dip observable for $irrFreq = 1$ in Figure 4. Although somewhat reduced in effect relative to top three parameters controlling storage dynamics in the Jackson Lake reservoir, the maximum storage releases $R_{\max}$ are predictably inversely correlated with storage (panel (b) on Figure 5). Yet, the three dominant parameters $S_{ref}$, $R_{ref}$ and $irrFreq$ yield what can be interpreted as the "signature" of the parametric influence of the release rule governing storage over time.

Transitioning to Jackson Lake's parametric sensitivities for releases, the overall magnitude of the normalized Morris indices are substantially reduced and less consistent relative to those for storage (Figure 5 panel (b)). A potential reason for this is a dampening effect, as parameters that increase current releases also decrease future storage, and consequently limit future releases. Another potential reason for the diminished sensitivities overall in panel (b) is that contrary to storage that registers the cumulative effects of parametric differences, release sensitivities peak on particular days, making other time periods less sensitive in comparison. The parametric sensitivities for the Jackson Lake release have an opposite "signature" as that of storage

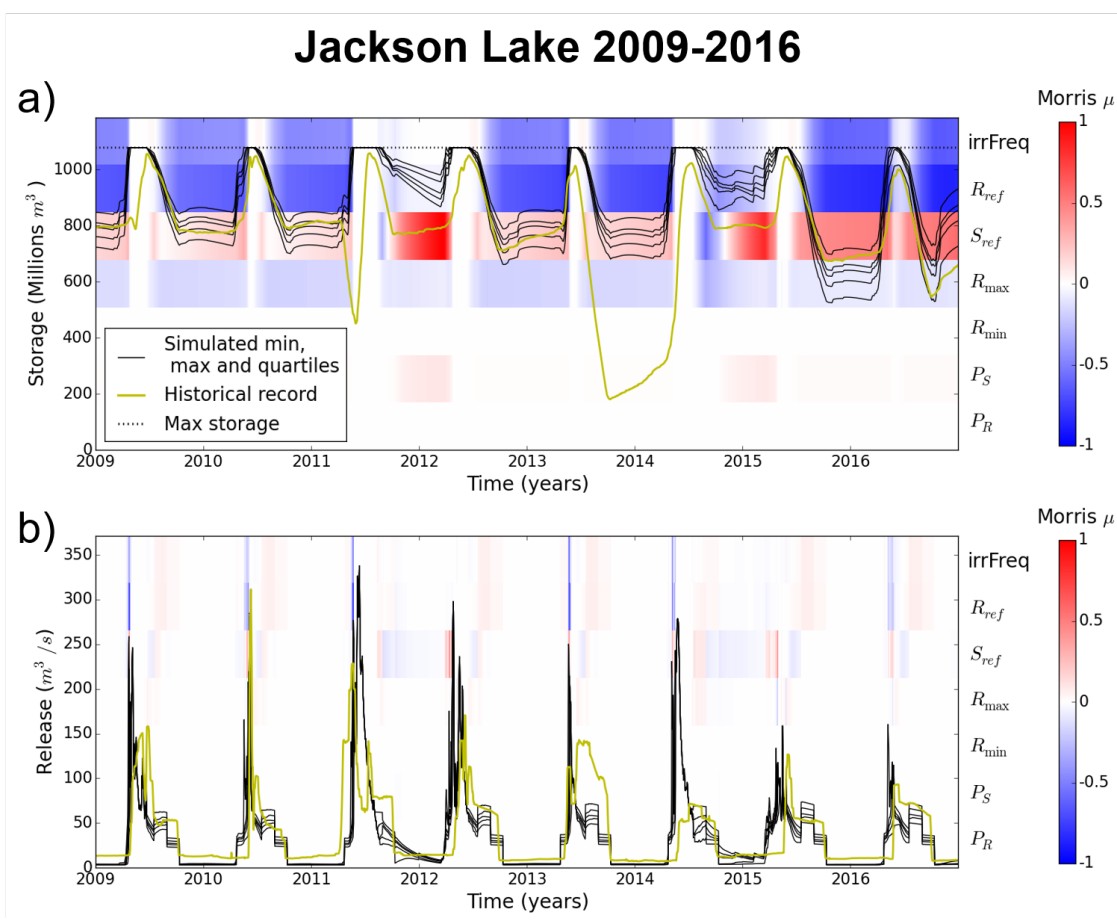

**Figure 5.** Foreground: comparison of the observed (gold line) and simulated (black lines) trajectories at Jackson Lake, for a) reservoir storage and b) release. Background: daily sensitivities to reservoir rule parameters on left y-axis.

during large stretches in summer and fall over 2009-2016. Indeed the three dominant parameters are $S_{ref}$, $R_{ref}$ and $irrFreq$, with higher values of $S_{ref}$ correlating with lower release whereas higher values of $R(S^*)$ and $irrFreq$ correlate with higher releases. Both "signatures" are consistent because parameters that have a sustained impact on release are expected to have an opposite effect on storage.

5  **4.1.2   Comparison with historical operations**

Overall a comparison of historical versus simulated storage and releases in Figure 5 shows a broad agreement during the eight-year study period, despite two major departures, especially apparent for storage. In 2011, early-spring release in the historical record created flood control storage and enabled peak flows to be lower than in the simulated ensemble. Observations of large drawdown in the summer of 2013, with the reservoir replenished only in 2014, are not matched by the simulations. In both

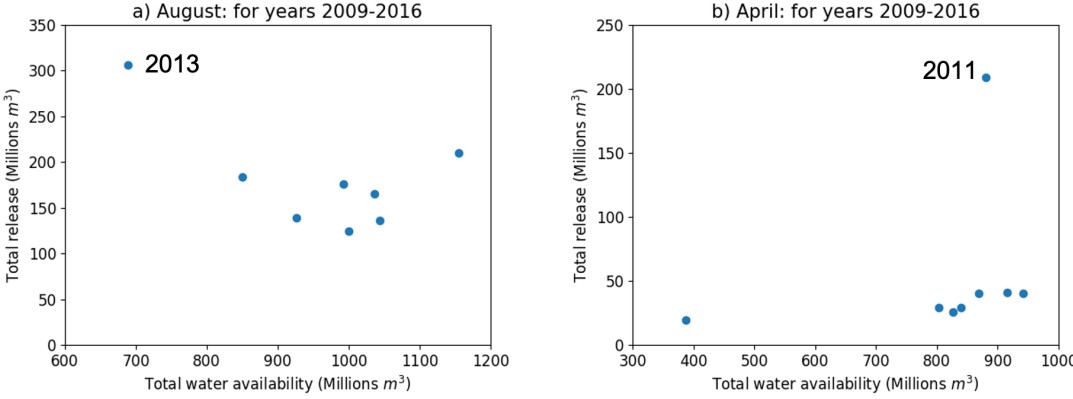

**Figure 6.** Total monthly historical release vs. water availability (beginning-of-month storage plus inflows) for each year between 2009 and 2013 at Jackson Lake. a) left: August and b) right: April

situations, we compared release over a whole month to total water availability (initial storage plus total inflows over the month) for each of the 8 years covered by our analysis. In both cases, results from Figure 6 shows that both events correspond to a major departure with the expectation that release should be indexed on water availability. In 2013 (panel (a)), releases are much higher than any other year despite water availability that August being the lowest of the 8 years. This corresponds to a low-flow period during which extra water is released to help downstream reservoirs meet demand; we contrast this coordinated historical response with simulation results in detail in Section 4.3. In 2011 (panel (b)), releases are over four times higher than normal despite water availability being comparable to conditions for 6 of 7 other years. This is a prelude to an intense snowmelt season, requiring anticipation and coordination from the two main reservoirs tasked with flood control in the USRB: Jackson Lake and Palisades (U.S. Bureau of Reclamations, 2012). We constrast this response with simulation results in Section 4.4.

## 4.2 Absence of downstream coordination in controls

We now transition our focus to the fourth reservoir on the USRB reservoir cascade, Minidoka, which is considerably smaller than the first three. Our analysis in Figure 7 focuses on flows from a single year to clarify the complex interactions between upstream releases and Minidoka operations. Starting with inflows to the reservoir shown in panel (a) on Figure 7, the dominant parameters controlling inflows are $R_{ref}$ and $irrFreq$ (and to a lesser extent $S_{ref}$). The time-varying pattern of dominant release sensitivities across year 2013 (panel (b)) mirrors that of inflows, as dominant parameters tend to positively and negatively correlate with inflows and releases alike at the same time of year. Moreover, the strong release sensitivities to the seasonal $irrFreq$ parameter from May to October can only be due to interactions with upstream reservoirs, because $irrFreq$ only influences irrigation reservoirs' release rule whereas Minidoka is classified as a "Water supply" reservoir. These results suggest that the upstream reservoirs' rules are a dominating factor in this downstream reservoir's release decisions.

However closely variations in simulated releases in panel (b) of Figure 7 tend to follow simulated inflows in panel (a), these releases show unexpected high frequency fluctuations that are artifacts that are not meant to occur in the reservoir's release rule. This shows the unintended consequences of interactions with upstream reservoirs. In other words, it would arguably be very difficult to calibrate the parameters of Minidoka's release rule without accounting for the complex upstream interactions.

Mathematically, this is termed non-separability.

All of these insights from comparing inflow and release sensitivities are confirmed by looking at Minidoka's 2013 storage sensitivities in panel (c). Similar to the release sensitivities in panel (b), the influence of $irrFreq$ on storage is a direct signature of interactions with upstream releases. In fact, the dominant storage sensitivities for the whole year are end-of-April sensitivities to $irrFreq$ and $R_{ref}$ (dark red on panel (c)). The former parameter is not defined for the Minidoka release rule, whereas the

latter should be associated with negative sensitivity (with the color blue) in absence of upstream interactions. The simulated reservoir filling for Minidoka is strongly influenced by parametric artifacts outside of its own parameterization. Beyond that, the picture of time-varying storage sensitivities is extremely complex. For instance, the direction of storage sensitivity to $irrFreq$ (i.e., positive or negative correlation with storage) does not always appear to be clear and consistent with that same parameter's sensitivities for inflows and releases (compare panels on Figure 7). This apparent complexity cannot be dissociated

from upstream interactions, again reinforcing that parameterizing Minidoka's release rule cannot be done separately from the parameterizations of the upstream reservoirs. This meets aim (i), confirming that separate parameterization and calibration of individual reservoirs in a cascade is an approximation. The two next sections explore some possible unintended consequences of this assumption.

## 4.3   Drought risk

We now transition to the reservoir operations along the USRB's reservoir cascade for the consecutive dry years of 2012 and 2013. We contrast coordinated historical operations, illustrated here by storage levels in the basin's three main reservoirs, with the simulations results from our ensemble of hydrological model runs – which we term *simulated* storage in this Section and in the next (Section 4.4). We also analyze the sensitivity of simulated storage to WBM's parametric controls.

The 2012-2013 low-flow event led to a significant simulated drawdown at upstream Jackson Lake in 2013, previously

observed in Figure 5. The strong deviations in the dynamics of historical (gold lines) and simulated (black lines) reservoir operations for both years 2012 and 2013 are apparent in Figure 8. Recall that the two most downstream reservoirs in the Snake River reservoir cascade — Minidoka and Milner – are smaller reservoirs that must stay full during the irrigation season so farmers can draw water through gravity irrigation. Therefore, it is key that American Falls, the main reservoir in the Snake River plain located just upstream of Minidoka, is not empty so that it can keep regulating water levels in downstream reservoirs,

ensuring irrigation needs are met. For this reason our analysis will start with American Falls (panel (c) on Figure 8) and work its way upstream to shed light on the historical observed coordination, and lack thereof in the simulations, during the 2012-2013 low-flow period. The pace and magnitude of the drawdown are the defining differences between historical and simulated operations at American Falls. For both years, historical operations show reservoir levels decreasing at a near-constant rate from nearly full in early May to about $5-10\%$ by the end of summer. The drawdown season spans 4-5 months and the reservoir

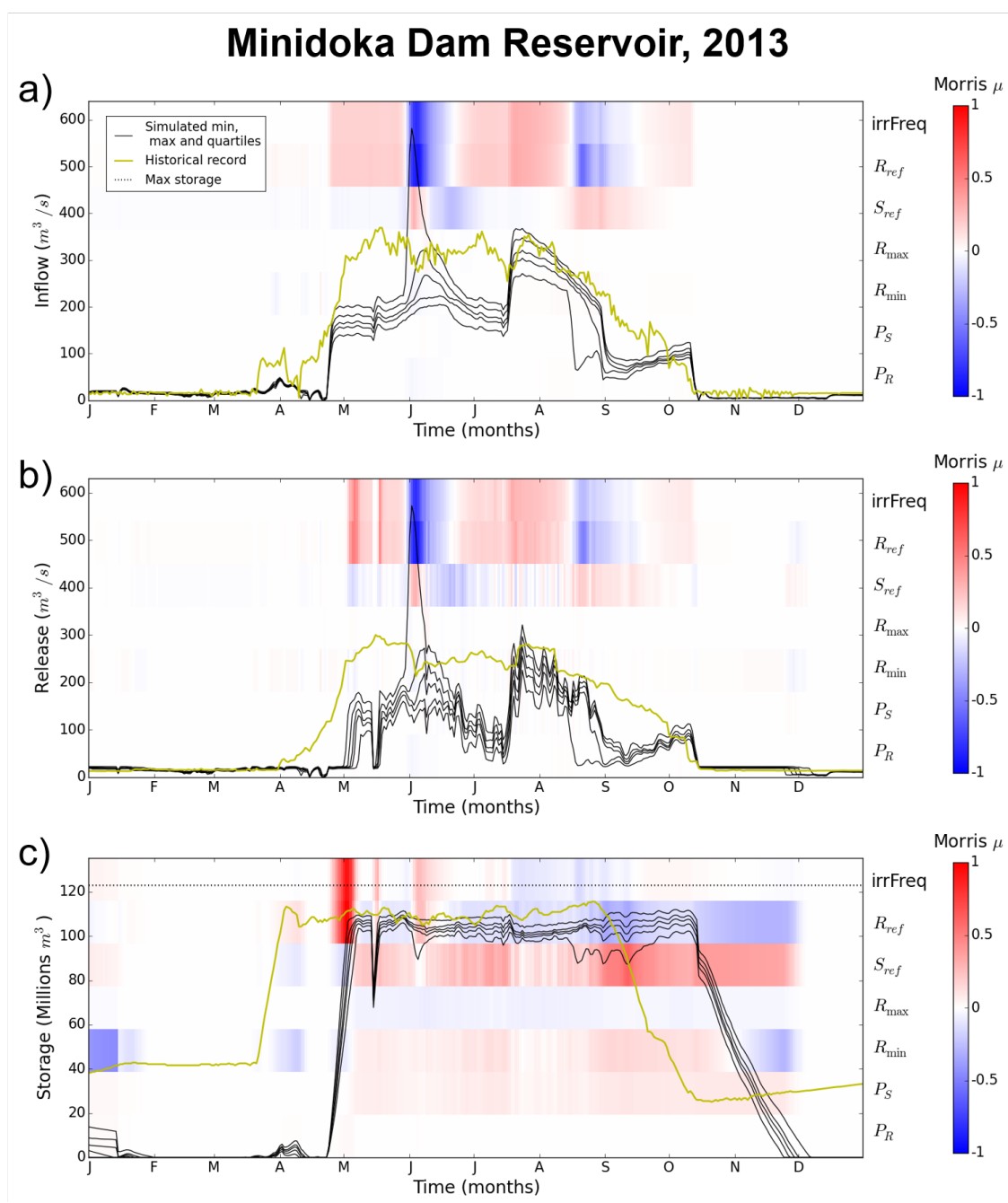

**Figure 7.** Simulated values (max, min and quartiles, shown with black lines) with historical values (gold line), and sensitivity to input variables (background), for (top to bottom): Minidoka reservoir's inflow, release and storage.

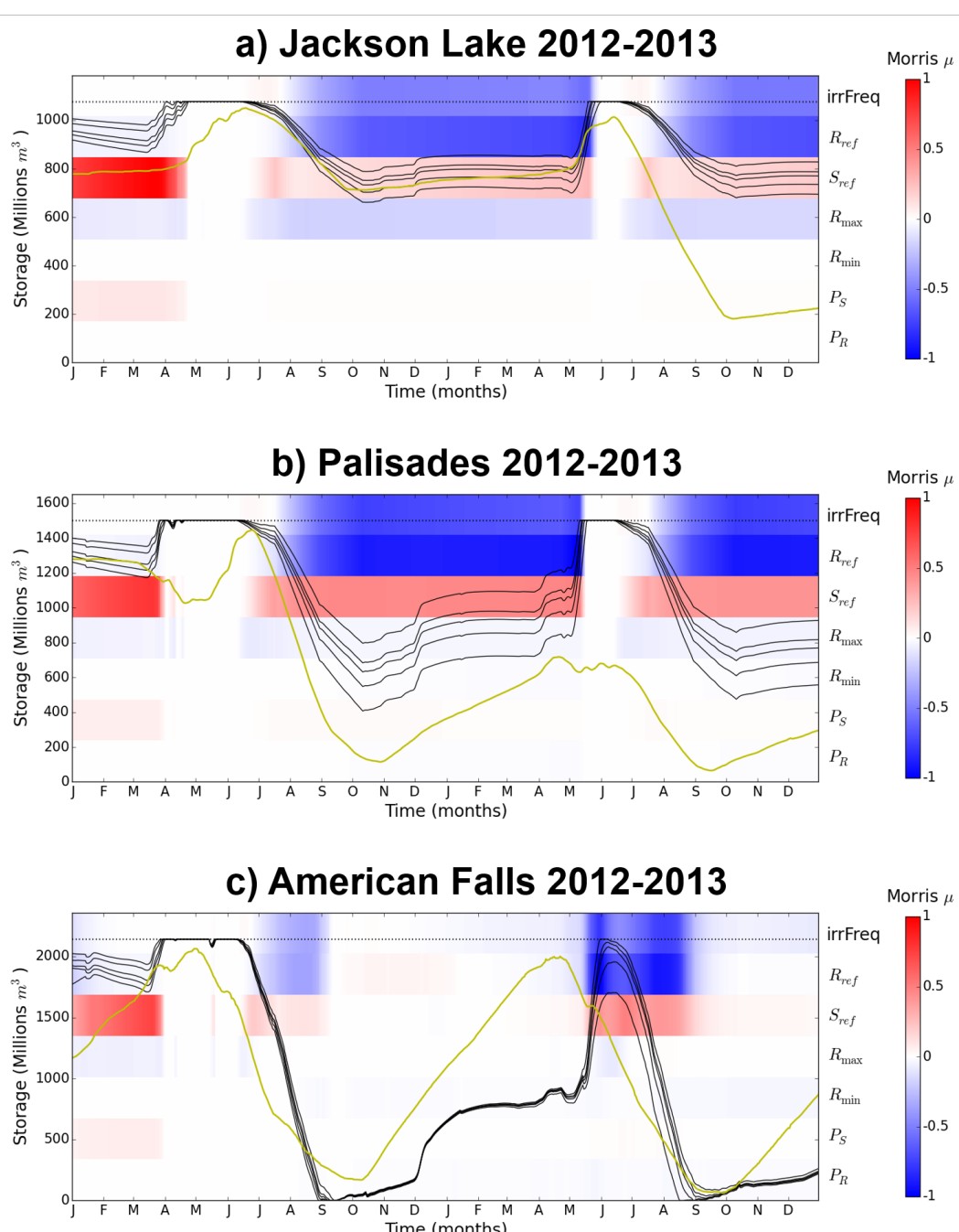

**Figure 8.** Recorded and historical storage during the dry years 2012 and 2013 with sensitivities of the simulated variables illustrated in the background. From top to bottom: storage at three largest USRB reservoirs from upstream to downstream.

never loses its capacity to regulate downstream reservoir levels. Alternatively, simulated drawdown seasons are much shorter – two and half months from mid-June to the beginning of September – and the reservoir swings from full (in 2012) or nearly so (in 2013) to completely empty either for the whole ensemble (in 2012) or nearly half of it (in 2013). In other words, American Falls loses its capacity to regulate irrigation delivery or is simulated to be dangerously close to doing so.

The reason for this contrasting behavior can be found with upstream operations. For instance, the historical storage trajectory at Palisades (panel (b)) shows a marked drawdown from early July to late October 2012. On average, the reservoir released over $0.5\ km^3$ more towards American Falls in the observed record than it does in the simulations, and this enabled American Falls to keep its capacity to regulate irrigation withdrawals on the Snake River plain. Simulated storage sensitivities however, reflect the lack of coordination across the ensemble of simulations. Indeed, Method of Morris results show that the main controls on

storage from April 2012 onwards are the same as for large reservoirs for which at-site controls dominate upstream interferences (see section 4.1). These controls, and the simulated storage trajectories, fully ignore any connection with the simulated events unfolding downstream.

Yet, in 2013 historical storage levels at Palisades (yellow line on panel (b)) had not recovered from the exceptional 2012 drawdown due to a combination of low carryover storage and insufficient snowmelt. Palisades reservoir could no longer supply

extra water to the Snake River Plain. Instead, exceptional historical Jackson Lake drawdown in the summer of 2013 (panel (a)) supplied over $0.5\ km^3$ extra water to the Snake River Plain compared with what the simulations record. Thus, complex multi-year and multi-reservoir coordination was needed to avert adverse drought impacts on agriculture. The simulations do not account for this coordination, as demonstrated by both simulated storage and by the consistent parametric controls at both Jackson and Lake and Palisades for both years. The ensemble of simulations let American Falls empty whereas the two largest

reservoirs upstream of it remain close to full.

### 4.4   Flood risk

We next evaluate if these representational deficits in simulating coordinated operations also yield consequential errors in the Spring of 2011, where the observed operations averted a flood by exploiting forecast-based anticipatory releases in the two upstream large reservoirs at Jackson Lake and Palisades. Following the flow from upstream Jackson Lake to downstream

Palisades (Table 2), we contrast coordinated historical storage and discharge levels observed in the Spring of 2011, with the simulations results from our ensemble of hydrological model runs and the associated release and storage sensitivities to WBM's parametric controls.

#### 4.4.1   Jackson Lake

Starting upstream, we focus on the storage and release dynamics, both simulated (black lines) and historical (gold lines), at

Jackson Lake (Figure 9). All simulation results fill the reservoir entirely between May 14 at the earliest and May 26 at the latest (panel (a)); this period coincides with maximal release sensitivity (panel (b)). Note that the Method of Morris found that the dominant controls on simulated release during May 14-26 (panel (b)) are the same as the dominant controls on simulated storage prior to that period (panel(a)), with strong negative sensitivities to $R_{ref}$ and $irrFreq$ and strong positive sensitivity to

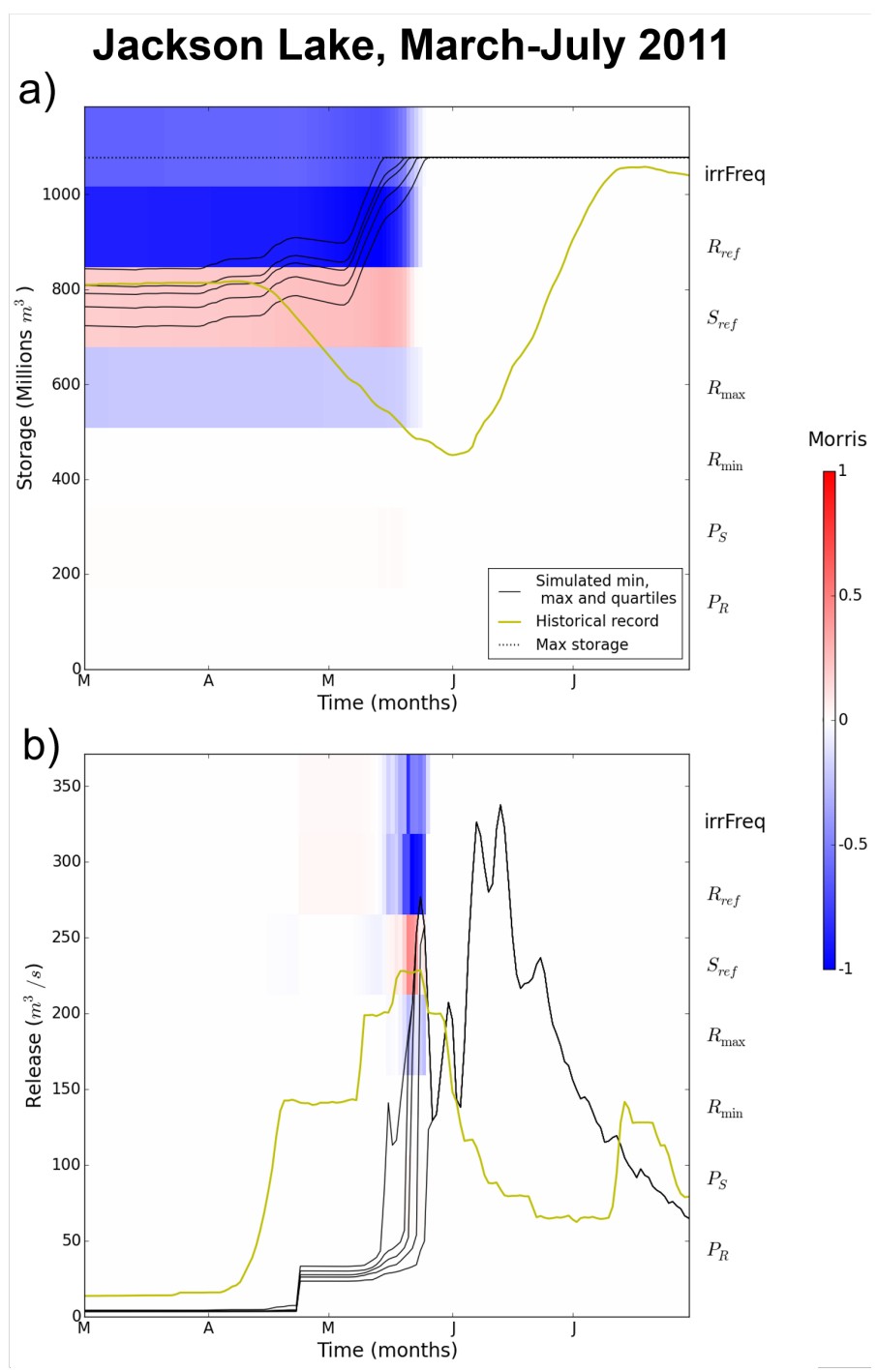

**Figure 9.** Simulated values (max, min and quartiles, shown with black lines) with historical values (gold line), and sensitivity to input variables (background), for Jackson Lake's release and storage.

$S_{ref}$. The dominance of these three parameters corresponds well with our prior results, as detailed in Section 4.1. These results however run contrary to the real-world expectation that for reservoirs with no upstream interactions, release rule parameters should influence release and storage in opposite directions.

This is because during the snowmelt-driven peak flow season, higher simulated storage leads to quicker reservoir filling which takes away the reservoir's capacity to regulate peak flows. Once the reservoir is full, simulated peak releases out of Jackson Lake are much higher than the historical observed releases. These have been mitigated by real-world reservoir operators who started releasing water in early April to decrease reservoir storage by almost half between then and early June. This created enough storage space to absorb runoff from peak snowmelt season in June, while simultaneously reducing releases to limit the reservoir's contribution to downstream high flows. By contrast, all simulated releases only increase gradually when the reservoir gets close to full capacity. Due to this lack of foresight-driven preventive releases in the simulations, Jackson Lake is full by the end of May and unable to absorb peak flow in June. This represents a large and consequential structural error in model's representation of flooding operations and vulnerabilities.

### 4.4.2 Palisades

Moving to the next reservoir downstream, Figure 10 illustrates the simulated (black lines) and historical (gold lines) storage and release dynamics for the Palisades reservoir in March-July 2011. All simulation results fill the reservoir entirely between May 5 at the earliest and May 9 at the latest (panel (a)). Similar to Jackson Lake, this coincides with a period of maximal release sensitivity according to Method of Morris results. The dominant controls on both storage and release are identical: same parameters ($irrFreq$, $R_{ref}$ and $S_{ref}$) with the same directional effects. Put simply, parameters that favor reservoir filling in simulations diminish Palisades reservoir's capacity to store water and to absorb peak snowmelt season flows, leading to heightened simulated releases. Since Palisades is downstream of Jackson Lake and snowmelt occurs earlier at lower altitudes, simulated filling occurs earlier, and consequently the WBM abstraction of the reservoir is subsequently unable to absorb snowmelt peaks, including the one event occurring May 24-26 as the result of Jackson Lake filling. This is evidenced by parametric release sensitivities and the concurrent simulated release peak (panel(b)) around these dates that necessarily come from upstream – there is no on-site release sensitivity when Palisades is full.

By contrast, historical operations favored preventive releases as early as the end of March at Palisades, to free up almost $1.3$ $km^3$ of storage space by early May – precisely at the time when the onset of snowmelt fills the reservoir up in simulations. This leaves over $1.1$ $km^3$ of storage space by early June, and the comparison of Figures 9 and 10 shows that both Jackson Lake and Palisades filled at a near-constant pace throughout June, nearing being completely full around July 10. This controlled and coordinated filling of both reservoirs ensured that releases well below $700$ $m^3/s$ at Palisades, a full $900$ $m^3/s$ lower than the simulated peak across virtually all of the simulated ensemble. The simulated peak is almost $40\%$ higher than the highest observed daily discharge over the past 40 years. Maximal Palisades release over this period – $1140 m^3/s$ on the 20 June 1997 – corresponds to a flooding event which led to six counties declaring a state of disaster, leading to over USD 11 million in relief by the federal U.S. government (National Oceanic and Atmospheric Administration, Retrieved 13 April 2020). Coordination is mediated by seasonal forecasts based on snowpack height, and is apparent through the reduction in Jackson Lake release

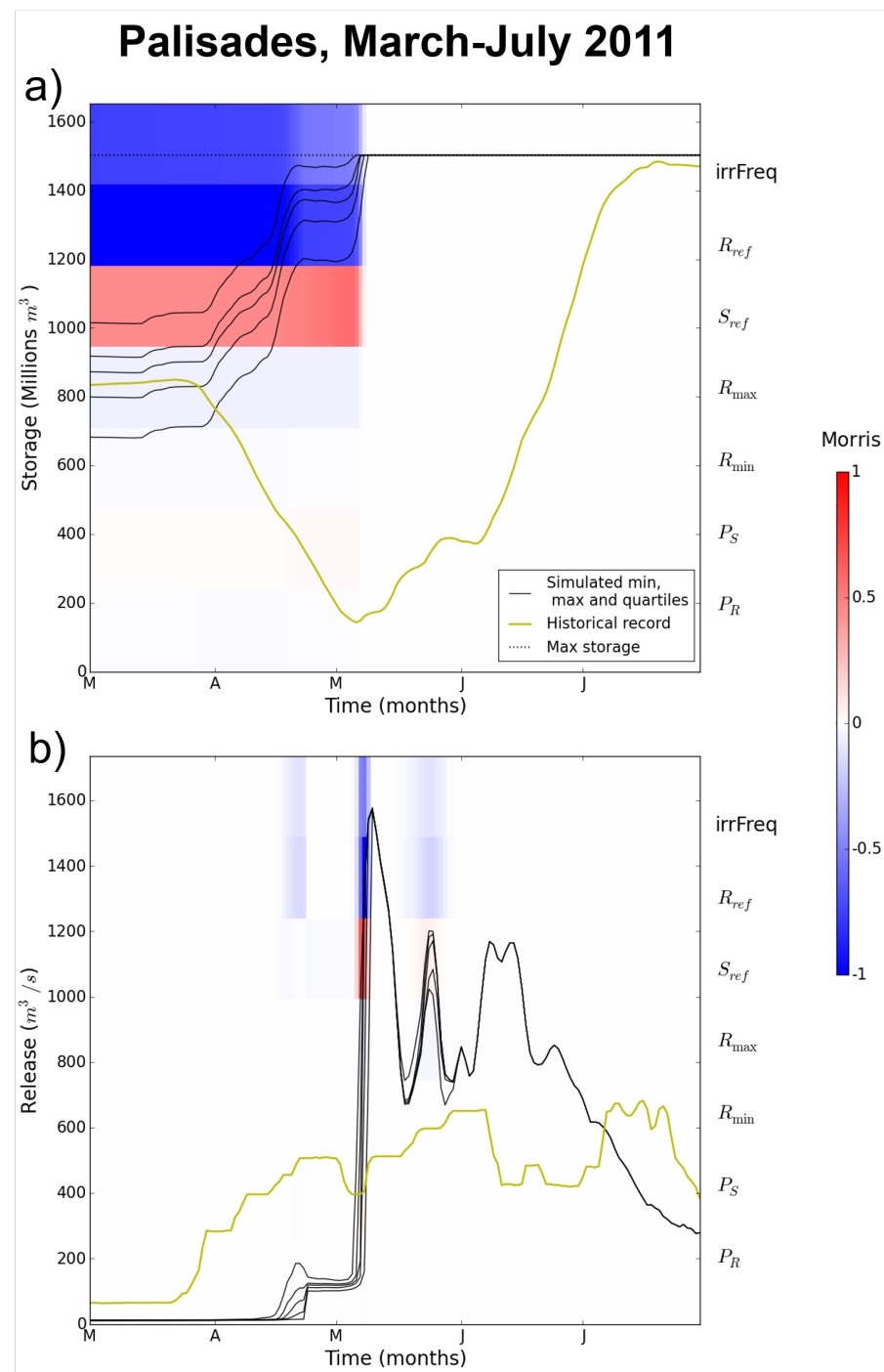

**Figure 10.** Simulated values (max, min and quartiles, shown with black lines) with historical values (gold line), and sensitivity to input variables (background), for Palisades reservoir's release and storage.

| Year | | 2011 | 2012 | 2013 |
|---|---|---|---|---|
| Risk | | Flooding | Water shortage | |
| Key variable | | Max Palisades release | Min American Falls storage | |
| Historical record | | 682 $m^3/s$ | 170 $hm^3$ | 63 $hm^3$ |
| WBM simulations | (ensemble median) | 1573 $m^3/s$ | 0 $hm^3$ | 19 $hm^3$ |
| | (worst case) | 1578 $m^3/s$ | 0 $hm^3$ | 0 $hm^3$ |
| Offline water balance | (ensemble median) | 582 $m^3/s$ | 314 $hm^3$ | 395 $hm^3$ |
| | (worst case) | 747 $m^3/s$ | 272 $hm^3$ | 357 $hm^3$ |

**Table 3.** Comparison of key variables for both the 2011 flood event and the 2012-2013 drought event, for the historical record (displaying coordination between reservoir), the hydrological model (no coordination), and the offline water balance (modifying model outputs with simple coordination rules).

(Figure 9 panel (b)) when Palisades starts filling back up. As a result, neither reservoir ever loses its capacity to regulate streamflow by filling completely, and that downstream releases are capped. The simulation is strongly inconsistent with the institutional flood control objectives of the reservoirs (U.S. Bureau of Reclamations, 2012).

### 4.5 Offline water balance experiments

For both the low-flow and high-flow events, our analysis reveals how the absence of simulated coordination between the reservoir of the USRB cascade results in artificial erroneous water shortage and flooding. The actual operational observations capture upstream to downstream coordination in storages and releases that enabled real-world operators to avoid these outcomes. To support our hypothesis that coordination is the difference between modeled and observed outcomes, our offline water balance experiments add simple coordination mechanisms that quantitatively mimic the real-world observations. The

water balance models take offline inflow, release and storage trajectories for each simulated ensemble member from our global sensitivity analysis during the events of interest. The coordination mechanisms we add depend on the event and are described below. Overall our addition of simple coordination rules (Table 3) show that for both events, coordination is enough to avoid the false modelled flooding in the 2011 event, and erroneous water shortage in 2012-2013.

### 4.5.1   2011 flood event

We develop a simple offline water balance model for the two flood control reservoirs (Jackson Lake and Palisades), with the inflow, release and storage trajectories from every simulated ensemble member. To simulate observed coordination, we replace releases with a simple policy starting the last week of March – matching the timing at which operators started emptying Palisades. Palisades release is set a full $100m^3/s$ lower than the maximum historical daily release of $682m^3/s$, and a policy is set at Jackson Lake to match observations from Figure 9). Releases are set to be 1) $200m^3/s$ when Palisades is empty enough

(less than $40\%$ full) or Jackson Lake is nearly (over $98\%$) full, and 2) cut release back to $50m^3/s$ otherwise. The routing delay was fixed at one day, a conservative assumption making any excessive release from Jackson Lake immediately consequential

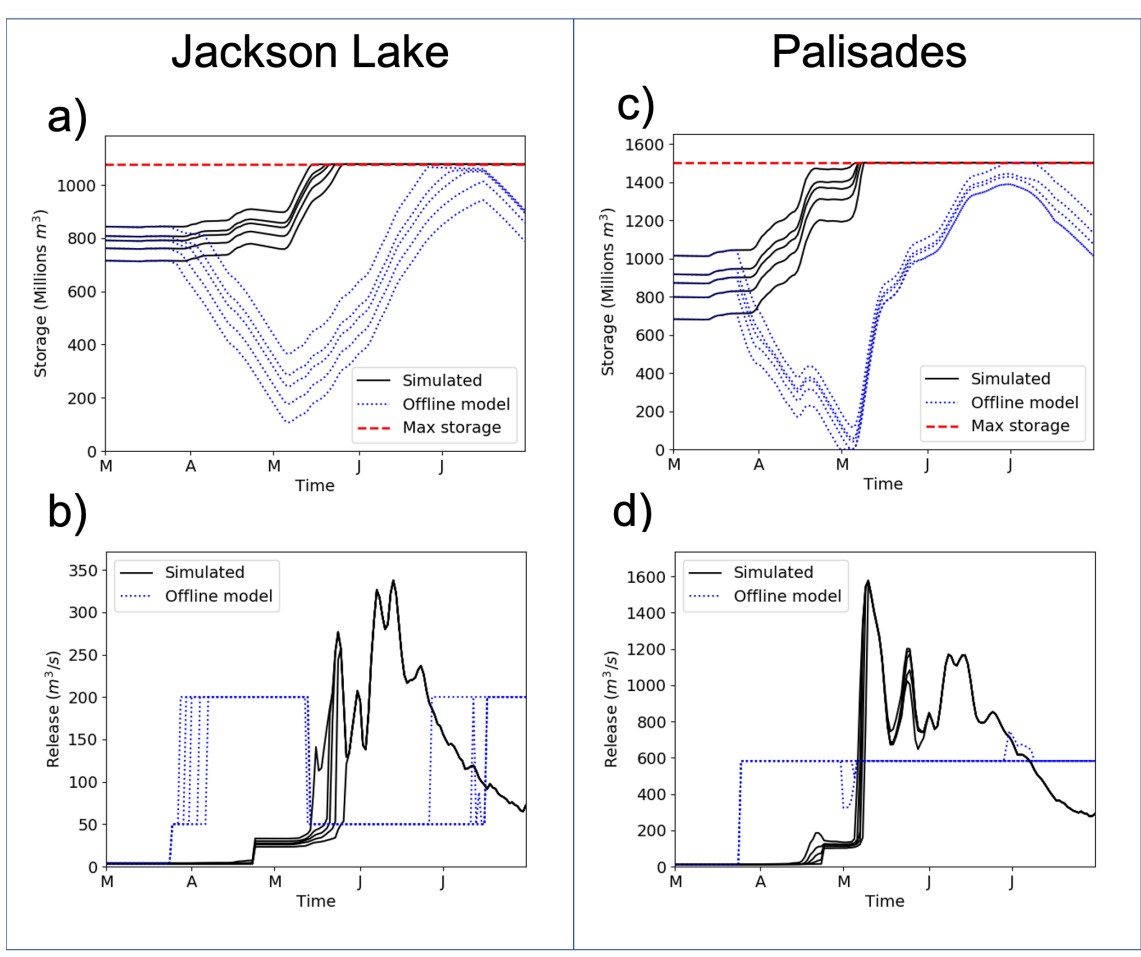

**Figure 11.** Results from the coordinated offline reservoir water balance model (dotted blue lines), compared with hydrological model simulation results from non-coordinated operations (continuous black lines): min, max values and quartiles for both ensembles. Differences in storage (panels (a) and (c) for Jackson Lake and Palisades respectively) are due to a simple coordinated released policy starting the last week of March (panels (b) and (d) for Jackson Lake and Palisades respectively).

for Palisades reservoir levels. Table 3 shows that including a simple coordinated flood control policy is enough to eliminate the flood peak obtained in model results. Figure 11 shows that coordination enables to avoid filling Jackson Lake across the whole ensemble, and avoids filling Palisades in most cases. The only ensemble members for which Palisades gets filled are the ones that start with much higher initial storage at both reservoirs; even then, filling only happens in late June and peak flows are less than half those simulated without coordination.

### 4.5.2 2012-2013 drought event

For both 2012 and 2013, we take model results from Jackson Lake, Palisades and American Falls reservoirs offline for each of the 400 ensemble members. We then simulate operations obtained by releasing an extra $50 m^3/s$ during the summer than planned according to release rules, as long as the reservoir is over $20\%$ full. The total extra volume thus released is consistent with the difference between historical and WBM-based simulated releases. Similar to observations from Figure 8, this policy concerns only Palisades in 2012, and both Palisades and Jackson Lake in 2013. To make sure that underestimating routing times between reservoirs does not falsely cause the reservoirs to store water, we choose a conservatively high (for the area) routing time of 7 days between each reservoir and the one downstream. Results displayed in Table 3 show that with this simple coordination water balance measure, American Falls would not have emptied in either year and across the full ensembles. Besides, reservoirs upstream of it would not have lost their capacity to supply downstream agriculture either (see Supplementary Information Figures 1 and 2).

## 5 Discussion

This work analyzes a state-of-the-art release rule from a large-scale, high-resolution hydrological model to understand the potential consequences of not capturing real-world operational coordination across reservoirs when simulating flood and drought events. It focuses on the USRB, a Western U.S. basin featuring a reservoir cascade managed with a high level of coordination to avoid both floods and water shortages, two risks made prominent by the area's geography and climate. An ensemble simulated and analyzed using the screening method known as the method of Morris provides evidence that parameterizing each reservoir in a cascade independently of the others is an approximation. This assumption implies that reservoirs are not coordinated, which has unintended consequences as our work showcased 1) a quick and complete drawdown of reservoirs in irrigation hotspots during hot, dry summers, and 2) the simulation of potentially catastrophic floods with untimely filling across the cascade. The historical record, and experiments based on offline water balance models of the reservoir cascade, demonstrate that in both instances, coordinated reservoir management avoided the occurrence of these events.

In both the high-flow and the low-flow events, coordination and control decisions are mediated both by other reservoirs' operations and by other decision-relevant variables. This is obvious for the averted flood of 2011 where snowpack monitoring led to forecasts of large snowmelt with enough lead time to make space in two key reservoirs and coordinate their response. Similarly, the 2012-2013 decisions are mediated by water demands in the Snake River Plain. In both cases, the mix of institutional communication – between reservoirs and farmer representatives – as well as monitoring of key water supply and demand predictors are instrumental to implementing successful coordination actions in the face of adverse climatic events. Recent research on the water management institutions of the Upper Snake River basin suggests that they are well-equipped to show resilience in the face of expected climate change (Kliskey et al., 2019; Gilmore, 2019).

There is a growing body of literature highlighting the potentially highly interdependent nature of state-aware reservoir operations and institutional coordination in large multi-purpose reservoir cascades (Quinn et al., 2019). The importance of institutional context as well as location specific nature of selecting key variables for informing forecasts is a significant challenge

to large-scale hydrological modeling. Poor abstractions of forecast informed reservoir operations and basin specific institutions that support coordinated emergency responses limit the value of hydrological modeling in understanding vulnerabilities to extremes. In a context where high-resolution modeling (Wood et al., 2011; Bierkens et al., 2015) is framed as a key element for informing, monitoring and forecasting these risks at exquisitely fine spatial and temporal resolutions, it is urgent to move beyond validation based exclusively on goodness-of-fit. Model evaluations need to 1) identify key human and natural processes leading to flow extremes, and 2) validate that these processes are present in the hydrological model. As recent developments in the literature on reservoir representations in hydrological models illustrate, there has been a growing sophistication in representations of release rules without addressing the key concern of capturing the key variables managers use to address unusual flow conditions in complex coupled human and natural systems. Parametrizations that are "good" in the sense that they score well with respect to one or more goodness-of-fit indicators may not necessarily represent the underlying processes correctly (e.g., Legates and McCabe Jr., 1999; Gupta et al., 2009). This point has recently been illustrated for reservoir representations in large-scale hydrological models through the flawed structural behavior of an upper Mekong (Lancang) basin model where reservoirs had been omitted (Dang et al., 2020). This is why we did not attempt to calibrate reservoir rule parameters in this work. Besides becoming an increasingly difficult task going downstream, it would only have served to mask a portion of structural model errors without actually addressing them (a.k.a "right for the wrong reasons"). To the contrary, this paper takes the view that the unintended consequences from these errors need to be exposed before "well-calibrated" but structurally deficient representations are used to assess out-of-sample flood and drought risks with future flow conditions that are often very different from those used for model calibration and evaluation. In this case, we exposed the need to refine representations of human-mediated coordination and controls in hydrological models, so they do not flag false vulnerabilities in a world where rapidly-developing global crises are expected to yield large capital investments.

Approaches to address this need will have to contend with trade-offs between the quality of multi-reservoir operations modeling, computational costs, and data availability (Masaki et al., 2017). The most straightforward way to represent complex human coordination processes and the key variables they rely on is to integrate actual management rules directly into hydrological models (Zagona et al., 2001; Yates et al., 2005). Such rule systems demonstrably improve hydrological models (Qiu et al., 2019), but they necessitate a direct knowledge of operations that is unavailable in most cases. Alternatively, machine learning techniques have been developed to infer reservoir operator behavior from historical observations, but often assume that decisions are taken as a function of a set of standard hydrologic variables on a reservoir-by-reservoir basis (Hejazi et al., 2008; Ehsani et al., 2016; Coerver et al., 2018; Turner et al., 2020). Recently though, applications to multi-reservoir systems in California have seen these techniques extended to consider impacts of forecast variables such as snowpack depth on operations (Yang et al., 2016), and to infer drought vulnerability from monthly operations (Giuliani and Herman, 2018). Our work demonstrates that further research is needed in this direction to fully account for complex feedbacks between climate variables, water supply and flood control objectives, and release decisions. Emerging techniques enabling storage level monitoring even in inaccessible areas including war zones (Müller et al., 2016; Avisse et al., 2017) could then make it possible to generalize machine learning-based approaches.

An alternative to reproducing historical operations is to improve operations through optimization instead. Such optimization needs to consider the distinct and sometimes conflicting management objectives, including but not limited to protection against water shortages and floods. Using the example of a single reservoir with multiple commitments in terms of flood control, water supply and hydropower production, Giuliani et al. (2014) showed that the multiple vulnerabilities associated with historical operations could be mitigated using multiobjective heuristics. This emerging approach, called evolutionary multi-objective direct policy search (EMODPS Giuliani et al., 2016), proposes reservoir rules that trade-off flood and drought vulnerabilities with other reservoir management objectives. It has been successfully applied to a flood- and drought- prone multi-reservoir system (Quinn et al., 2017). Yet, this approach is also computationally expensive and needs to use offline water balance models to parameterize parsimonious reservoir rules that can be input into large-scale hydrological models.

## 6  Conclusions

The interactions between the multiple stakeholders in major river basin systems are complex, as is the interplay between the key variables they use to monitor and manage flood and drought risks. Although large-scale hydrological models have generally sought to abstract this complexity in their representation of human processes, they at present struggle to capture coordination and control processes in multi-reservoir systems. The current standard practice treats each reservoir's release independently from other reservoirs' storage levels. This paper demonstrates the unintended consequences this can have for flood and drought assessment, using a well-established hydrological model with advanced representations of multi-reservoir operations in the Upper Snake River Basin. Our diagnostic assessment of a state-of-the-art release rule abstractions in large-scale hydrological models exploits time-varying sensitivity analysis based on the Method of Morris to show how the behavioral controls in parameterized reservoir representations can inadvertently lead to amplifying errors. The diagnostic methodology used, which combines the Method of Morris with real-world observations and offline water balance modeling along a reservoir cascade, can be replicated with other hydrological models, release rule representations, and river basins. It provides insights of cumulative reservoir rules impacts across the basin at a daily time step. Application to the reservoir cascade on the Upper Snake River Basin, with its complex institutions and careful monitoring of frequent flood and water risks, showed how failure to represent the appropriate monitoring and reservoir coordination processes could lead a hydrological model to simulate flood and drought events that actual basin operators would unequivocally avoid.

This finding is consequential at a time where reservoir rules of increasing sophistication are being proposed to come to a better agreement between observed and simulated releases, and where the monitoring and forecasting of water-related risks at extremely high resolutions is hailed at the future of hydrology. It demonstrates the necessity to complement goodness-of-fit testing by devising validation techniques checking for the structural behavior of human-operated structures in the face of water emergencies. This is not a task for hydrologists alone, as developments across water resources management, operations research, machine learning and assimilation of remotely sensed data, among others, all have a role to play in tackling the challenge whose urgency this work highlights.

## Acknowledgments

*Competing interests.* The authors declare that they have no conflict of interest.

*Author contributions.* PMR and RBL obtained funding for the study. DSG, SZ, AP, SG and RBL developed the WBM model for the USRB this study used. DSG, SZ, AP and SG enabled the porting of the WBM model to CR and PMR. CR, PMR and JRL contributed to the design of the analysis. CR performed the analysis under the mentoring of PMR. CR wrote the paper and led revisions in close collaboration with PMR and with contributions from DSG, SZ, AP, JRL and RBL.

This work was supported by the U.S. Department of Energy, Office of Science, Biological and Environmental Research Program, Earth and Environmental Systems Modeling, MultiSector Dynamics, Contract No. DE-SC0016162 and the material based upon work supported by the National Science Foundation under Grant No. 1639524 in the Innovations at the Nexus of Food, Energy and Water Systems (INFEWS) program.

*Code and data availability.* Core WBM code is available from the authors by request, and so is the code for this sensitivity analysis. Result data and code necessary to draw the figures are available online at https://github.com/charlesrouge/UpperSnakeRiver_reservoirs_WBM.

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
