# Peer review of "Coordination and Control: Limits in Standard Representations of Multi-Reservoir Operations in Hydrological Modeling"

_Hydrology and Earth System Sciences, 2019_

## Referee Comment (RC1) · Anonymous Referee #1 · 10 Jan 2020

This study deploys the Method of Morris to evaluate the sensitivity of release and storage time series to adjustments in the parameters of generic reservoir operating rules. The analysis is performed using a high resolution hydrological model (WBM) and focuses on a cascade of reservoirs located in the Upper Snake River Basin. The paper is well written, with clearly defined methodology and easy-to-follow results section. The experiments conducted are ill-suited to the aims of the study, leaving too many confounding factors. Specifically, I disagree with the authors key claim that the approach exposes coordination between reservoirs missing from the generic operating schemes (see specific comments below). While I do not doubt that coordination does occur in this particular cascade system, I cannot see how this sensitivity analysis exposes

that coordination unequivocally. I also have some concerns with the study design, and in particular the use of the high-resolution hydrological model, which introduces a severe and unnecessary computational constraint while also supplying the reservoirs with (likely) inaccurate historical inflows. The latter leaves the reader unsure as to whether the difference between simulated and observed storage/release is more a function of erroneous inflow than inaccurate operating decisions. This is particularly important when demonstrating the inadequacy of the reservoirs rules in representing flood and drought (it could simply be that the upstream hydrology from the model is delivering the wrong volumes of water). Since the actual observed inflows are available for this system, it would seem far more prudent to develop a simple, offline cascade model. I suspect such a radical change to the experimentation at this stage would be unrealistic, so I would instead encourage the authors to change the aims and storyline offered here. The simulations are sound and there is a great need for the community to learn more about the nature and performance of generic reservoir schemes. I would support a revision if either (a) the authors can convincingly rebut my concerns listed below, or (b) a new angle is developed with a more defensible conclusion.

Specific comments to authors:

Section 3.2. The justification for the 10% decision is unconvincing. Suppose a dam has an average inflow of 100 cumecs. An Rmin of 0.1 would be 10 cumecs and would vary by a maximum of 1 cumec. So the left-hand side of the operating curve hardly moves. In contrast, if your Rmax is 5 (=500 cumecs) then you'd vary this parameter by plus or minus 50 cumecs. So the right hand side of the curve will shift wildly in comparison. This is surely why Rmin appears to be unimportant in your sensitivity analysis; you've barely moved it. I appreciate that there are physical reasons why Rmin would be expected to be somewhat less variable than Rmax in absolute terms (although both are highly uncertain), but the uniform 10% assumption is not ideal either, and it invalidates your later statements about which parameters constitute the signature of parametric influence (page 15).

[Figure]

Section 4.2 through 4.4. The results described here show that the operations of upstream dams can affect the decisions made at downstream dams. This occurs because any change upstream has an effect on the inflows into a downstream dam, thus affecting its storage levels and therefore its releases (which are a function of storage). Sensitivity analysis exposes how the decisions taken at one dam are affected by the rules deployed at other dams (often with some intriguing complexities). This is insufficient, however, to claim that the models must be missing *coordination* (which presumably means dam operator A looking at the storage levels of dams B and C to inform his or her decision). Similarly, the results that describe inadequate representation of flood and drought mitigation *could* be caused by uncoordinated reservoir schemes. But inadequate mitigation could also be simply because the operating schemes at individual dams are insufficient. It's possible that more realistic operating schemes at the individual dams (i.e., not necessarily incorporating coordination, but with more realistic structure and/or parameterization) would provide the correct mitigation responses. It's also possible that the failure of these models to represent flood and drought mitigation is partly caused by bias in your inflow data (which is not shown or compared to observed at the upstream dam). Given these possibilities, the proposed framework fails to demonstrate unequivocally that a missing piece of the reservoir model is coordination.

Other minor comments:

Figure 1. Difficult to interpret due to color scheme used.

Table 1. Please provide some additional justification for these parameters. Has there been any validation done? Why would the Rref for a water supply reservoir be 0.1? This parameter should vary widely across water supply reservoirs as a function of demand relative to inflow (and 0.1 is very low). Generally I don't see why the purpose of the reservoir would control these parameters. Rmin (and indeed the whole left-hand section of the operating curve) would likely be strongly determined by environmental flow requirements, which are independent of reservoir purpose. I would also suggest

discussing the omission of seasonality in the operational parameters (which could apply to non-irrigation reservoirs). If there is no good justification for these rules then this is ok (it's not the purpose of your experiment to defend the status quo)—in this instance just make a statement to inform the reader.

P6 line 1 - reservoir data *were* derived...

P6 line 1 - not clear who performed these manual updates

P6 line 7 - How accurate are these demand data compared to USGS estimates, which are US specific.

Section 2.1. I hadn't studied the Method of Morris previously, but this is an excellent description that educates the reader.

P16 line 7 - The high variability in release (relative to observed) is surely just caused by the right-hand side slope of the operating curve being too steep, or the Sref being too low. Why should this point to incorrect representation of coordination?

P16 line 27 - this may have been a low flow event, but your figure for Jackson shows clearly that the drawdown was in large part caused by sustained high release through 2013.

It would be helpful to see inflow, storage, and release graphs for the full time series for all reservoirs (perhaps in supplemental material).

---

## Referee Comment (RC2) · Anonymous Referee #2 · 10 Jan 2020

This article provides an evaluation of the consequences of the lack of a representation of reservoir coordination within a multi-reservoir system when simulating flood and drought events in large-scale hydrological models. The model Water Balance Model simulates a multi-reservoir system in USA. The model includes the representation of each reservoir operation policy (using predefined parameters according to each reservoir purpose) but it does not represent the coordination between reservoirs. The global sensitivity analysis Method of Morris is used to assess the effect of the parameterization to the model outputs. Authors conclude that the representation of reservoir policies independently is not enough and that, in addition, we need to capture reservoir coordination in large-scale models to properly simulate flood and drought effects

in multi-reservoir systems.

The article is well written and structured. The Introduction makes a good review of the hydrological impacts of multi-reservoir systems and previous attempts in representing reservoir systems in hydrological models. The methodology is well defined with the exception a few aspects that need further explanation. The article does not explicitly say where the parameters of the reservoir rules (Table 1) come from. Moreover, the authors do not specify the parameter ranges. If the values in Table 1 were obtained by calibration in a previous work, the authors could show the ranges applied in that calibration or just reference that work. If there is no previous calibration, how the predefined values of the parameters produce a good agreement between observed and simulated storages and releases (e.g. Figure 5) in normal climate conditions? The results and discussion are also clear and well structured but there is a lack of discussion of how the methodology applied here can be used by others. I was wondering if this could be done using a different model where the parameters of the reservoir rules are unknown and need to be obtained by calibration. Lastly, I think that the paper needs further and clearer discussion on why the lack of representation of reservoir coordination is most likely to be the main reason of this failure to simulate flood and drought events.

In conclusion, this paper makes a relevant contribution to the growing discussion around the representation of reservoir systems in hydrological models and it has clear practical implications. The authors provide practical recommendations and possible solutions. While the representation of reservoir coordination is still very difficult to implement in models, this study highlights its importance and the need to, at least, consider this limitation when modelling catchments containing reservoir systems under extreme conditions.

Other comments:

- While the authors provide a justification for the 10% range used for the sensitivity analysis, in my opinion, it would be interesting to also show what range of variation

around base values should be applied to properly simulate any of the drought and flood events.

- Page 15, Lines 17-19: What if the releases were represented as cumulative releases, the sensitivity would be as consistent as for the storage?

- Page 21, Lines 2-4: Could you please provide with a possible explanation to this unexpected result?

---

## Referee Comment (RC3) · Anonymous Referee #3 · 16 Jan 2020

The manuscript presents a new large scale reservoir operations model with generic operating rules associated with the reservoir main operational purpose such as flood control or irrigation, or default. The reservoir model stands out from equivalent models in that the releases are decided daily based on the daily storage level, shapes with combined log and exponential curves that accelerate the release in times of floods when close to full capacity and slows down the release in times of droughts with differences in the thresholds and propensity to release and store based on the purpose of the reservoir. The overall release is scaled by the long term mean annual flow. The model is implemented at high resolution ( .6 km, daily time step) over the Upper Snake River Basin, which is a snowmelt driven basin. The method of Morris is used to identify

the reservoir release parameters that tend to be most influential in the reservoir release and storage variations throughout an 8 year period. Upstream reservoirs are used to evaluate the approach while downstream reservoirs are used to evaluate the impact of upstream reservoirs. A flood and drought events are evaluated with respect to observed operations to categorize the error associated with the lack of representation of reservoir coordination. Authors conclude that reservoir coordination is needed to represent flood and drought in typical reservoir models, and that optimization of rules with foresight would help in this endeavor. All simulations were performed on very high performing computational resources taking 2 days for 8 year simulation over the Upper Snake River Basin.

The subject is very interesting for the HESS community and the manuscript is well written but there are a number of concerns that would need to be addressed before consideration for publication. The main concerns are about the two (great) highlights of the paper : the new model and the time sensitive analytics; i) the manuscript presents a new large scale model, with a very interesting concept for the releases that is however not enough evaluated and discussed, and ii) the approach to quantify the contribution of reservoir coordination to better represent floods and droughts needs to be improved – it is based on inference statements and the model could be modified to include information about upstream reservoir release to demonstrate the point about coordination. Minor feedbacks are that the reference to typical reservoir model is misleading and the analytics with the method of Morris is very hard to follow.

1) Reference to typical reservoir operations model seems misleading. At the scale of the Upper Snake River Basin, typical reservoir operations models have a nodal architecture and represent accurate reservoir operating rules that can be revisited in optimization mode and especially in forecast mode to mitigate reservoir and drought events. The manuscript here refers to very large scale spatially distributed reservoir models that have been developed initially to be fully coupled with hydrology model and research land-surface-atmosphere interactions. Those models are typically applied

over multiple independent large river basins. I would suggest to not refer to typical reservoir model where most of the community understand reservoir models where rules can be optimized and are applied to one basin at a time. Please refer to large scale distributed reservoir model or equivalent differentiation from nodal operational reservoir models.

2) A new large scale reservoir operations model : please provide more details - what is the river routing process for this high spatial resolution and daily time step? A recommendation in the introduction is not to aggregate reservoir storage but many reservoirs have less than 2 days in travel time. How does the reservoir model decision release algorithm adjust stability?

- How are the 6 parameters initialized? Are the necessary data widely available? What are the assumptions?

- Evaluation of the smoother release curves with other models. In other equivalent models that are cited (Hanasaki, Doell, Biemans, Voisin, etc) , releases are decided daily based on reservoirs minimum and maximum capacities, minimum environmental flow and tend to follow monthly storage and releases targets with no foresight, but using long term mean monthly inflow, which also tends to be regulated or natural flow depending on the models. What is the improvement for those rules? The obvious features are the changes in release rates - how does it improve the flow representation in general?

- reservoir coordination. Note that the use of a rolling past 20-year of mean monthly regulated inflow provides a minimum of reservoir coordination mostly during extreme events. "Some" coordination is represented through the use of mean monthly regulated flow and also the allocation of water demand to a number of reservoirs based on how full they are. This feature is not present in this model representation, and would likely not drastically change extreme events. Yet it does represent "coordination" around releases and other water management performance metrics than flow and storage, rather

coordination on meeting basin-scale water demand. There were statement throughout the paper saying that there was no coordination at all, which seemed then inaccurate and should be clarified.

- evaluation of the model and transfer to other regions: • whether the coordination between reservoirs was represented or not, how does it affect the vulnerability metrics at the scale of the basin, which is what those models were initially developed for? • Most of those models have been developed for application to a wide range of climatology conditions. The model here is applied to a relatively very small basin for its kind. If this manuscript will be used as reference for this large scale reservoir models, it should be either evaluated with respect to other generic rules, or the applicability to larger regions and very different regions should be presented.

3) Evaluation of the contribution of reservoir coordination – artifact of the model? - the main assumption is that the daily releases are based on storage only. All other equivalent models used an estimate of the expected monthly inflow. The main conclusion of the paper is that the coordination between reservoirs should be represented. While I do believe in this conclusion, it seems that the reference to "typical reservoir model" is not justified if the monthly inflow (proxi for foresight without forward running all the models involved) is not represented at all like in other models. My recommendation would be to modify the experiment to evaluate perhaps incremental and simple levels of coordination ( aka adding inflow as parameter for the decision release, or a proxi for inflow) to complement the interpretation of the results and provide more quantifiable statements.

4) Evaluation of the contribution of reservoir coordination during extreme events I found it extremely hard to follow the text and interpretation of the drivers of the release (annual flow versus objective of this reservoir or upstream reservoirs, and shape of release) by just looking at the figures. Most of the text describes the observed operations and coordination and how the model does not capture it. It is unclear how the method of Morris helps with the interpretation during extreme events. While the visualization is

very nice to show the data, it seems that those figures could go in the supplemental material and another figure that compiles those time series and support the text would help.

5) Overall discussion and recommendation versus computational resources needs The authors conclude that foresight should be represented, which is also very sound. Yet the computational resources brought forward for such a relatively small basin are huge which decrease the feasibility at a continental or global scale. Optimization also bring other uncertainties and more computational needs. While the authors seem to indicate that this is what we should do, those were actually drawbacks and motivation for developing those large scale generic models. The recommendation is confusing and perhaps the authors could provide a clarification on new model performances to make it possible now? Please also note that nodal models that typically support reservoir operation optimizations do not provide the spatially distributed feedback into the hydrology model to represent hydrology-land-surface-atmosphere interactions. Maybe the authors meant that we need different types of large scale reservoir models? This would be very sound – just need to be clearer about recommendations then.

---

## Author Comment (AC1) · 12 Feb 2020

**Throughout this response, the reviewer's text is presented in black, our response in blue, and the proposed revisions in green.**

This study deploys the Method of Morris to evaluate the sensitivity of release and storage time series to adjustments in the parameters of generic reservoir operating rules. The analysis is performed using a high resolution hydrological model (WBM) and focuses on a cascade of reservoirs located in the Upper Snake River Basin. The paper is well written, with clearly defined methodology and easy-to-follow results section. The experiments conducted are ill-suited to the aims of the study, leaving too many confounding factors. Specifically, I disagree with the authors key claim that the approach exposes coordination between reservoirs missing from the generic operating schemes (see specific comments below). While I do not doubt that coordination does occur in this particular cascade system, I cannot see how this sensitivity analysis exposes that coordination unequivocally.

We want to thank the reviewer for their thoughtful and constructive comments. We believe they provide a basis for us to clarify and be more explicit with the aims and assumptions of our work, both with textual explanations and supplementary data. The concern that our work does not distinguish when non-coordination is the most likely cause of the simulated drought and flood errors is also well-taken. In a revised version, we will show how coordinated operations would have avoided some (most) of the water extreme related impacts simulated in Sections 4.3 and 4.4, even in a model where (as in any hydrological model) there are other sources of error.

Our detailed response to the key concern expressed by the reviewer (that the errors seen in the paper may not be due to the absence of the coordination in the reservoir release rule) considers multiple aspects in this order:

1) Coordination is not represented in reservoir operation rules as presented in the current literature on large-scale hydrological models. This is not something this paper tries to demonstrate but a fact supported by the literature review made in the introduction.

2) Instead, this paper tries to diagnose some possible consequences of this non representation of coordination. Thus, our study is not focused on identifying the dominant parameters in the rule (for a reservoir in isolation, those can be deduced from the equations). Our Method of Morris analysis is a quantitative diagnostic for better understanding how the influence of the reservoir rules' parameters evolves both through time and through the multi-reservoir cascade. This diagnosis is helpful for addressing two key questions (i) does the assumption that reservoirs can be parameterized separately (made in studies that propose new and more sophisticated release rules) hold? And (ii) what are the potential consequences of this assumption for flood and drought assessments by increasingly detailed hydrological models in a changing world?

3) Whereas question (i) can be addressed by looking at simulation results alone (see Section 4.2), question (ii) is better examined by using a benchmark where we know for a fact that efficient multi-reservoir coordination has historically minimized flood and drought risk: the USRB. We provided evidence of that coordination when comparing historical and simulated storage and release trajectories. This is apparent when narrating the drought

event in Section 4.3: p19 lines 2-4, 8-11 and 15-18, and when narrating the flood event in Section 4.4: p21 lines 25-34 and p23 lines 16-20. We however do agree with the reviewer that we did not provide direct evidence that coordination led to release behaviors that could not be accounted for with improved parameterization of the reservoir release rules. Please refer to point (A) below for a detailed explanation of how we propose to address this in the revision.

4) More specifically, our comparative analyses between historical and simulated release decisions in Sections 4.3 and 4.4 show qualitatively different behavior when contrasting historical data with simulation of the same period. We agree with the reviewer's concern that due to potential other errors in the model, our time-variant diagnosis approach does not in itself rule out that the observed differences may have little to do with the coordination issue. Please refer to point (B) below for a detailed explanation of how we propose to address this in the revision.

5) Please also note that as this is a diagnostic study, our goal is not to fully explain the errors or to correct them. There is to our knowledge no study of a real operational context using a hydrological modeling experiment where one would absolutely know every source of error or potential confounding factor. That being said, we are grateful to the reviewer for remarks that underlined that we need to clarify the presence or absence of confounding factors.

In a revised version, we will:

A) provide more quantitative classification and proof of coordination in the historical record, for both the high flow and low flow event. We will do so by taking the most upstream reservoir, Jackson Lake, and showing that local explanatory variables (storage and incoming monthly inflows) are not enough to explain observed releases. We will then insert and explain the following figures below. For the drought in August 2013 (Section 4.3; left panel) as for the flood in April 2011 (Section 4.4; right panel), the release response (y-axis) to storage + monthly inflows (that's total water availability, x-axis), is unusual. In the 2013 drought, the reservoir has already been emptied to rescue American Falls levels downstream but releases are still the highest in 8 years, whereas in the 2011 flood, releases are around 8 times as high whereas water availability is similar for 6 of the other 7 years. In both cases the release difference with other years cannot be explained by immediately available variables.

[Figure]

This proof of coordination will be inserted in Section 4.1 to add to the comparison of simulated and historical operations.

B) In Sections 4.3 and 4.4, we will demonstrate where observed historical coordination would have been enough to avert the simulated water extremes, notwithstanding other sources of error in the model (which naturally exist). To do this, we'll build a simple offline surrogate of the reservoir cascade and conduct the following 'what-if' experiments where we look at the consequences of storage dynamics based on combining observed releases (mirroring coordination) with simulated inflows (incorporating other sources of error in the model):

(i) for the drought event (Section 4.3) we'll apply the observed releases from Palisades assuming a long (7-day) delay between Palisades releases and American Falls inflows, and demonstrate that American Falls would not have emptied (i.e., lost its capacity to regulate irrigation) in that case. Similarly we'll do the same with Jackson Lake operations and examine how they avoid emptying Palisades in 2013. Note that the 7-day routing delay is a conservative assumption: if higher releases with long delays are enough for the water to reach the downstream reservoir in time, then the shorter actual delays would be enough as well.

(ii) for the flood event (Section 4.4) we'll focus on Jackson Lake first, and show how the observed release behavior would have avoided for the reservoir to lose its regulation capacity by becoming full, even with the simulated inflows. Then we'll translate this different release behavior to Palisades assuming a less than 1-day routing time (the conservative assumption here is that the flood event propagates very quickly) and demonstrate how observed releases would have avoided filling Palisades completely even with simulated inflows. We'll then translate the consequences of this to American Falls.

Please note that for both (i) and (ii) preliminary experiments have been conducted to guarantee that the results will be what we say they are; but we did not have time to produce clean and easy-to-follow figures for this discussion comment.

C) we will add a supplementary Section 3.1 to develop the methodology beyond explaining the rollout of the method of Morris, moving current Sections 3.1 and 3.2 to 3.2 and 3.3 respectively. This section 3.1 will detail:

1) that the analysis will rely on the comparison of coordinated observed operations with non-coordinated simulations (while explaining how we will look for a proof of this coordination)
2) the reasons for using the method of Morris, namely the identification of temporal "signatures", i.e. combinations of variables that are important at the same time, and the study of how these temporal "signatures" may travel through time and through the cascade, or get inverted
3) The supplementary online testing to demonstrate the impact of coordinated operations on avoiding simulated flood and droughts (as described in (B))

D) update the abstract to reflect the above points and clarify the methods and goals of the study.

I also have some concerns with the study design, and in particular the use of the high-resolution hydrological model, which introduces a severe and unnecessary computational constraint while also supplying the reservoirs with (likely) inaccurate historical inflows. The latter leaves the reader unsure as to whether the difference between simulated and observed storage/release is more a function of erroneous inflow than inaccurate operating decisions. This is particularly important when demonstrating the inadequacy of the reservoirs rules in representing flood and drought (it could simply be that the upstream hydrology from the model is delivering the wrong volumes of water). Since the actual observed inflows are available for this system, it would seem far more prudent to develop a simple, offline cascade model. I suspect such a radical change to the experimentation at this stage would be unrealistic, so I would instead encourage the authors to change the aims and storyline offered here. The simulations are sound and there is a great need for the community to learn more about the nature and performance of generic reservoir schemes. I would support a revision if either (a) the authors can convincingly rebut my concerns listed below, or (b) a new angle is developed with a more defensible conclusion.

Our introduction highlights that large scale hydrological models are being employed with increasingly ambitious assessment goals (p3 lines 2-14). These large-scale assessments of hydrological risks are not carried out with surrogates of full-scale models. In this regard, WBM and the reservoir rule system represent a state-of-the-art representative of this class of assessment model that is distinguished in its representation of human infrastructure systems. Our study has a clear and direct design: evaluating model outcomes in the same conditions as large-scale hydrological vulnerability assessments are carried out. This will be explicitly clarified in the methodology.
Moreover, it is non-trivial to connect complex large regional infrastructures to the broader natural components of the water balance. For instance, inflows to other reservoirs need to reflect the consequences of simulated upstream releases (including what that means for flow routing along the river course where there are lateral inflows). There is also the question of which other parts of the model should be included: for instance there are potential consequences for water withdrawals in case of low-flow conditions. This would have

required testing and arbitrarily excluding components of the water balance from the online surrogate to avoid making it overly complex.

This being said, we acknowledge that supplementary experiments involving offline surrogates would complement the main experiment (conducted with a full-scale model) to help clarify the contribution of coordination in averting impacts from water extremes. Therefore, we therefore thanks the reviewer for their suggestion here.

Specific comments to authors:

Section 3.2. The justification for the 10% decision is unconvincing. Suppose a dam has an average inflow of 100 cumecs. An Rmin of 0.1 would be 10 cumecs and would vary by a maximum of 1 cumec. So the left-hand side of the operating curve hardly moves. In contrast, if your Rmax is 5 (=500 cumecs) then you'd vary this parameter by plus or minus 50 cumecs. So the right hand side of the curve will shift wildly in comparison. This is surely why Rmin appears to be unimportant in your sensitivity analysis; you've barely moved it. I appreciate that there are physical reasons why Rmin would be expected to be somewhat less variable than Rmax in absolute terms (although both are highly uncertain), but the uniform 10% assumption is not ideal either, and it invalidates your later statements about which parameters constitute the signature of parametric influence (page 15).

This criticism of the chosen multipliers would be exactly on point if the goal of this sensitivity analysis was to ascertain which parameters in the reservoir rule are dominant. Instead (and as noted above),  this analysis diagnoses how the parameter signature (i.e., the set of parameters that dominate and in which direction they influence outputs) evolves throughout the cascade and with time. The core contribution is not what single parameter controls are dominant, but that the dynamics in sensitivities are complex, highly dynamic, and non-separable. This is the core concern expressed in our Introduction, that these parametric behaviors are in direct contradiction to the current reservoir rule forms representation of reservoirs as being individually separable and independent parameterization problems without any information on coordination..

We acknowledge that the rationale for using the Method of Morris has not been explained with the requisite precision in the previous submission, so it will be explicit in Section 3 in a revised version.

Section 4.2 through 4.4. The results described here show that the operations of upstream dams can affect the decisions made at downstream dams. This occurs because any change upstream has an effect on the inflows into a downstream dam, thus affecting its storage levels and therefore its releases (which are a function of storage). Sensitivity analysis exposes how the decisions taken at one dam are affected by the rules deployed at other dams (often with some intriguing complexities). This is insufficient, however, to claim that the models must be missing *coordination* (which presumably means dam operator A looking at the storage levels of dams B and C to inform his or her decision). Similarly, the results that describe inadequate representation of flood and drought mitigation *could* be caused by uncoordinated reservoir schemes. But inadequate mitigation could also be simply because

the operating schemes at individual dams are insufficient. It's possible that more realistic operating schemes at the individual dams (i.e., not necessarily incorporating coordination, but with more realistic structure and/or parameterization) would provide the correct mitigation responses. It's also possible that the failure of these models to represent flood and drought mitigation is partly caused by bias in your inflow data (which is not shown or compared to observed at the upstream dam). Given these possibilities, the proposed framework fails to demonstrate unequivocally that a missing piece of the reservoir model is coordination.

This comment echoes the reviewer's main concern above and further illustrates it. We thank the reviewer for taking the time to formulate it. It has been helpful in formulating our response to the reviewer's main concern.

Other minor comments:

Figure 1. Difficult to interpret due to color scheme used.

We agree and will amend the figure to improve the color scheme

Table 1. Please provide some additional justification for these parameters. Has there been any validation done?

The general form of the reservoir rule was first presented by Proussevitch et al. (2013) and validated using the GRanD database (Lehner and Liemann, 2011). Variants of this rule have been used with a daily time step on the Niger river basin (Oyerinde et al., 2016), and with large-scale assessments using WBM (Grogan et al, 2015; 2017; Zaveri et al, 2016; Liu et al., 2017). The fine-tuning of the parameters when establishing this version of the rule was made using a set of 22 large North-American and Eurasian reservoirs in offline mode., including the two largest reservoirs in the USRB (Palisades and American Falls, daily release NSE 0.70 and 0.60 respectively). Similar to what happens when a reservoir rule that classifies reservoirs by purpose is used in a large-scale model, we did not fine-tune the rule to each reservoir. This allows us to use the reservoir rule in conditions that are similar to what is done in most state-of-the-art hydrological models.

We will clarify the above in Section 2.4: the reviewer is right to point out that readers should know where the figures in Table 1 come from.

Why would the Rref for a water supply reservoir be 0.1? This parameter should vary widely across water supply reservoirs as a function of demand relative to inflow (and 0.1 is very low).

Thanks for this question that warrants a clarification. In this work, "water supply" means that water is meant to be withdrawn directly from the reservoir, instead of being diverted downstream after release. Therefore, the reservoir must remain as full as possible.

We will provide this clarification in a revised version.

Generally I don't see why the purpose of the reservoir would control these parameters. Rmin (and indeed the whole left-hand section of the operating curve) would likely be strongly determined by environmental flow requirements, which are independent of reservoir purpose. I would also suggest discussing the omission of seasonality in the operational parameters (which could apply to non-irrigation reservoirs). If there is no good justification for these rules then this is ok (it's not the purpose of your experiment to defend the status quo)ãˇAˇ Tin this instance just make a statement to inform the reader.

The idea that reservoir purpose controls release rule parameters is reproduced here because it is ingrained in the literature (see p2 lines 23-35 and p7 lines 11-14). We agree with the reviewer that no parametrization is perfect, be it in our paper or in the wider literature. However, and as the reviewer is right to point out, our goal is not improve existing parameterizations. Instead, it is to diagnose them using a reservoir rule with several state-of-the-art characteristics.

P6 line 1 - reservoir data *were* derived…
P6 line 1 - not clear who performed these manual updates

We will correct the typo and specify that authors made the manual updates

P6 line 7 - How accurate are these demand data compared to USGS estimates, which are US specific.

The GPW is based on US census data, and is processed to a gridded resolution on par with our model grid, whereas USGS estimates are county level. We verified the difference between the two demand estimates and found it to be below 50%, in a basin where industrial and domestic demands are two orders of magnitude smaller than irrigation demands. Therefore, we deemed the GPW data to be acceptable for this study.

Section 2.1. I hadn't studied the Method of Morris previously, but this is an excellent description that educates the reader.
We appreciate this kind comment.

P16 line 7 - The high variability in release (relative to observed) is surely just caused by the right-hand side slope of the operating curve being too steep, or the Sref being too low. Why should this point to incorrect representation of coordination?

We are not sure whose reservoir's Sref or right-hand slope the reviewer is referring to. We are also not sure what they mean by "incorrect": in Section 4.2 we do not assess whether the representation is correct. Instead, as stated clearly at the top of the Section (p 15 lines 32-33), "Our analysis in Figure 6 focuses on flows from a single year to clarify the complex interactions between upstream releases and Minidoka operations"

In other words, in Section 4.2 and Figure 6 we use results from the method of Morris to demonstrate that release rule parameters upstream influence inflows, storage and release

decision at Minidoka. In other words, we show that the assumption that this reservoir can be parameterized separately from the others does not hold here.

This means that the reservoirs should be parameterized jointly. And that would account for coordination. We will insert a sentence in Section 4.2 to make this point explicit.

P16 line 27 - this may have been a low flow event, but your figure for Jackson shows clearly that the drawdown was in large part caused by sustained high release through 2013.

There is no contradiction between the two statements. In fact, as can be shown in the Figure in this document (left panel), there was little water and high releases. This is explained by coordination with downstream reservoirs.

We will insert that figure to clarify this.

It would be helpful to see inflow, storage, and release graphs for the full time series for all reservoirs (perhaps in supplemental material).

We are happy to present this data in supplementary material, using similar graphs as in the paper.

---

## Author Comment (AC2) · 12 Feb 2020

**Throughout this response, the reviewer's text is presented in black, our response in blue, and the proposed revisions in green.**

The manuscript presents a new large scale reservoir operations model with generic operating rules associated with the reservoir main operational purpose such as flood control or irrigation, or default. The reservoir model stands out from equivalent models in that the releases are decided daily based on the daily storage level, shapes with combined log and exponential curves that accelerate the release in times of floods when close to full capacity and slows down the release in times of droughts with differences in the thresholds and propensity to release and store based on the purpose of the reservoir. The overall release is scaled by the long term mean annual flow. The model is implemented at high resolution ( .6 km, daily time step) over the Upper Snake River Basin, which is a snowmelt driven basin. The method of Morris is used to identify the reservoir release parameters that tend to be most influential in the reservoir release and storage variations throughout an 8 year period. Upstream reservoirs are used to evaluate the approach while downstream reservoirs are used to evaluate the impact of upstream reservoirs. A flood and drought events are evaluated with respect to observed operations to categorize the error associated with the lack of representation of reservoir coordination. Authors conclude that reservoir coordination is needed to represent flood and drought in typical reservoir models, and that optimization of rules with foresight would help in this endeavor. All simulations were performed on very high performing computational resources taking 2 days for 8 year simulation over the Upper Snake River Basin.

Thanks for this detailed and accurate account of our work. One (very minor) clarification being that each simulation does not take 2 days (see line 9 on page 13).

The subject is very interesting for the HESS community and the manuscript is well written but there are a number of concerns that would need to be addressed before consideration for publication. The main concerns are about the two (great) highlights of the paper : the new model and the time sensitive analytics; i) the manuscript presents a new large scale model, with a very interesting concept for the releases that is however not enough evaluated and discussed, and ii) the approach to quantify the contribution of reservoir coordination to better represent floods and droughts needs to be improved – it is based on inference statements and the model could be modified to include information about upstream reservoir release to demonstrate the point about coordination.

Thanks for this comment.

(i) Regarding the general concern on the model, we would like to point out that the hydrological model is not new, and neither is the rationale for the release rule. In fact, WBM is a well established representative of the broader class of large scale hydrological assessment models that are being used in regional to global applications, as highlighted in p2 lines 19-22 and p5 lines 18-29. As for the release rule itself, it has also been used with WBM in the past, and its precise provenance will be spelled out in Section 2.4 in a revised version.

What is more, the release rule we use is not put forward for its novelty but for being a state-of-the-art representative of the emerging types of rules that are being employed as reviewed in our Introduction. We also insist on this fact when we detail the rule in Section 2.4 (see p. 11 lines 8-15)

(ii) It is not our intent to precisely quantify the exact contribution of reservoir coordination to overall prediction errors, but rather to use an institutionally complex multi-reservoir cascade that is known to exploit high levels of coordination to illustrate qualitative artifactual behaviors that can emerge from the absence of coordination in how large-scale hydrological models abstract major storages.

This being said, we agree that a new kind of release rule for large-scale hydrological models could take into account coordination. The goal of the paper is not to propose such a rule and demonstrate it, but to diagnose the issue to motivate future efforts in the research community. We will clarify that in a revised version, in the methodology and in abstract or introduction.

Minor feedbacks are that the reference to typical reservoir model is misleading and the analytics with the method of Morris is very hard to follow.

We address the reviewer's concern within their point 1) below. As for the analytics, we appreciate the feedback from the 3 reviewers and will make clarifications to interpretability as discussed in our responses here and for Reviewers 1 and 2.

1) Reference to typical reservoir operations model seems misleading. At the scale of the Upper Snake River Basin, typical reservoir operations models have a nodal architecture and represent accurate reservoir operating rules that can be revisited in optimization mode and especially in forecast mode to mitigate reservoir and drought events. The manuscript here refers to very large scale spatially distributed reservoir models that have been developed initially to be fully coupled with hydrology model and research land-surface-atmosphere interactions. Those models are typically applied over multiple independent large river basins. I would suggest to not refer to typical reservoir model where most of the community understand reservoir models where rules can be optimized and are applied to one basin at a time. Please refer to large scale distributed reservoir model or equivalent differentiation from nodal operational reservoir models.

We agree that had the goal been to propose the most accurate operating rules possible for this basin, we would have chosen very different rule forms. However, the goal of our study is to diagnose the implications for how state-of-the-art large scale hydrological assessment models represent reservoir operation rules. We understand that this points needs to be made even more explicit, and the revision will add a subsection to the methodology that will point this out.

As for the reviewer's concern with the geographical scale of our study, we would like to point out that release rules such as they are encoded in large-scale hydrological models must not lead to large qualitative errors in mid-size basins such as the USRB. Indeed, larger-scale studies would contain a set of basins that are not hydrologically connected with 1) a number

of mid-size basins, and 2) large basins containing several headwater basins such as the USRB. Therefore, insights from a diagnostic approach on the USRB will be relevant to large-scale assessments if we apply the release rule in the same way (i.e., without over-parameterizing it).

For these reasons, the release rule is applied here as it would be within a large-scale hydrological model, that is, without fine-tuning the parameters to each individual reservoir. This will also be made explicit in the revision.

 2) A new large scale reservoir operations model : please provide more details

We would like to clarify that our paper is not focused on a "large scale reservoir operation model". In fact our point is quite the opposite: pointing out the limitations of standard rule-based representations in hydrological models that consider and parameterize reservoirs separately as reviewed in detail in our Introduction.

Revision will explicitly remind the reader that the reservoirs are not coordinated in the model, but only in real-life operations. We will provide evidence of historical operations in the period of record.

 - what is the river routing process for this high spatial resolution and daily time step? A recommendation in the introduction is not to aggregate reservoir storage but many reservoirs have less than 2 days in travel time. How does the reservoir model decision release algorithm adjust stability?

Thanks for this. We reply to each sentence separately and in order:

1) Similar to the above, we would like to point out that the release rule presented here does not consider what happens downstream, where the river routing is done by the hydrological model (WBM).
2) Likewise, we are not trying to issue a recommendation of ours when observing in the introduction that Shin et al (2019) recommend not aggregating reservoir storages. That second paragraph in the introduction sets the context, which is the evolution of hydrological models towards use in ever better-resolved models. Our work looks at what happens with a model that is relatively highly-resolved, but far from hyperresolution. At the resolution we use though, it makes little sense to aggregate storages because they are less than two days apart: this would negate the sought advantages of a higher resolution. As we understand the Shin et al paper, the search for better-represented dynamics is the reason for their recommendation.
3) We apologize but we are not sure what you mean by "adjust stability" here. This said, the reservoir rule mechanisms are completely exposed in Section 2.4, so we invite the reviewer to check whether the answer to their question is positive and negative.

- How are the 6 parameters initialized? Are the necessary data widely available? What are the assumptions?

Thanks for pointing out our lack of explanation of where the parameters of the reservoir rules come from. In a revised version, we will insert in Section 4.2 the following paragraph..

The general form of the reservoir rule was first presented by Proussevitch et al. (2013) and validated using the GRanD database (Lehner and Liemann, 2011). Variants of this rule have been used with a daily time step on the Niger river basin (Oyerinde et al., 2016), and with large-scale assessments using WBM (Grogan et al, 2015; 2017; Zaveri et al, 2016; Liu et al., 2017). The fine-tuning of the parameters when establishing this version of the rule was made using a set of 22 large North-American and Eurasian reservoirs in offline mode., including the two largest reservoirs in the USRB (Palisades and American Falls, daily release NSE 0.70 and 0.60 respectively). Similar to what happens when a reservoir rule that classifies reservoirs by purpose is used in a large-scale model, we did not fine-tune the rule to each reservoir. This allows us to use the reservoir rule in conditions that are similar to what is done in most state-of-the-art hydrological models.

As for the assumptions, we would like to point out that the key one, common to release rules for large-scale hydrological models is that by construction, each reservoir gets separate parameters that do not depend explicitly on the behavior of reservoirs upstream. This will be made explicit in Section 2.4 where the release rule is introduced.

- Evaluation of the smoother release curves with other models. In other equivalent models that are cited (Hanasaki, Doell, Biemans, Voisin, etc), releases are decided daily based on reservoirs minimum and maximum capacities, minimum environmental flow and tend to follow monthly storage and releases targets with no foresight, but using long term mean monthly inflow, which also tends to be regulated or natural flow depending on the models. What is the improvement for those rules? The obvious features are the changes in release rates - how does it improve the flow representation in general?

The reviewer cites four release rules from four papers (Hanasaki et al (2006), Doll et al (2003), Biemans et al (2011), Voisin et al (2013)) that have release rules that decide a monthly release target in order to analyse outputs with a monthly time step. Given this, it makes sense for the rules to be based on monthly parameters.

Yet, as detailed in our introduction (p3 lines 15-35) the transition of applications to include flood concerns means that shorter timescales have to be considered. The smooth monthly curves are not appropriate for finer time scaled extremes (floods) and for large storages those extremes carryover in the effects on water operations for droughts. So in short, there is tension and difficulty in resolving floods and droughts in complex cascades like USRB using standard rule forms and assumptions of independence between reservoirs.

The revision will clarify that the version of the Hansaki et al rule that is used for flooding is modified by Mateo et al (2014) at p3 line 16 to be used with daily time step. That sentence will be amended in that regard.

Please note that we do not try to use a "best" rule but a rule that shares some key characteristics with state-of-the-art available rules (and we would urge the reviewer to refer

to newer representations, e.g. see references in p3, lines 30-35, rather than those they cite here).

- reservoir coordination. Note that the use of a rolling past 20-year of mean monthly regulated inflow provides a minimum of reservoir coordination mostly during extreme events. "Some" coordination is represented through the use of mean monthly regulated flow and also the allocation of water demand to a number of reservoirs based on how full they are. This feature is not present in this model representation, and would likely not drastically change extreme events. Yet it does represent "coordination" around releases and other water management performance metrics than flow and storage, rather coordination on meeting basin-scale water demand. There were statement throughout the paper saying that there was no coordination at all, which seemed then inaccurate and should be clarified.

We thank the reviewer for this comment. It underlines that there are different interpretations of what coordination means. In this paper, we focus on active forms of coordination by human operators, where the dynamics across space and time in release decisions occur in a way that cannot be explained by the immediate or short-term hydrological or climate conditions. A key feature of coordination as understood in this paper is that upstream reservoirs react to and anticipate downstream water issues.

By contrast, the reviewer seems to focus in this comment on two forms of passive coordination, in which reservoirs adjust to varying on-site conditions that are typically imposed on downstream reservoirs by upstream operations. These conditions unfold concurrently to the release decision (inflows influenced by upstream water management, or at-site withdrawals).

We will clarify this difference between our paper's focus (active coordination) and what is common in release rules in large-scale hydrological models (passive coordination) in the introduction (in the review of the literature) and possibly in the abstract.

Side note on inflows as a means of coordination: the reservoir model uses long-term (5 year) mean (regulated) inflow, and the paper illustrates that this may not be sufficient to represent coordination (we will clarify in the revision what "long-term mean" means in Section 2.4 and Figures 3 and 4) .

- evaluation of the model and transfer to other regions: âA ˘ c whether the coordination ´ between reservoirs was represented or not, how does it affect the vulnerability metrics at the scale of the basin, which is what those models were initially developed for?

We thank the reviewer for keeping their eye on the end-goal here. We agree that large-scale hydrological models are increasingly used to assess water vulnerability. Vulnerability metrics are useful when the difference between a situation and the other is quantitative, but this work is striving to highlight qualitative differences: an adverse event vs. its absence.

This being said, we agree that it is important to put the results into context. For instance, we do explain the consequences of losing control of regulating downstream releases at American Falls in a low-flow event (p 16 lines 29-32). We will add a sentence clarifying that

emptying American Falls instantly disrupts irrigation schedules and forces farmers to watch their crops wither.

As for the 2011 flood event, we will add that the peak simulated release at Palisades in Spring 2011 is 50% higher than historical daily release at any point in the last 40 years. The historical maximum corresponds to the 1997 flood, which led to and six counties being declared in a state of disaster and delivered over $11 million in relief by the federal U.S. government (https://www.weather.gov/safety/flood-states-id)

âA ˘ c Most of those models have been developed for application to a wide range of ´ climatology conditions. The model here is applied to a relatively very small basin for its kind. If this manuscript will be used as reference for this large scale reservoir models, it should be either evaluated with respect to other generic rules, or the applicability to larger regions and very different regions should be presented.

Thanks for this comment. Indeed, we are using an example of generic rule that shares key common characteristics with other generic rules; these rules are usually applied to larger hydrological areas.

As explained in a previous comment, it is important to point out what the absence of coordination may lead to if using these rules in headwater basins that would be part of larger-scale studies with large-scale hydrological model.

We will clarify this reason for zooming in to a midsize basin either in the introduction or in the methodology section.

3) Evaluation of the contribution of reservoir coordination – artifact of the model? - the main assumption is that the daily releases are based on storage only. All other equivalent models used an estimate of the expected monthly inflow. The main conclusion of the paper is that the coordination between reservoirs should be represented. While I do believe in this conclusion, it seems that the reference to "typical reservoir model" is not justified if the monthly inflow (proxi for foresight without forward running all the models involved) is not represented at all like in other models. My recommendation would be to modify the experiment to evaluate perhaps incremental and simple levels of coordination ( aka adding inflow as parameter for the decision release, or a proxi for inflow) to complement the interpretation of the results and provide more quantifiable statements.

We do agree that models that schedule release over a monthly time step also use inflows over a monthly time step. Then, release is naturally a function of available water (beginning-of-month storage plus monthly inflow) makes more sense than determining release as a function of storage alone. Likewise, daily release decisions should be a function of beginning-of-day storage and daily inflows. This rationale of taking the same time step for inflows and releases is common to most release rules for large-scale hydrological models, up to the most recent rules (e.g. Yassin et al, 2019, in this journal). We will precise in Section 2.4 that the rule is separately implemented at the hourly time step in the model, to assimilate inflow into outflow calculations, in order to produce a total release at the daily time step in the model.

Beyond this theoretical reasoning, we conducted another check for added confidence that the results are not an artefact of using monthly expected inflows. In the two events examined in this paper, we produced the figure below, which plots historical monthly release (y-axis) as a function of available water (storage + monthly inflows) for all years of the modeled period 2009-2016 at the most upstream reservoir (Jackson Lake). Results show that historical operations could not be replicated simply by incorporating inflows (even monthly inflows) to the release rule.

For the drought in August 2013 (Section 4.3; left panel) as for the flood in April 2011 (Section 4.4; right panel), the release response is unusual. In the 2013 drought, the reservoir has already been emptied to rescue American Falls levels downstream but releases are still the highest in 8 years, whereas in the 2011 flood, releases are around 8 times as high whereas water availability is similar for 6 of the other 7 years.

[Figure]

We will insert the above figure in the revised version, assorted with an explanation that highlights how the anomalous releases (in August 2013 and April 2011 respectively) correspond to coordination that could not be replicated by simply considering expected monthly inflows.

4) Evaluation of the contribution of reservoir coordination during extreme events I found it extremely hard to follow the text and interpretation of the drivers of the release (annual flow versus objective of this reservoir or upstream reservoirs, and shape of release) by just looking at the figures. Most of the text describes the observed operations and coordination and how the model does not capture it. It is unclear how the method of Morris helps with the interpretation during extreme events. While the visualization is very nice to show the data, it seems that those figures could go in the supplemental material and another figure that compiles those time series and support the text would help.

Thanks for this comment that will help us clarify our manuscript. We believe that we were not explicit enough in explaining the rationale for the method of Morris.

In a revised version, we will explain in Section 3 (methods) that we use the method of Morris mainly to identify temporal "signatures", i.e. combinations of variables that are concurrently

dominant. Then we can study the evolution of these temporal "signatures" through time and through the reservoir cascade, to gain insights into how the model operates.

5) Overall discussion and recommendation versus computational resources needs The authors conclude that foresight should be represented, which is also very sound. Yet the computational resources brought forward for such a relatively small basin are huge which decrease the feasibility at a continental or global scale. Optimization also bring other uncertainties and more computational needs. While the authors seem to indicate that this is what we should do, those were actually drawbacks and motivation for developing those large scale generic models. The recommendation is confusing and perhaps the authors could provide a clarification on new model performances to make it possible now? Please also note that nodal models that typically support reservoir operation optimizations do not provide the spatially distributed feedback into the hydrology model to represent hydrology-land-surface-atmosphere interactions. Maybe the authors meant that we need different types of large scale reservoir models? This would be very sound – just need to be clearer about recommendations then.

We fully agree with the reviewer that there is a research challenge ahead in terms of incorporating human complexity into large-scale hydrological models. We recognize that a first step in addressing this challenge is to signal the consequences of ignoring it, and that is what our paper is trying to do. Clearly, it does not try to provide an easy fix, and while the discussion outlines some ways forward, we do not pretend solutions are going to be easy to design and implement.

---

## Author Comment (AC3) · 12 Feb 2020

**Throughout this response, the reviewer's text is presented in black, our response in blue, and the proposed revisions in green.**

This article provides an evaluation of the consequences of the lack of a representation of reservoir coordination within a multi-reservoir system when simulating flood and drought events in large-scale hydrological models. The model Water Balance Model simulates a multi-reservoir system in USA. The model includes the representation of each reservoir operation policy (using predefined parameters according to each reservoir purpose) but it does not represent the coordination between reservoirs. The global sensitivity analysis Method of Morris is used to assess the effect of the parameterization to the model outputs. Authors conclude that the representation of reservoir policies independently is not enough and that, in addition, we need to capture reservoir coordination in large-scale models to properly simulate flood and drought effects.

We thank the reviewer for their clear understanding of our paper, and more broadly for their thoughtful and comprehensive review.

The article is well written and structured. The Introduction makes a good review of the hydrological impacts of multi-reservoir systems and previous attempts in representing reservoir systems in hydrological models.

We thank the reviewer for their kind words.

The methodology is well defined with the exception a few aspects that need further explanation. The article does not explicitly say where the parameters of the reservoir rules (Table 1) come from. Moreover, the authors do not specify the parameter ranges. If the values in Table 1 were obtained by calibration in a previous work, the authors could show the ranges applied in that calibration or just reference that work. If there is no previous calibration, how the predefined values of the parameters produce a good agreement between observed and simulated storages and releases (e.g. Figure 5) in normal climate conditions?

Thanks for pointing out our lack of explanation of where the parameters of the reservoir rules come from. In a revised version, we will insert in Section 4.2 the following paragraph..

The general form of the reservoir rule was first presented by Proussevitch et al. (2013) and validated using the GRanD database (Lehner and Liemann, 2011). Variants of this rule have been used with a daily time step on the Niger river basin (Oyerinde et al., 2016), and with large-scale assessments using WBM (Grogan et al, 2015; 2017; Zaveri et al, 2016; Liu et al., 2017). The fine-tuning of the parameters when establishing this version of the rule was made using a set of 22 large North-American and Eurasian reservoirs in offline mode., including the two largest reservoirs in the USRB (Palisades and American Falls, daily release NSE 0.70 and 0.60 respectively). Similar to what happens when a reservoir rule that classifies reservoirs by purpose is used in a large-scale model, we did not fine-tune the rule to each reservoir. This allows us to use the reservoir rule in conditions that are similar to what is done in most state-of-the-art hydrological models.

Concerning the obtention of parameter ranges from calibration: we commend the reviewer for their rigor but would like to point out that in the literature on reservoir release rules for large-scale hydrological models (discussed in depth in the Introduction), parameter ranges are not given and instead, single values are suggested. In our diagnosis of these rules, we choose to follow a similar methodology.

The results and discussion are also clear and well structured but there is a lack of discussion of how the methodology applied here can be used by others. I was wondering if this could be done using a different model where the parameters of the reservoir rules are unknown and need to be obtained by calibration.

The goal of this paper is not to propose a diagnosis framework on what constitutes a "good enough" representation of reservoir operations, but to shed light on what unintended consequences of not representing coordination can be. It should be read as a diagnosis of the kind of effects that can emerge with existing representations, if used within large-scale hydrological for flood and drought assessments.

The revised version will insert a subsection in the methodology that clarifies our rationale in the design of the diagnostic analysis, and how the Method of Morris is only part of the analysis. This will also clarify that we are not trying to implement an approach to be used (although we welcome others to carry their own diagnostics) but rather, we want to make a point about reservoir representations in hydrological models.

Lastly, I think that the paper needs further and clearer discussion on why the lack of representation of reservoir coordination is most likely to be the main reason of this failure to simulate flood and drought events.

We agree with this comment. We have written a detailed answer to the same comment by Reviewer's #1 and rather than offering a boiled-down version, we believe it is better to refer to it.

In conclusion, this paper makes a relevant contribution to the growing discussion around the representation of reservoir systems in hydrological models and it has clear practical implications. The authors provide practical recommendations and possible solutions. While the representation of reservoir coordination is still very difficult to implement in models, this study highlights its importance and the need to, at least, consider this limitation when modelling catchments containing reservoir systems under extreme conditions.

We would like to thank the reviewer for their kind words and accurate assessment.

Other comments:

- While the authors provide a justification for the 10% range used for the sensitivity analysis, in my opinion, it would be interesting to also show what range of variation around base values should be applied to properly simulate any of the drought and flood events.

As we discuss above, the intent of this study is not to correct the errors observed but rather, to diagnose them as an artifact of reservoir rule representations using a state-of-the-art

assessment model example in an institutionally complex reservoir cascade. For this diagnosis, the 10% range is adequate in showing how the parametric effects across the reservoir cascade in the USRB are highly complex, interdependent, and dynamic.

- Page 15, Lines 17-19: What if the releases were represented as cumulative releases, the sensitivity would be as consistent as for the storage?

This is correct. However, if we used cumulative releases then at Jackson Lake we would have the exact same and opposite sensitivity indexes compared to storage. This information is captured in the current results.

- Page 21, Lines 2-4: Could you please provide with a possible explanation to this unexpected result?

Lines 5-12 provide this explanation. We will make the link with lines 2-4 explicit in the revision, so it is clear to readers (including reviewers).

---

## Author Response (AR1)

**Response to reviewers**

We thank the editor and three reviewers for the high-quality, constructive comments. They have really helped to improve the paper.

We addressed all reviewers' comments as completely as we could. As a result, most sections have been amended. The main changes to the paper are listed as follows:

➔ A new Section 3.1 explains in detail the methodology for the analysis, and why we used the Method of Morris.
➔ In the results, a new paragraph (Section 4.1.2 and Figure 6) justifies why we argue coordination between reservoirs is observable in historical operations, both to avoid flooding during the Spring 2011 and to avoid drought in 2012-13.
➔ Offline water balance models have been setup for both the drought and flood events where we are comparing simulation results to historical operations. They show that the coordination measures observed in the historical record would have been sufficient to avoid flood and drought in the simulation, if they had been implemented. These models are discussed in Sections 4.3 and 4.4 (and a new Figure 11)
➔ In Section 4.4, we decided to focus exclusively on the two reservoirs in the basin with a flood control role: Palisades and American Falls.

In what follows, page and line numbers given in "Response" parts refer to the original submission, whereas page and line numbers given in "Manuscript changes" parts refer to the revised submission.

**Reviewer 1:**

Comment 1: This study deploys the Method of Morris to evaluate the sensitivity of release and storage time series to adjustments in the parameters of generic reservoir operating rules. The analysis is performed using a high resolution hydrological model (WBM) and focuses on a cascade of reservoirs located in the Upper Snake River Basin. The paper is well written, with clearly defined methodology and easy-to-follow results section. The experiments conducted are ill-suited to the aims of the study, leaving too many confounding factors. Specifically, I disagree with the authors key claim that the approach exposes coordination between reservoirs missing from the generic operating schemes (see specific comments below). While I do not doubt that coordination does occur in this particular cascade system, I cannot see how this sensitivity analysis exposes that coordination unequivocally.

Response 1: We want to thank the reviewer for their thoughtful and constructive comments. We believe they provide a basis for us to clarify and be more explicit with the aims and assumptions of our work, both with textual explanations and supplementary data. The

concern that our work does not distinguish when non-coordination is the most likely cause of the simulated drought and flood errors is also well-taken. In a revised version, we will show how coordinated operations would have avoided some (most) of the water extreme related impacts simulated in Sections 4.3 and 4.4, even in a model where (as in any hydrological model) there are other sources of error.

Our detailed response to the key concern expressed by the reviewer (that the errors seen in the paper may not be due to the absence of the coordination in the reservoir release rule) considers multiple aspects in this order:

1) Coordination is not represented in reservoir operation rules as presented in the current literature on large-scale hydrological models. This is not something this paper tries to demonstrate but a fact supported by the literature review made in the introduction.

2) Instead, this paper tries to diagnose some possible consequences of this non representation of coordination. Thus, our study is not focused on identifying the dominant parameters in the rule (for a reservoir in isolation, those can be deduced from the equations). Our Method of Morris analysis is a quantitative diagnostic for better understanding how the influence of the reservoir rules' parameters evolves both through time and through the multi-reservoir cascade. This diagnosis is helpful for addressing two key questions (i) does the assumption that reservoirs can be parameterized separately (made in studies that propose new and more sophisticated release rules) hold? And (ii) what are the potential consequences of this assumption for flood and drought assessments by increasingly detailed hydrological models in a changing world?

3) Whereas question (i) can be addressed by looking at simulation results alone (see Section 4.2), question (ii) is better examined by using a benchmark where we know for a fact that efficient multi-reservoir coordination has historically minimized flood and drought risk: the USRB. We provided evidence of that coordination when comparing historical and simulated storage and release trajectories. This is apparent when narrating the drought event in Section 4.3: p19 lines 2-4, 8-11 and 15-18, and when narrating the flood event in Section 4.4: p21 lines 25-34 and p23 lines 16-20. We however do agree with the reviewer that we did not provide direct evidence that coordination led to release behaviors that could not be accounted for with improved parameterization of the reservoir release rules. Please refer to point (A) below for a detailed explanation of how we propose to address this in the revision.

4) More specifically, our comparative analyses between historical and simulated release decisions in Sections 4.3 and 4.4 show qualitatively different behavior when contrasting historical data with simulation of the same period. We agree with the reviewer's concern that due to potential other errors in the model, our time-variant diagnosis approach does not in itself rule out that the observed differences may have little to do with the coordination issue. Please refer to point (B) below for a detailed explanation of how we propose to address this in the revision.

5) Please also note that as this is a diagnostic study, our goal is not to fully explain the errors or to correct them. There is to our knowledge no study of a real operational context using a

hydrological modeling experiment where one would absolutely know every source of error or potential confounding factor. That being said, we are grateful to the reviewer for remarks that underlined that we need to clarify the presence or absence of confounding factors.

Manuscript changes 1:

A) We have provided a more quantitative classification and proof of coordination in the historical record, for both the high flow and low flow event. In support of this, we now include the figure below as Figure 6 and added a paragraph in Section 4.1 (Page 17 lines 5-18) to discuss it. We included this Figure because both panels show unusual historical release patterns concurrent with both the flood and drought events. For the drought in August 2013 (Section 4.3; left panel) as for the flood in April 2011 (Section 4.4; right panel), the release response (y-axis) to storage + monthly inflows (that's total water availability, x-axis), is unusual. In the 2013 drought, the reservoir has already been emptied to rescue American Falls levels downstream but releases are still the highest in 8 years, whereas in the 2011 flood, releases are around 8 times as high whereas water availability is similar for 6 of the other 7 years. In both cases the release difference with other years cannot be explained by immediately available variables.

[Figure]

[Figure]

As noted in our methodology (page 12 lines 7-11),  we aim to "find qualitative behaviors in historical (i.e., observed) operations that cannot be accounted for by a reservoir's release rule simply by changing parameter values or integrating a near-term (less than a month) inflow forecast such as in some existing rules (e.g., Biemans et al, 2011). Other variables would be necessary to incorporate these behaviors within the hydrological models. They will be interpreted as preliminary indications of cooperation, and the events they highlight will be investigated further."

B) In Sections 4.3 and 4.4, we built simple offline water balance model of the reservoir cascade to investigate where observed historical coordination would have been sufficient to avert the simulated water extremes, notwithstanding other sources of error in the model (which will always exist). These simple offline water balance models of the reservoir

cascade, detailed below for (i) the drought event, and (ii) the flood event, were used to conduct 'what-if' experiments where we look at the consequences of storage dynamics based on combining simulated inflows and storages (incorporating other sources of error in the model) with a releases that are modified from the simulated ensemble to introduce coordination. In detail, we have for each experiment conducted with an offline water balance model:

(i) for the drought event (Section 4.3, page 21 line 30 to p 22 line 3):

"To test whether the observed differences between the historical record and simulation are an artefact of other errors in the hydrological model, we built a simple offine water balance model over 2012-2013, taking the three reservoirs' simulated water balance offline. For each of the 400 ensemble members, we then simulated operations obtained by releasing 8 million m3 more each day in July and August than planned by release rules -- a total extra release of 496 million m3 [matching the difference between historical and simulated trajectories observed in Figure 8]. The extra release concerns only Palisades in 2012, and both Palisades and Jackson Lake in 2013. To make sure that underestimating routing times between reservoirs does not falsely cause the reservoirs to store water, we choose a conservatively high (for the area) routing time of 7 days between each reservoir and the one downstream. Results (Supplementary Information Figures 1 and 2) show that with this simple coordination water balance measure, American Falls would not have emptied -- and reservoirs upstream of it would not have lost their capacity to supply downstream agriculture either."

Note that the 7-day routing delay is a conservative assumption: if higher releases with long delays are enough for the water to reach the downstream reservoir in time, then the shorter actual delays would be enough as well.

(ii) for the flood event (Section 4.4) we focus exclusively on the two reservoirs that provide flood control in the USRB: Jackson Lake and Palisades (and we took American Falls out of the analysis). This improved focus enabled us to better describe the coordination between these two reservoirs, and to highlight the real-world consequences of peak releases as high as those simulated out of Palisades (page 25 lines 18-21). We also describe an additional offline reservoir water balance experiment (Section 4.4.3 starting p25, line 26), that quantitatively verifies that a simple early release / two-reservoir coordination policy avoids catastrophic flooding within this version of the model.

C) we have added additional text in a new Section 3.1 (p 10 line 28 to p 12 line 27) to clarify the changes in our methodology and go beyond solely explaining the method of Morris, moving current Sections 3.1 and 3.2 to 3.2 and 3.3 respectively. In particular, section 3.1 now details:

1. that the analysis is seeks to "diagnose the potential consequences of non-coordination within large-scale models' reservoir release rule, our dual objectives are to (A)     understand how these consequences play out through space and time in     a large-scale hydrological model and (B) compare and contrast the implications of the reservoir rule representation for capturing real-world coordinated

reservoir operations", (page 10 lines 29-32).

2. the method of Morris is used a evaluative model diagnostic tool, where "the focus is not solely on which parameters in the release rule are most influential in regulating flows, but importantly clarifying how the set of dominant controls on water flows and reservoir storage levels evolves along a complex multi-reservoir cascade through time. Parametric sensitivities are then used alongside storage and release trajectories for the simulated ensemble to assess how sets of dominant controls in a point in time and at a given reservoir can be associated to high- or low-flow conditions." (page 12 lines 1-5)

3. that "the comparison will be supplemented by an analysis exploiting offline reservoir water balance models that implement basic coordination on the simulated ensembles at times where there are indications of cooperation. These offline reservoir water balance models help to show that coordination alone could explain the qualitative differences between simulated and historical operations. Note that we cannot fully quantify and explain all sources of errors: there is to our knowledge no study of a real operational context using a hydrological modeling experiment where one would absolutely know every source of error or potential confounding factor." (page 12 lines 11-19)

D) update the abstract to reflect the above points and clarify the methods and goals of the study. We inserted (page 1 lines 9-12):

"We employ a time-varying sensitivity analysis that utilizes Method of Morris factor screening to track how the release rule parameters that control reservoir storage and release evolve 1) along the cascade, and 2) in time according to seasonal high- and low-flow events. We combine this with a comparative analysis of historical operation and targeted experiments with simple offline reservoir water balance models."

Comment 2: I also have some concerns with the study design, and in particular the use of the high-resolution hydrological model, which introduces a severe and unnecessary computational constraint while also supplying the reservoirs with (likely) inaccurate historical inflows. The latter leaves the reader unsure as to whether the difference between simulated and observed storage/release is more a function of erroneous inflow than inaccurate operating decisions. This is particularly important when demonstrating the inadequacy of the reservoirs rules in representing flood and drought (it could simply be that the upstream hydrology from the model is delivering the wrong volumes of water). Since the actual observed inflows are available for this system, it would seem far more prudent to develop a simple, offline cascade model. I suspect such a radical change to the experimentation at this stage would be unrealistic, so I would instead encourage the authors to change the aims and storyline offered here. The simulations are sound and there is a great need for the community to learn more about the nature and performance of generic reservoir schemes. I

would support a revision if either (a) the authors can convincingly rebut my concerns listed below, or (b) a new angle is developed with a more defensible conclusion.

Response 2: Our introduction highlights that large scale hydrological models are being employed with increasingly ambitious assessment goals (p3 lines 2-14). These large-scale assessments of hydrological risks are not carried out with surrogates (offline simpler versions) of full-scale models. In this regard, WBM and the reservoir rule system represent a state-of-the-art representative of this class of assessment model that is distinguished in its representation of human infrastructure systems. Our study has a clear and direct design: evaluating model outcomes in the same conditions as large-scale hydrological vulnerability assessments are carried out.

Moreover, it is non-trivial to connect complex large regional infrastructures to the broader natural components of the water balance. For instance, inflows to other reservoirs need to reflect the consequences of simulated upstream releases (including what that means for flow routing along the river course where there are lateral inflows). There is also the question of which other parts of the model should be included: for instance there are potential consequences for water withdrawals in case of low-flow conditions. This would have required testing and arbitrarily excluding components of the water balance from the online surrogate to avoid making it overly complex.

This being said, we acknowledge that supplementary experiments involving simple offline reservoir water balance models would complement the main experiment (conducted with a full-scale model) to help clarify the contribution of coordination in averting impacts from water extremes. Therefore, we therefore thank the reviewer for their suggestion here.

Manuscript changes 2: We explicitly clarify that our study aims "to evaluate model outcomes in the same conditions as large-scale hydrological vulnerability assessments are carried out" (page 12 lines 24-25).

We also used complementary experiments using offline simple reservoir water balance models (see Manuscript changes 1).

Comment 3: Section 3.2. The justification for the 10% decision is unconvincing. Suppose a dam has an average inflow of 100 cumecs. An Rmin of 0.1 would be 10 cumecs and would vary by a maximum of 1 cumec. So the left-hand side of the operating curve hardly moves. In contrast, if your Rmax is 5 (=500 cumecs) then you'd vary this parameter by plus or minus 50 cumecs. So the right hand side of the curve will shift wildly in comparison. This is surely why Rmin appears to be unimportant in your sensitivity analysis; you've barely moved it. I appreciate that there are physical reasons why Rmin would be expected to be somewhat less variable than Rmax in absolute terms (although both are highly uncertain), but the uniform 10% assumption is not ideal either, and it invalidates your later statements about which parameters constitute the signature of parametric influence (page 15).

Response 3: This criticism of the chosen multipliers would be exactly on point if the goal of this sensitivity analysis was to ascertain which parameters in the reservoir rule are dominant. Instead (and as noted above), this analysis diagnoses how the parameter signature (i.e., the set of parameters that dominate and in which direction they influence outputs) evolves throughout the cascade and with time. The core contribution is not what single parameter controls are dominant, but that the dynamics in sensitivities are complex, highly dynamic, and non-separable. This is the core concern expressed in our Introduction, that these parametric behaviors are in direct contradiction to the current reservoir rule forms representation of reservoirs as being individually separable and independent parameterization problems without any information on coordination..

Manuscript changes 3: We acknowledge that the rationale for using the Method of Morris has not been explained with the requisite precision in the previous submission, so it is now explicit in Section 3, page 12 lines 1-5 in particular (see also the Manuscript changes #1 in this document).

Comment 4: Section 4.2 through 4.4. The results described here show that the operations of upstream dams can affect the decisions made at downstream dams. This occurs because any change upstream has an effect on the inflows into a downstream dam, thus affecting its storage levels and therefore its releases (which are a function of storage). Sensitivity analysis exposes how the decisions taken at one dam are affected by the rules deployed at other dams (often with some intriguing complexities). This is insufficient, however, to claim that the models must be missing *coordination* (which presumably means dam operator A looking at the storage levels of dams B and C to inform his or her decision). Similarly, the results that describe inadequate representation of flood and drought mitigation *could* be caused by uncoordinated reservoir schemes. But inadequate mitigation could also be simply because the operating schemes at individual dams are insufficient. It's possible that more realistic operating schemes at the individual dams (i.e., not necessarily incorporating coordination, but with more realistic structure and/or parameterization) would provide the correct mitigation responses. It's also possible that the failure of these models to represent flood and drought mitigation is partly caused by bias in your inflow data (which is not shown or compared to observed at the upstream dam). Given these possibilities, the proposed framework fails to demonstrate unequivocally that a missing piece of the reservoir model is coordination.

Response 4: This comment echoes the reviewer's main concern above and further illustrates it. We thank the reviewer for taking the time to formulate it. It has been helpful in formulating our response to the reviewer's main concern (please see Response 1 and Changes to Manuscript 1).

Comment 5: Figure 1. Difficult to interpret due to color scheme used.

Response 5: We agree.

Manuscript changes 5: We amended the figure to improve the color scheme, and clarified the legend.

Comment 6: Table 1. Please provide some additional justification for these parameters. Has there been any validation done?

Response 6: the reviewer is right to point out that readers should know where the figures in Table 1 come from.

Manuscript changes 6: We added this in Section 2.4 (page 6 lines 2-14):

The general form of the reservoir rule was first presented by Proussevitch et al. (2013) and validated using the GRanD database (Lehner et al., 2011). Variants of this rule have been used with a daily time step on the Niger river basin (Oyerinde et al., 2016), and with large-scale assessments using WBM (Grogan et al, 2015; 2017; Zaveri et al, 2016; Liu et al., 2017). The fine-tuning of the parameters when establishing this version of the rule was made using a set of 22 large North-American and Eurasian reservoirs in offline mode, including the two largest reservoirs in the USRB (Palisades and American Falls, daily release NSE 0.70 and 0.60 respectively). Similar to what happens when a reservoir rule that classifies reservoirs by purpose is used in a large-scale model, we did not fine-tune the rule to each reservoir. This allows us to use the reservoir rule in conditions that are similar to what is done in most state-of-the-art hydrological models.

Comment 7: Why would the Rref for a water supply reservoir be 0.1? This parameter should vary widely across water supply reservoirs as a function of demand relative to inflow (and 0.1 is very low).

Response 7: Thanks for this question. According to the convention followed in WBM's release rule, water is withdrawn directly from water supply reservoirs, therefore the goal is to keep water in the reservoir unless it is close to being completely full.

Manuscript changes 7: We amended the sentence on the "water supply" use (page 10 lines 12-14)

"The primary function for the ``Water supply'' use is to keep releases minimal and storage maximal in order to maximize the quantity of water that can be drawn directly from the reservoir -- except for circumstances that require flood control at near-full storage."

Comment 8: Generally I don't see why the purpose of the reservoir would control these parameters. Rmin (and indeed the whole left-hand section of the operating curve) would likely be strongly determined by environmental flow requirements, which are independent of reservoir purpose. I would also suggest discussing the omission of seasonality in the

operational parameters (which could apply to non-irrigation reservoirs). If there is no good justification for these rules then this is ok (it's not the purpose of your experiment to defend the status quo)ă˘Ă˘ Tin this instance just make a statement to inform the reader.

Response 8: The idea that reservoir purpose controls release rule parameters is reproduced here because it is ingrained in the literature (see p2 lines 23-35 and p7 lines 11-14). We agree with the reviewer that no parametrization is perfect, be it in our paper or in the wider literature. However, and as the reviewer is right to point out, our goal is not to improve existing parameterizations. Instead, it is to diagnose them using a reservoir rule with several state-of-the-art characteristics.

Comment 9:

P6 line 1 - reservoir data *were* derived…

P6 line 1 - not clear who performed these manual updates

Manuscript changes 9: We corrected the typo and specified that authors made the manual updates (using the "we" page 6 line 5)

Comment 10: P6 line 7 - How accurate are these demand data compared to USGS estimates, which are US specific.

Response 10: The GPW is based on US census data, and is processed to a gridded resolution on par with our model grid, whereas USGS estimates are county level. We verified the difference between the two demand estimates and found it to be below 50%, in a basin where industrial and domestic demands are two orders of magnitude smaller than irrigation demands. Therefore, we deemed the GPW data to be acceptable for this study.

Comment 11: Section 2.1. I hadn't studied the Method of Morris previously, but this is an excellent description that educates the reader.

Response 11: We appreciate this kind comment.

Comment 12: P16 line 7 - The high variability in release (relative to observed) is surely just caused by the right-hand side slope of the operating curve being too steep, or the Sref being too low. Why should this point to incorrect representation of coordination?

Response 12: We are not sure whose reservoir's Sref or right-hand slope the reviewer is referring to. We are also not sure what they mean by "incorrect": in Section 4.2 we do not

assess whether the representation is correct. Instead, as stated clearly at the top of the Section (p 15 lines 32-33), "Our analysis in Figure 6 focuses on flows from a single year to clarify the complex interactions between upstream releases and Minidoka operations"

In other words, in Section 4.2 and Figure 6 we use results from the method of Morris to demonstrate that release rule parameters upstream influence inflows, storage and release decision at Minidoka. In other words, we show that the assumption that this reservoir can be parameterized separately from the others does not hold here.

This means that the reservoirs should be parameterized jointly. And that would account for coordination.

Manuscript changes 12: We amended the last sentence of Section 4.2 to read: "Joint parameterization would explicitly account for coordination within reservoir operations, but would also require searching a parameter space of very high dimensionality."

Comment 13: P16 line 27 - this may have been a low flow event, but your figure for Jackson shows clearly that the drawdown was in large part caused by sustained high release through 2013.

Response 13: There is no contradiction between the two statements. In fact, as can be shown in the Figure in this document (left panel), there was little water and high releases. This is explained by coordination with downstream reservoirs.

Manuscript changes 13: We inserted that figure (as Figure 6 page 17) to clarify this.

Comment 14: It would be helpful to see inflow, storage, and release graphs for the full time series for all reservoirs (perhaps in supplemental material).

Manuscript changes 14: We presented this data in supplementary material, using similar graphs as in the paper.

**Reviewer 2:**

Comment 1: This article provides an evaluation of the consequences of the lack of a representation of reservoir coordination within a multi-reservoir system when simulating flood and drought events in large-scale hydrological models. The model Water Balance Model simulates a multi-reservoir system in USA. The model includes the representation of each reservoir operation policy (using predefined parameters according to each reservoir

purpose) but it does not represent the coordination between reservoirs. The global sensitivity analysis Method of Morris is used to assess the effect of the parameterization to the model outputs. Authors conclude that the representation of reservoir policies independently is not enough and that, in addition, we need to capture reservoir coordination in large-scale models to properly simulate flood and drought effects.

Response 1: We thank the reviewer for their clear understanding of our paper, and more broadly for their thoughtful and comprehensive review.

Comment 2: The article is well written and structured. The Introduction makes a good review of the hydrological impacts of multi-reservoir systems and previous attempts in representing reservoir systems in hydrological models.

Response 2: We thank the reviewer for their kind words.

Comment 3: The methodology is well defined with the exception a few aspects that need further explanation. The article does not explicitly say where the parameters of the reservoir rules (Table 1) come from. Moreover, the authors do not specify the parameter ranges. If the values in Table 1 were obtained by calibration in a previous work, the authors could show the ranges applied in that calibration or just reference that work. If there is no previous calibration, how the predefined values of the parameters produce a good agreement between observed and simulated storages and releases (e.g. Figure 5) in normal climate conditions?

Response 3: Thanks for pointing out the need to clarify how the parameters were specified for the reservoir rules .

As for parameter ranges for parameter sampling, the methodology does specify and explain the rationale for this (p 12 lines 17-20), even though we acknowledge precisions should be given. Please note that parameters ranges given in the methodology are not informed from release rule calibration but by the necessity to design a diagnostic experiment whose results are easy to interpret. A major reason for this is that calibration ranges are usually not considered in studies using already-parameterized release rules.

Manuscript changes 3: we inserted in Section 2.4 a paragraph explaining the origin of the rule and of its current parameterizations, as well as the associated assumptions  (p 8 lines 2-14; the paragraph can also be in response to Reviewer #1, under "Manuscript changes 6").

We also explained why we import previously-calibrated parameter values and apply a sampling range that has more to do with the design of experiment than with parameter calibration. On one hand, (page 12 lines 24-26) we:

 "aim to evaluate model outcomes in the same conditions as large-scale hydrological vulnerability assessments are carried out. For instance, we do not fine-tune reservoir rule

parameters to individual basins, and instead use purpose-based parameterizations typical in large-scale studies (e.g., Biemans et al.,  2011; Voisin et al., 2013a; Yoshikawa et al., 2014)."

On the other hand (page 13 lines 25-30):

"This analysis uses a range of plus / minus 10% around base values for all parameters in Table 1. These modest 10% ranges would be conservative if our focus were calibration and not diagnosis. Results (Section 4) will demonstrate that our narrow sampling yields quite substantial effects when compounded across the reservoir cascade in periods where coordinated operations are significant. Besides, there are two reasons for choosing the same range across all parameters: 1) it accounts for the fact that each parameter does not have the same base value across all reservoirs, and 2) it facilitates comparisons between different parameters' sensitivity indices."

Comment 4: The results and discussion are also clear and well structured but there is a lack of discussion of how the methodology applied here can be used by others. I was wondering if this could be done using a different model where the parameters of the reservoir rules are unknown and need to be obtained by calibration.

Response 4: The goal of this paper is not to propose a calibration or parametric model identification  framework on what constitutes a "good enough" representation of reservoir operations, but to shed light on what the unintended consequences of not representing coordination can be. This study contributes a diagnostic evaluation of the effects that can emerge with existing reservoir rule representations, if used within large-scale hydrological for flood and drought assessments.

Manuscript changes 4: The revised version has a subsection (3.1, start page 10 lines 28) in the methodology that clarifies our rationale in the design of the diagnostic analysis, and how the Method of Morris is only part of the analysis. It also clarifies that we are using the example of the USRB to draw "lessons can then be applied to understand the potential consequences of not considering coordination between reservoirs over larger study areas" (page 12 lines 22-23). This study is not intended to focus on more general aspects of parameter calibration or model identification..

Comment 5: Lastly, I think that the paper needs further and clearer discussion on why the lack of representation of reservoir coordination is most likely to be the main reason of this failure to simulate flood and drought events.

Response 5: We agree with this comment. We have written a detailed answer to the same comment by Reviewer 1 (please refer toResponse 1 to Reviewer 1)

Comment 6: In conclusion, this paper makes a relevant contribution to the growing discussion around the representation of reservoir systems in hydrological models and it has clear practical implications. The authors provide practical recommendations and possible solutions. While the representation of reservoir coordination is still very difficult to implement in models, this study highlights its importance and the need to, at least, consider this limitation when modelling catchments containing reservoir systems under extreme conditions.

Response 6: We would like to thank the reviewer for their kind words and accurate assessment.

Comment 7: While the authors provide a justification for the 10% range used for the sensitivity analysis, in my opinion, it would be interesting to also show what range of variation around base values should be applied to properly simulate any of the drought and flood events.

Response 7: As we discuss above, the intent of this study is not to correct the errors observed but rather, to diagnose them as an artifact of reservoir rule representations using a state-of-the-art assessment model example in an institutionally complex reservoir cascade. For this diagnosis, the 10% range is adequate in showing how the parametric effects across the reservoir cascade in the USRB are highly complex, interdependent, and dynamic.

Manuscript changes 7: where the ranges are introduced in the manuscript, we added explanations (page 13 lines 25-30) which we also quote at the end of "Manuscript changes 3" of this document.

Comment 8: Page 15, Lines 17-19: What if the releases were represented as cumulative releases, the sensitivity would be as consistent as for the storage?

Response 8: This is correct. However, if we used cumulative releases then at Jackson Lake we would have the exact same and opposite sensitivity indexes compared to storage. This information is captured in the current results.

Comment 9: Page 21, Lines 2-4: Could you please provide with a possible explanation to this unexpected result?

Response 9: Lines 5-12 provide this explanation.

Manuscript changes 9: We replaced "Instead" by the much more appropriate "This is because" at p 22 line 22 (formerly p 21 line 4)

**Reviewer 3:**

Comment 1: The manuscript presents a new large scale reservoir operations model with generic operating rules associated with the reservoir main operational purpose such as flood control or irrigation, or default. The reservoir model stands out from equivalent models in that the releases are decided daily based on the daily storage level, shapes with combined log and exponential curves that accelerate the release in times of floods when close to full capacity and slows down the release in times of droughts with differences in the thresholds and propensity to release and store based on the purpose of the reservoir. The overall release is scaled by the long term mean annual flow. The model is implemented at high resolution ( .6 km, daily time step) over the Upper Snake River Basin, which is a snowmelt driven basin. The method of Morris is used to identify the reservoir release parameters that tend to be most influential in the reservoir release and storage variations throughout an 8 year period. Upstream reservoirs are used to evaluate the approach while downstream reservoirs are used to evaluate the impact of upstream reservoirs. A flood and drought events are evaluated with respect to observed operations to categorize the error associated with the lack of representation of reservoir coordination. Authors conclude that reservoir coordination is needed to represent flood and drought in typical reservoir models, and that optimization of rules with foresight would help in this endeavor. All simulations were performed on very high performing computational resources taking 2 days for 8 year simulation over the Upper Snake River Basin.

Response 1: Thanks for this detailed and accurate account of our work. One (very minor) clarification being that each simulation does not take 2 days (see line 9 on page 13).

Comment 2: The subject is very interesting for the HESS community and the manuscript is well written but there are a number of concerns that would need to be addressed before consideration for publication. The main concerns are about the two (great) highlights of the paper : the new model and the time sensitive analytics; i) the manuscript presents a new large scale model, with a very interesting concept for the releases that is however not enough evaluated and discussed, and ii) the approach to quantify the contribution of reservoir coordination to better represent floods and droughts needs to be improved – it is based on inference statements and the model could be modified to include information about upstream reservoir release to demonstrate the point about coordination.

Response 2: Thanks for this comment.

(i) Regarding the general concern on the model, we would like to point out that the hydrological model is not new, and neither is the rationale for the release rule. In fact, WBM is a well established representative of the broader class of large scale hydrological assessment models that are being used in regional to global applications, as highlighted in

p2 lines 19-22 and p5 lines 18-29. As for the release rule itself, it also has a published record of application in WBM (Grogan et al, 2015; 2017; Zaveri et al, 2016; Liu et al., 2017).

What is more, the release rule we use is not put forward for its novelty but for being a state-of-the-art representative of the emerging types of rules that are being employed as reviewed in our Introduction. We highlight this when we detail the rule in Section 2.4 (see p. 11 lines 8-15)

(ii) It is not our intent to precisely quantify the exact contribution of reservoir coordination to overall prediction errors, but rather to use an institutionally complex multi-reservoir cascade that is known to exploit high levels of coordination to illustrate qualitative artifactual behaviors that can emerge from the absence of coordination in how large-scale hydrological models abstract major storages.

This being said, we agree that a new kind of release rule for large-scale hydrological models could take into account coordination. The goal of the paper is not to propose such a rule and demonstrate it, but to diagnose the issue to motivate future efforts in the research community to develop and evaluate new reservoir operation representations that better account for coordination.

Manuscript changes 2:

1. We give details on the provenance and previous uses of the release rule in Section 2.4 ( page 8 lines 2-14)

2. We clarified this goal of our paper in a revised version, in the methodology (bottom page 10, first sentence Section 3.1).

Comment 3: Minor feedbacks are that the reference to typical reservoir model is misleading and the analytics with the method of Morris is very hard to follow.

Response 3: We address the reviewer's concern within their point 1) below. As for the analytics, we appreciate the feedback from the 3 reviewers.

Manuscript changes 3: We made clarifications to interpretability as discussed in our responses here and for Reviewers 1 and 2.

Comment 4: Reference to typical reservoir operations model seems misleading. At the scale of the Upper Snake River Basin, typical reservoir operations models have a nodal architecture and represent accurate reservoir operating rules that can be revisited in optimization mode and especially in forecast mode to mitigate reservoir and drought events. The manuscript here refers to very large scale spatially distributed reservoir models that have been developed initially to be fully coupled with hydrology model and research

land-surface-atmosphere interactions. Those models are typically applied over multiple independent large river basins. I would suggest to not refer to typical reservoir model where most of the community understand reservoir models where rules can be optimized and are applied to one basin at a time. Please refer to large scale distributed reservoir model or equivalent differentiation from nodal operational reservoir models.

Response 4:

We agree that had the goal been to propose the most accurate operating rules possible for this basin, we would have chosen very different rule forms. However, the goal of our study is to diagnose the implications for how state-of-the-art large scale hydrological assessment models represent reservoir operation rules. We understand that this point needs to be made even more explicit, and the revision will add a subsection to the methodology that will point this out.

As for the reviewer's concern with the geographical scale of our study, we would like to point out that release rules such as they are encoded in large-scale hydrological models must not lead to large qualitative errors in mid-size basins such as the USRB. Indeed, larger-scale studies would contain a set of basins that are not hydrologically connected with 1) a number of mid-size basins, and 2) large basins containing several headwater basins such as the USRB. Therefore, insights from a diagnostic approach on the USRB will be relevant to large-scale assessments if we apply the release rule in the same way (i.e., without over-parameterizing it).

For these reasons, the release rule is applied here as it would be within a large-scale hydrological model, that is, without fine-tuning the parameters to each individual reservoir.

Manuscript changes 4: We added a subsection in the methodology (3.1, bottom page 10) to detail our rationale and assumptions explicitly in a single location. Among other things, it explains why we use a single mid-sized basin, and why we do not tailor the rule to this basin (page 12 lines 20-26)

"The geographical extent of the USRB is small compared with that of areas traditionally considered in large-scale hydrological models. This is because a focus on a smaller area makes drawing quantitative and qualitative lessons from our diagnostic analysis easier. These lessons can then be applied to understand the potential consequences of not considering coordination between reservoirs over larger study areas. This is also why we do not fine-tune reservoir rule parameters to individual basins, and instead use purpose-based parameterizations typical in large-scale studies (e.g., Biemans et al., 2011; Voisin et al., 2013; Yoshikawa et al., 2014)."

Comment 5: A new large scale reservoir operations model : please provide more details

Response 5: We would like to clarify that our paper is not focused on a "large scale reservoir operation model". In fact our point is quite the opposite: pointing out the limitations of

standard rule-based representations in hydrological models that consider and parameterize reservoirs separately as reviewed in detail in our Introduction.

Manuscript changes 5: The revision explicitly reminds the reader that the reservoirs are not coordinated in the model, but only in real-life operations, when introducing the methodology (page 10 lines 29-32):

"To diagnose the potential consequences of non-coordination within large-scale models' reservoir release rule, our dual objective is to (A) understand how these consequences play out through space and time in a large-scale hydrological model and (B) compare and contrast with the impacts of real-world coordinated reservoir operations."

We also provide evidence of cooperation within Section 4.1, with Figure 6 and Section 4.1.2, both on page 17.

Comment 6: what is the river routing process for this high spatial resolution and daily time step? A recommendation in the introduction is not to aggregate reservoir storage but many reservoirs have less than 2 days in travel time. How does the reservoir model decision release algorithm adjust stability?

Response 6: Thanks for this. We reply to each sentence separately and in order:

1.  Similar to the above, we would like to point  out that the release rule presented here does not consider what happens downstream, where the river routing is done by the hydrological model (WBM).

2.  Likewise, in the introduction's discussion of Shin et al (2019) (p 2 lines 5-7 original submission) we did not make any prescriptive recommendation related to aggregating reservoir storages. It is part of the second paragraph in the introduction whose role is to set context, which is the evolution of hydrological models towards higher resolution applications. Our work looks at what happens with a model that is relatively highly-resolved, but far from hyperresolution. At the resolution we use though, it makes little  sense to aggregate storages because they are less than two days apart: this would negate the sought advantages of a higher resolution..

3.  We apologize but we are not sure what you mean by "adjust stability" here. This said, the reservoir rule mechanisms are completely specified in Section 2.4.

Comment 7: How are the 6 parameters initialized? Are the necessary data widely available? What are the assumptions?

Response 7: Thanks for pointing out our lack of explanation of where the parameters of the reservoir rules come from. As for the assumptions, we would like to point out that the key one, common to release rules for large-scale hydrological models is that by construction,

each reservoir gets separate parameters that do not depend explicitly on the behavior of reservoirs upstream.

Manuscript changes 7: In a revised version, we inserted in Section 2.4 a paragraph explaining the origin of the rule and of its current parameterizations, as well as the associated assumptions (p 8 lines 2-14; the paragraph can also be in response to Reviewer #1, under "Manuscript changes 6").

The key assumption that this rule (similar to other rules) assumes non-coordinated reservoirs has also been made explicit in Section 2.4 where the release rule is introduced page 8 lines 30-32):

"Note that this parameterization, similar to those of other state-of-the-art rules in large-scale hydrological models, do not account for possible coordination mechanisms in multi-actor, multi-reservoir systems."

Comment 8: Evaluation of the smoother release curves with other models. In other equivalent models that are cited (Hanasaki, Doell, Biemans, Voisin, etc) , releases are decided daily based on reservoirs minimum and maximum capacities, minimum environmental flow and tend to follow monthly storage and releases targets with no foresight, but using long term mean monthly inflow, which also tends to be regulated or natural flow depending on the models. What is the improvement for those rules? The obvious features are the changes in release rates - how does it improve the flow representation in general?

Response 8: The reviewer cites four release rules from four papers (Hanasaki et al (2006), Doll et al (2003), Biemans et al (2011), Voisin et al (2013)) that have release rules that decide a monthly release target in order to analyse outputs with a monthly time step. Given this, it makes sense for the rules to be based on monthly parameters.

Yet, as detailed in our introduction (p3 lines 15-35) the transition of applications to include flood concerns means that shorter timescales have to be considered. The smooth monthly curves are not appropriate for finer time scaled extremes (floods) and for large storages those extremes carryover in the effects on water operations for droughts. So in short, there is tension and difficulty in resolving floods and droughts in complex cascades like USRB using standard rule forms and assumptions of independence between reservoirs.

Please note that we do not try to use a "best" rule but a rule that shares some key characteristics with state-of-the-art available rules (and we would urge the reviewer to refer to newer representations, e.g. see references in p3, lines 30-35, rather than those they cite here).

Manuscript changes 8: The revision clarifies that the version of the Hansaki et al rule that is used for flooding is modified by Mateo et al (2014) at p3 line 16 to be used with daily time step (See p 3 lines 18-19)

Comment 9: reservoir coordination. Note that the use of a rolling past 20-year of mean monthly regulated inflow provides a minimum of reservoir coordination mostly during extreme events. "Some" coordination is represented through the use of mean monthly regulated flow and also the allocation of water demand to a number of reservoirs based on how full they are. This feature is not present in this model representation, and would likely not drastically change extreme events. Yet it does represent "coordination" around releases and other water management performance metrics than flow and storage, rather coordination on meeting basin-scale water demand. There were statement throughout the paper saying that there was no coordination at all, which seemed then inaccurate and should be clarified.

Response 9: We thank the reviewer for this comment. It underlines that there are different interpretations of what coordination means. In this paper, we focus on active forms of coordination by human operators, where the dynamics across space and time in release decisions occur in a way that cannot be explained by the immediate or short-term hydrological or climate conditions. A key feature of coordination as understood in this paper is that upstream reservoirs react to and anticipate downstream water issues.

By contrast, the reviewer seems to focus in this comment on two forms of passive coordination, in which reservoirs adjust to varying on-site conditions that are typically imposed on downstream reservoirs by upstream operations. These conditions unfold concurrently to the release decision (inflows influenced by upstream water management, or at-site withdrawals).

Side note on inflows as a means of coordination: the reservoir model uses long-term (5 year) mean (regulated) inflow (see page 8 lines 2; Figure 3 on page 9), and the paper illustrates that this may not be sufficient to represent coordination

Manuscript changes 9: We clarified the difference between our paper's focus (active coordination) and what is common in release rules in large-scale hydrological models (passive coordination) in the introduction by defining more precisely what we mean by non-coordination (page 4 lines 4-5): "consequences of a release decision on downstream reservoir levels (and management objectives) are not considered"

Comment 10: evaluation of the model and transfer to other regions: âA ˘ c whether the coordination ´ between reservoirs was represented or not, how does it affect the vulnerability metrics at the scale of the basin, which is what those models were initially developed for?

Response 10: We thank the reviewer for keeping their eye on the end-goal here. We agree that large-scale hydrological models are increasingly used to assess water vulnerability. Vulnerability metrics are useful when the difference between a situation and the other is quantitative, but this work is striving to highlight qualitative differences: an adverse event vs. its absence.

This being said, we agree that it is important to put the results into context. For instance, we do explain the consequences of losing control of regulating downstream releases at American Falls in a low-flow event (p 16 lines 29-32).

Manuscript changes 10: We clarified that emptying American Falls instantly disrupts irrigation schedules (and forces farmers to watch their crops wither), by amending the sentence (page 20 lines 5-7) that now reads:

"Therefore, it is key that American Falls, the main reservoir in the Snake River plain located just upstream of Minidoka, is not empty so that it can keep regulating water levels in downstream reservoirs, ensuring irrigation needs are met."

As for the 2011 flood event, we added that "The simulated peak is almost 40% higher than the highest observed daily discharge over the past 40 years. Maximal Palisades release over this period -- 1140m3/s on the 20 June 1997 -- corresponds to a flooding event which led to six counties declaring a state of disaster, leading to over USD 11 million in relief by the federal U.S. government."

Comment 11: Most of those models have been developed for application to a wide range of ´ climatology conditions. The model here is applied to a relatively very small basin for its kind. If this manuscript will be used as reference for this large scale reservoir models, it should be either evaluated with respect to other generic rules, or the applicability to larger regions and very different regions should be presented.

Response 11: Thanks for this comment. Indeed, we are using an example of generic rule that shares key common characteristics with other generic rules; these rules are usually applied to larger hydrological areas.

As explained in a previous comment, it is important to point out what the absence of coordination may lead to if using these rules in headwater basins that would be part of larger-scale studies with large-scale hydrological model.

Manuscript changes 11: We clarified this reason for zooming in to a midsize basin in the methodology section (Section 3.1, page 12 lines 20-26)

Comment 12: Evaluation of the contribution of reservoir coordination – artifact of the model? - the main assumption is that the daily releases are based on storage only. All other equivalent models used an estimate of the expected monthly inflow. The main conclusion of the paper is that the coordination between reservoirs should be represented. While I do believe in this conclusion, it seems that the reference to "typical reservoir model" is not justified if the monthly inflow (proxi for foresight without forward running all the models involved) is not represented at all like in other models. My recommendation would be to modify the experiment to evaluate perhaps incremental and simple levels of coordination (

aka adding inflow as parameter for the decision release, or a proxi for inflow) to complement the interpretation of the results and provide more quantifiable statements.

Response 12: We do agree that models that schedule release over a monthly time step also use inflows over a monthly time step. Then, release is naturally a function of available water (beginning-of-month storage plus monthly inflow) makes more sense than determining release as a function of storage alone. Likewise, daily release decisions should be a function of beginning-of-day storage and daily inflows. In WBM, this is implicitly the case because releases are actually calculated on an hourly time-step, enabling the model to account for day-to-day inflow rate variations as the day progresses; this produces a daily release total.

Note that this rationale of taking the same time step for inflows and releases is common to most release rules for large-scale hydrological models, up to the most recent rules (e.g. Yassin et al, 2019, in this journal).

Beyond this theoretical reasoning, we conducted another check for added confidence that the results are not an artefact of using monthly expected inflows. In the two events examined in this paper, we produced the Figure 1 below, which plots historical monthly release (y-axis) as a function of available water (storage + monthly inflows) for all years of the modeled period 2009-2016 at the most upstream reservoir (Jackson Lake). Results show that historical operations could not be replicated simply by incorporating inflows (even monthly inflows) to the release rule.

For the drought in August 2013 (Section 4.3; left panel) as for the flood in April 2011 (Section 4.4; right panel), the release response is unusual. In the 2013 drought, the reservoir has already been emptied to rescue American Falls levels downstream but releases are still the highest in 8 years, whereas in the 2011 flood, releases are around 8 times as high whereas water availability is similar for 6 of the other 7 years.

[Figure]

[Figure]

Manuscript changes 12 : We clarify in Section 2.4 that the rule is separately implemented at the hourly time step in the model, to assimilate inflow into outflow calculations, in order to produce a total release at the daily time step in the model (page 8 lines 5-6):

"WBM's reservoir module operates on a hourly time step to closely follow storage variations and yield a daily release total."

We inserted this Figure in the revised version (as Figure 6 on page 17), assorted with an explanation that highlights how the anomalous releases (in August 2013 and April 2011 respectively) correspond to coordination that could not be replicated by simply considering expected monthly inflows (Section 4.1.2 on page 17)

Comment 13: Evaluation of the contribution of reservoir coordination during extreme events I found it extremely hard to follow the text and interpretation of the drivers of the release (annual flow versus objective of this reservoir or upstream reservoirs, and shape of release) by just looking at the figures. Most of the text describes the observed operations and coordination and how the model does not capture it. It is unclear how the method of Morris helps with the interpretation during extreme events. While the visualization is very nice to show the data, it seems that those figures could go in the supplemental material and another figure that compiles those time series and support the text would help.

Response 13: Thanks for this comment that will help us clarify our manuscript. We believe that we were not explicit enough in explaining the rationale for the method of Morris.

Manuscript changes 13: In the revised manuscript, we explain in a new Section 3.1 (methods) that when we use the method of Morris (page 12 lines 1-5):

"The focus is not solely on which parameters in the release rule are most influential in regulating flows, but importantly clarifying how the set of dominant controls on water flows and reservoir storage levels evolves along a complex multi-reservoir cascade through time. Parametric sensitivities are then used alongside storage and release trajectories for the simulated ensemble to assess how sets of dominant controls in a point in time and at a given reservoir can be associated to high- or low-flow conditions."

Comment 14: Overall discussion and recommendation versus computational resources needs The authors conclude that foresight should be represented, which is also very sound. Yet the computational resources brought forward for such a relatively small basin are huge which decrease the feasibility at a continental or global scale. Optimization also bring other uncertainties and more computational needs. While the authors seem to indicate that this is what we should do, those were actually drawbacks and motivation for developing those large scale generic models. The recommendation is confusing and perhaps the authors could provide a clarification on new model performances to make it possible now? Please also note that nodal models that typically support reservoir operation optimizations do not provide the spatially distributed feedback into the hydrology model to represent hydrology-land-surface-atmosphere interactions. Maybe the authors meant that we need different types of large scale reservoir models? This would be very sound – just need to be clearer about recommendations then.

Response 14: We fully agree with the reviewer that there is a research challenge ahead in terms of incorporating human complexity into large-scale hydrological models. We recognize that a first step in addressing this challenge is to signal the consequences of ignoring it, and that is what our paper is trying to do. Clearly, it does not try to provide an easy fix, and while the discussion outlines some ways forward, we fully acknowledge that solutions are not going to be easy to design and implement.

[revised manuscript text omitted]

---

## Author Response (AR2)

Dear Editor,

Please find enclosed a new version of our manuscript "Coordination and control: limits in standard representations of multi-reservoir operations in hydrological modeling".

We addressed all comments in our response. As you will see, we made changes throughout the manuscript, in order to clarify our contribution and detail how our methods support it.

We identify our contribution as being twofold:
1) We provide evidence that the common modeling practice of parameterizing each reservoir in a cascade independently from the others is a significant approximation.
2) We demonstrate potential unintended consequences of this independence approximation when simulating the dynamics of hydrological extremes in complex reservoir cascades.

Papers rigorously falsifying common assumptions in hydrological modeling correspond to the publications standard upheld by this journal (for instance this year, https://hess.copernicus.org/articles/24/397/2020/). More generally, we would like to emphasize that it is not a standard in science that any evidence falsifying elements of standard modeling in a field can only be published by solving the highlighted problem itself (in this paper, this would amount to proposing a fully coordinated reservoir release rule). This logic would in fact be severely detrimental to hydrologic science itself and would serve to severely entrench standard modeling practices over making progress in understanding where innovations are needed.

Our responses and manuscript changes clarify how the Method of Morris is combined with other methods in a rigorous experimental design that fully supports our contribution. We hope that subsequent reviews and decisions will be specific in engaging with our response, with our revisions and with the technical content of our manuscript.

Sincerely,

The Authors

**Point-by-point responses to the Editor's comments**
*All page and line numbers refer to the marked manuscript*

Comments to the Author:
Dear Authors, I have very carefully read the reports of the two reviewers who kindly accepted to review the revised manuscript. As you see in their reports, both reviewers have raised significant concerns with the revised manuscript. They concur that the response to the original reviews has not always clarified where this was deemed necessary, but rather in some cases has clarified what may well be some significant flaws in the interpretation and conclusions drawn.

Thank you for this assessment. Here we argue for an additional round of reviews, and encourage the reviewers and editors to carefully assess the manuscript on its merits. In preparing this response we found it challenging to identify clear actionable items to improve the manuscript. This challenge is exacerbated by the limited technical specificity in the reviewers' engagement with our previous response to reviews. Additionally, we detailed our revision strategy during the previous discussion phase, and that was endorsed at the editorial level, but we do not see this round of review engaging with how the implementation of our revision strategy fell short of what was announced (we believe the implemented revision is entirely coherent with the strategy we detailed beforehand).

Still, we have considered these comments carefully and have found a number of edits that we hope will convince reviewers that our manuscript presents a clear contribution and relies on a rigorous methodology. In particular, we accept that a clearer communication (also suggested by Reviewer #1) on the exact nature of our contribution would have made it easier to assess our paper.

In this revision we have done our best to fix this and clarify our contributions. We have rewritten the abstract and made modifications throughout the manuscript (Introduction: p 2, lines 28-31, p 3 lines 1-5, p 5 lines 3-7; Methodology all of Section 3.1 p 15; Results with the addition of Section 4.5 starting p 25; Discussion p 28 lines 28-31). This is to clarify the context, rationale, and contributions of the work. In particular, we define our contribution more precisely as being (p 1 lines 9-12):

"The aim of this paper is twofold, (i) provide evidence that the common modeling practice of parameterizing each reservoir in a cascade independently from the others is a significant approximation, and (ii) demonstrate potential unintended consequences of this independence approximation when simulating the dynamics of hydrological extremes in complex reservoir cascades."

We then clarify the role of the Method of Morris as the centerpiece of our analysis, p 1 lines 16-18:

"We employ a time-varying sensitivity analysis that utilizes Method of Morris factor screening to explicitly track how the dominant release rule parameters evolve both along the cascade, and in time according to seasonal high- and low-flow events."

The following sentence explicitly links this use of the Method of Morris to the way we substantiate claim (i), p 1 lines 19-21:

"This enables us to address (i) by demonstrating how the progressive and cumulative dominance of upstream releases significantly dampens the ability of downstream reservoir rules' parameters to influence flow conditions."

Immediately after the above revision, the abstract has been edited to clarify how the Method of Morris is associated with other methods to back claim (ii), p 1 line 21 to p 2 line 4:

"We address (ii) by comparing simulation results with observed reservoir operations during critical low-flow and high-flow events in the basin. Our time-varying parameter sensitivity analysis with the Method of Morris clarifies how independent single-reservoir parameterizations and their tacit assumption of independence leads to reservoir release behaviors that generate artificial water shortages and flooding, whereas the observed coordinated cascade operations avoided these outcomes for the same events. To further explore the role of (non-)coordination in the large deviations from the observed operations, we use an offline multi-reservoir water balance model in which adding basic coordination mechanisms drawn from the observed emergency operations is sufficient to correct the deficiencies of the independently parameterized reservoir rules from the hydrological model."

We also overhauled Section 3.1 in our effort to demonstrate that the technical basis of our analysis has been carefully designed and executed. We hope that our clarifications reflect our efforts to carefully diagnose the problematic practice of independently parameterizing reservoir operational rules in large scale hydrological models that are employed to simulate institutionally complex multi-reservoir cascades.

I have proceeded to also carefully read the revised manuscript, and agree with the reviewers that there are issues in the experimental set-up. I also agree with the reviewers that the conclusions on the coordination between upstream and downstream reservoirs simply through showing that the generic reservoir rule without coordination fails to capture the observed operations may be not be fully substantiated.

We assume that the existence of human coordination in multi-reservoir operations and its absence in hydrological model representations is not a point of contention of the editor and reviewers. Obviously, systems such as  the cascade in the Upper Snake basin were built and are presently operated as coordinated infrastructure systems for the specific purpose of addressing real-world drought and flood period extremes.

Our experimental setup does show that the generic rule without coordination fails to capture the observed operations, but we would like to point out that it goes further. Indeed, our offline water balance experiments demonstrate that adding simple coordination mechanisms that mimic observations would have been enough to avoid the erroneous depiction of floods and water shortage in the model.

We understand that our low-key communication of this crucial point in our revision probably was not clear enough. Therefore, we have grouped all text regarding the rationale, design, execution and results of the coordinated offline water balance experiments in a new Section 4.5, and added a Table 3 (p 27) that highlights how key flooding and water shortage metrics are affected by modifying hydrological simulations results by adding simple coordination mechanisms in offline water balance model of the reservoir cascade.

We also chose to announce these offline water balance experiments both in the abstract (p 2 lines 1-4) and introduction  (p 5 lines 3-7).

Based largely on the combined recommendation of the reviewers, I propose that you carefully reflect on the critique of the reviewers, both on this second revision as well as on the orignal submission. This may require to some extent a redesign of the paper. Although the paper is very well written and presented, as did the reviewers I also struggled with the importance given in the paper to the application of the method of Morris, whilst the purported contribution (as the title also suggests) is on the coordinated operations between reservoirs. Perhaps this can be divided across two contributions. Even the abstract eludes to these being two separate parts of the manuscript, showing that you yourself seem to have struggled with the coherence of the paper.

As suggested both in our response to the paragraph above, and in the paper (see for instance Section 3.1, both in the first revision and in this one), the Method of Morris is the centerpiece of our study, but it is not the only method used in this work. The Method of Morris illuminates how non-coordinated operations lead to the behaviors observed in the simulations. Comparison between simulations and the historical record shows that these behaviors exhibit such large differences that they become qualitatively different -- simulating damaging water extremes that did not happen in the real world. Finally, offline water balance modeling of the reservoir cascade demonstrates  that adding basic coordination in the reservoir's operations is sufficient to address WBM's  inability to simulate the analyzed  high- and low-flow events.

Our methodological approach enables a rigorous analysis of the complex interplay between control rule characterization across reservoir cascades, where simulated downstream releases are almost entirely controlled by modeling choices at upstream reservoirs. Though not widely recognized in the hydrological modeling community, these dynamics pose a significant modeling challenge.  Method of Morris enables us to substantiate our claims that these problems exist by mathematically mapping over time which parameters and reservoirs are dominating the broader storage and release dynamics of the Upper Snake basin's cascade. Likewise, our water balance modeling reinforces our results by showing deficiencies in representing coordination is the simplest explanation for systematic errors in drought and flood periods.

We hope that the rewriting of the abstract and of other parts of the text clarifies the interplay between the methods for the reader. As suggested by the editor, we clarified the paper's two main contributions.

Following this reflection, the manuscript may well need to be re-drafted. It should then be re-submitted for further review.

As noted in both review rounds, our paper is well-written and our revisions in this response seek to clarify the context and contributions of the work.

Thank you for your handling of this paper.

**Point-by-point response to comments from Anonymous referee #1**
*All page and line numbers refer to the marked manuscript*

A key issue at stake (with all three reviewers) is the ability to demonstrate that coordination is an important missing component in the reservoir modeling, preventing adequate simulation of flood and drought response. If three independent pairs of eyes look at a paper and reach the same critique then this is sure a sign that either (a) there is a major problem with the conclusions drawn, or (b) the way in which the study is communicated is insufficient. I don't think the revision has really tackled this challenge, as I find myself at the same point as before, having carefully studied the paper and responses.

Thanks for this. Please let us clarify why we appreciate the reviewer's original concerns and believe that this paper fully tackles what the reviewer correctly identifies as its central challenge, i.e., demonstrating "that coordination is an important missing component in the reservoir modeling".

First of all, please recall that we made the following substantial changes (among others) between the original version and the revised version submitted in May:

A. We added Figure 6 to clearly explain that the historical record in the upstream reservoir cannot be accounted for with release rules that do not account for coordination.

B. We produced offline reservoir water balance experiments to demonstrate how very basic reservoir coordination mechanisms could have avoided the water extremes simulated by the WBM model, all else being equal.

C. We inserted a new Section 3.1 to outline how changes (A) and (B) can be articulated with Method of Morris results and the comparison between simulation results and the observed historical record. These changes were also introduced in the abstract.

How does that demonstrate "that coordination is an important missing component in the reservoir modeling"?

1) **Coordination is a missing component in the reservoir modeling process.** This is a fact, explored and explained at length in the introduction, both in the first submission and in the revision. To clarify things further, we decided to explicitly define coordination between reservoirs in this revision (p 2 lines 28-30): "Multi-reservoir coordination implies that release decisions at each reservoir in the basin are explicitly influenced by the current and future state of other reservoirs. So far such behavior has not been implemented in release rules for large-scale hydrological models."

2) **A first key reason why modeling coordination is important is that upstream reservoir rule parameterization influences downstream releases.** This was demonstrated in the paper by contrasting upstream and downstream controls (sections 4.1 and 4.2). To emphasize the importance of this finding in the overall narrative that omitting coordination has consequences, we highlighted in the revised abstract, introduction and methodology, that this finding corresponds to addressing

key contribution (i) providing "evidence that the common modeling practice of parameterizing each reservoir in a cascade independently from the others is a significant approximation" (p 1 lines 9-11)

3) **Another reason why modeling coordination is important is that it is present in key moments in historical operations, to avoid adverse consequences in high- and low-flow situations.** Please note that change (A) listed above, already present in the previous revision, addressed this point.

4) **Further, coordination is important because not representing it in the model leads to simulating severely amplified water shortages and floods.** Coordination would easily prevent these events. This is what happened in the historical record (demonstrated throughout Sections 4.3 and 4.4), but also by inserting simple coordination mechanisms in offline multi-reservoir water balance models (change (B)) above. To clarify this, we now grouped all text referring to experiments based on offline water balance models in a new Section 4.5 (see p 25).

Findings (3) and (4) address key contribution (ii): "demonstrate potential unintended consequences of this independence approximation when simulating the dynamics of hydrological extremes in complex reservoir cascades", now made explicit in abstract, introduction, and methodology.

We are unclear what issues the above reasoning raises. If there are issues, indications on what the reviewer's underlying rationale is would be very helpful.

As reviewer 3 suggested, this would be best addressed by actually incorporating coordination in the model (showing that this key piece allows for those important behaviors to be captured).

We thank the reviewer for agreeing that a separate cascade reservoir model forced with observations would address the reviewer's comment. We think that change (B) addressed this concern in the revised manuscript, and would be thankful for the reviewer to explain more precisely why they don't think it did.

If instead, this was a clarity issue on our part, we hope that the way we gathered all information on the offline water balance experiments in Section 4.5, and added Table 3 with clear results on what the absence vs presence of coordination meant, addressed the reviewer's concern. Please note that we further clarified the presence of these experiments in the abstract: "To further explore the role of (non-)coordination in the large deviations from the observed operations, we use an offline multi-reservoir water balance model in which adding basic coordination mechanisms drawn from the observed emergency operations is sufficient to correct the deficiencies of the independently parameterized reservoir rules from the hydrological model."

Yes, comparison of simulated water balance with / without coordination, carried out in change (B), does demonstrate clearly the extent of the role coordination can play to mitigate flood and drought.

This could also be done a separate cascade reservoir model forced with observations. Then one can simply remove the coordination to demonstrate clearly the extent of the role coordination plays (in this system) to mitigate flood and drought.

We find the idea of removing coordination challenging to carry out in practice. In our model we rather added it to demonstrate that erroneous outcomes from non-coordinated simulation could be corrected by adding simple coordination mechanisms. From a strictly logical standpoint, both approaches are strictly equivalent in completing the demonstration in the way the reviewer suggests.

Besides, please note that removing coordination from observations, i.e., from the historical record, would be extremely difficult to do, as it would require exact knowledge of all the decision rules in a many-reservoir system subject to complex regulations. There is to our knowledge no existing technique to do that. This is why we opted for adding a very simple coordination mechanism to the modeled release rule instead. This is something where we control the parameters.

I do not think reviewer 3 was suggesting you propose a new scheme for large-scale models (as implied by your response) but was suggesting that this would be the cleanest way of answering your chosen research question.

We agree with the reviewer that careful studies highlighting quantitatively a deficiency in a standard practice in large-scale hydrological modeling can help motivate the field to more fully engage to address it.

Proposing a new scheme for large-scale models is beyond the scope of this work. Besides, it is not a standard of science that one can only publish stories that point out problematic aspects of standard modeling practices by proposing a solution. Doing so would in many cases make it very difficult to challenge established assumptions, and would be detrimental -- in this case, to hydrological modeling.

This is an important research question being addressed, and I think it's a missed opportunity for a high-impact study if published in the current form. I struggle get a really clear picture of the importance of coordination from the inference statements offered, so I encourage the authors to provide at the heart of the analysis a modeled coordination component, and then resubmit.

Thank you for stressing the importance of the research question. We hope that the clarifications we proposed in the revised manuscript convinced the reviewer of the importance of representing coordination in multi-reservoir systems, and equally importantly, clarified 1) that the modeled coordination component was added as was requested by the

reviewer, but 2) proposing a solution that would modify reservoir release rule with general coordination rules would be outside of the scope of this work.

**Point-by-point response to comments from Anonymous referee #3**
*All page and line numbers refer to the marked manuscript*

Many thanks to the authors for the detailed response. The first overarching comment and detailed comments #1,2,3,5 ( with respect to the description of existing models and that the paper is not about a new model) have been clarified. The organization and approach have also been clarified – with a prognostic approach to expose the time varying influence of generic rule parameters followed by a qualitative assessment of how those drivers differ from observation in times of extreme events. An additional offline experiment has been added to show that if generic rules could be adjusted during extreme events, the representation would be improved.

We thank the reviewer for detailing the areas they have no outstanding issues with.

The overarching #2 comment and the detailed comment #4 are still concerning.

Our reply refers back to the reviewer's comments which we prefer to explicitly recall in order to propose a precise response.

Overarching #2 comment:

*(ii) the approach to quantify the contribution of reservoir coordination to better represent floods and droughts needs to be improved – it is based on inference statements and the model could be modified to include information about upstream reservoir release to demonstrate the point about coordination.*

As we pointed out in our original response to this comment, we do not propose or claim to propose an "approach to quantify the contribution of reservoir coordination". To clarify, we provide evidence that (1) not representing coordination is an approximation, and (2) this approximation can have clear unintended consequences. We argue that this is an important and necessary contribution in a research landscape where there is a large number of studies that propose reservoir release rules, invariably for single reservoirs (see introduction). Papers that demonstrate the potential unintended consequences of a common assumption are usually useful for the community that uses that assumption. For instance, HESS published earlier this year a paper by Dang et al. (see reference list) that demonstrated the potential consequences of eschewing the representation of reservoirs altogether in large-scale hydrological models.

We did add supplementary data and offline water balance experiments are not relying on "inference statements". However, we chose (maybe wrongly) to not elucidate what coordination is and that it is a fundamental trait in the design and operation of multireservoir cascades to manage hydrologic variability at large regional scales. Indeed, "information about upstream reservoir release" is by definition included in the inflows to a reservoir, regardless of coordination. If the upstream reservoir releases an excessive amount of water into the downstream reservoir, the downstream reservoir is not coordinating, it is coping with

poor upstream decisions. In this revision, we have clarified our meaning in the use of the word coordination (p 2 lines 28-30): "Multi-reservoir coordination implies that release decisions at each reservoir in the basin are explicitly influenced by the current and future state of other reservoirs. So far such behavior has not been implemented in release rules for large-scale hydrological models". If the reviewer would like to understand more on state-based coordination, studies cited in the same paragraph (p 2 line 28) discuss the issue of how shared state information on storages and releases is central to mathematical control formulations that lead to coordinated reservoir operations.

We also would like to clarify that coordination is extremely difficult to quantify in real systems, in part because real systems are multi-actor, multi-purpose and heavily regulated, which means actors are going to coordinate in different configurations depending on what they want to achieve in given circumstance, and how and with whom they can achieve it. In the water resource literature, coordination scholars generally focus on quantifying the economic benefits from coordination for a narrow range of management objectives and hydroclimatic outcomes (see e.g., Marques and Tilmant , 2013, given in the reference section). This is why we choose instead a methodology that provides clear evidence of our claims.

Detailed comment #4:

*4) Evaluation of the contribution of reservoir coordination during extreme events I found it extremely hard to follow the text and interpretation of the drivers of the release (annual flow versus objective of this reservoir or upstream reservoirs, and shape of release) by just looking at the figures. Most of the text describes the observed operations and coordination and how the model does not capture it. It is unclear how the method of Morris helps with the interpretation during extreme events. While the visualization is very nice to show the data, it seems that those figures could go in the supplemental material and another figure that compiles those time series and support the text would help.*

Given that the reviewers consistently indicate the Method of Morris text was difficult to understand, we have added more explicit reference to the "Method of Morris" to clarify where and how it used for readers who are less familiar with global sensitivity analysis.

For the drought, the previous revision (at p 21 lines 19-22 and 27-28) showed that the dominant upstream parametric controls as highlighted by the Method of Morris do not change with imminent downstream water shortage. To better guide the reader, we inserted an explicit reference to the Method of Morris (p 21 line 25) in the text that introduces sensitivity analysis results in that section (now p 21 lines 25-28, and p 21 line 33 to p 23 line 2).

For the flood, we clearly discuss in the first revision the dominant parametric controls and how they favor untimely reservoir filling. For Jackson Lake, see in the first revision p 22 lines 14-23 and 26-29; we added an explicit reference to the "Method of Morris" (now p 23 line 23) to that text (now p23 lines 23-32 and p 25 lines 1-4). For Palisades, we similarly added an

explicit "Method of Morris" reference (p 25 line 9) to references to time-variant sensitivity analysis results that already existed in the previous revision. This text is now p 25 lines 9-16.

We hope these clarifications, along with those in the abstract, help to better address the reviewer's comment.

We respond to the following paragraph sentence by sentence in an effort to make the response easier to follow.

The manuscript presents a complex analytics to demonstrate this intuitive fact that coordination between reservoirs needs to be represented to capture realistic dynamics during extreme events.

We are glad this is intuitive to the reviewer. We would also like to point out that this is not intuitive according to the existing literature on reservoir release rules in large-scale hydrological models. Indeed this literature, detailed in the introduction, does not represent coordination between reservoirs.

It is one thing to have the intuition something is important, and quite another to demonstrate how and when. Our study is the first to demonstrate the need for representing coordination using a diagnostic mathematical framework that explicitly maps how parameterizations upstream cumulatively dominate the value or effects of downstream parameterized rules.

The experimental approach with the method of Morris does not seem necessary to demonstrate that this coordination between reservoirs is needed to better represent dynamics during extreme events.

See above our response to detailed comment #4 from the first review.

The overall manuscript now comes out as two components – the first one that shows the method of Morris and how the parameterization of generic rules influence the results and how inadequate – or adequate for the wrong reasons- results are during extreme events.

We have rewritten the abstract to reflect the paper's dual contribution (p 1 lines 9-12). We hope this clarifies that the two components the reviewer describes address the same gap in the literature. Indeed, the Method of Morris is important in both contributions. As explained above, it is instrumental in understanding model behavior when it simulates artifactual water extremes. We hope the rewritten abstract, and changes throughout the manuscript, clarify how the methods work together.

As mentioned in the discussion by the authors, this is an important component for overall evaluation of complex processes and ensure that one has the simulated results needed for the right reasons (Objective A).

We understand "this" refers to the Method of Morris, and agree that it is our cornerstone for the whole analysis.

The second part of the manuscript is qualitative (objective B), with a discussion on how the inadequate results of the generic rules during extreme events is indeed due to operations that are not realistic – this includes the wrong influence of parameters.

Perhaps our use the word "qualitative" in the manuscript was confusing, so please let us clarify it in our response. By qualitative differences, we are not trying to say that they are not quantifiable. Instead we mean quantitive differences so large that they can only be accounted for by qualitatively different processes.

In other words, these are structural differences, and we systematically replaced the term "qualitative" with the term "structural" throughout the manuscript.

We would also like to clarify that our experimental setup is entirely quantitative, and this helps us to expose problems stemming from the structural (or qualitative) decision to not represent coordination between reservoirs:
1) Missing coordination in release rules is a structural (qualitative) difference in the formulation, which means its consequences can only be exposed through qualitatively different outcomes.
2) We needed to expose these structural (qualitative) differences in behavior to devise the purely quantitative offline reservoir cascade water balance modeling experiments that a) incorporate a simplified representation of observable coordination behavior, and b) can prove that adding simple coordination rules would have been enough to lead to quantitative differences so large in the simulation results they lead to qualitatively different outcomes.

Note that these offline reservoir water balance experiments are necessary to isolate the impact of coordination from confounding effects.

However the method of Morris is not enough to characterize missing processes.

We agree, and this is why we complement the Method of Morris with other methods, as now summarized in the abstract (p 1 line 19 to p 2 line 4), announced in the introduction  (p 5 lines 3-7) and explained in detail in Section 3.1 (see paragraph starting p 12 line 31).

Authors then follow with a qualitative discussion on how coordination between reservoirs is a missing process, and describe processes for headwater and cascading reservoirs.

We would like to point out, in line with our response above, that the "qualitative discussion" is in fact a discussion based on structural differences caused by the non-representation of coordination.

This discussion is entirely underpinned by quantitative methods and results. For instance, we quantify the difference between storage during the drought in model and reality; we quantify the difference between modeled flood peak and what constitutes a historical flood. But these figures are only interesting as they illustrate how large the differences in outcome brought in by a structural (qualitative) difference in formulation (missing coordination). Our

analysis of coordination provides a direct, simple, and quantified water balance explanation of the wet and dry period deviations by capturing very basic coordination.
We also added Table 3 to further quantify the difference between 1) the historical record (with coordination), 2) the hydrological model simulations (without coordination), 3) the offline water balance models of the reservoir cascade, that add coordination to simulation results.

The approach to expose this missing process is not supported/exposed in a novel way by the qualitative analytics, and does not stand out as novel, yet is presented as the main outcome of the paper.

As noted above, we assert that our conclusions are the result of a quantitative analysis. We pose a challenge to the reviewer to find evidence that the main conclusion from this manuscript is not novel. We have proposed a comprehensive literature review on reservoir rule representation in macro-scale hydrologic models (see Introduction). That review did not find any quantitative evaluation of the common assumption that reservoirs could be modeled independently from each other, even in multi-reservoir cascades. Likewise, we have not found any comprehensive strategy for integrating coordination in these representations. Therefore, we think that it is appropriate and timely for this manuscript to enter the literature. The Upper Snake River Basin is exemplary as a study basin as the evidence of coordinated management is transparent (see Section 2.1), and WBM is a well cited macro-scale hydrologic model that is indicative of the state of practice.

Basically, there is a misalignment between the novel analytics (Objective A - prognostic approach of how different generic rules parameters influence the results across events) and the conclusions and recommendation of the paper (Objective B - need to represent coordination and more accurate release rules), with the latter supported only by a qualitative study and therefore is not novel.

Again, we do claim that our diagnostic use of the Method of Morris is novel in the context of serving as an instrument that rigorously and mathematically substantiates our core contributions as described in our responses above. We have addressed the core logic in our experimental design in our response to Reviewer #1 above. We thank the reviewers for their initial comments that inspired our coupling of Method of Morris diagnostics with water balance modeling that explore the effects of basic coordination. The offline reservoir water balance experiments that clearly show that including coordination is sufficient to prevent the severe unintended consequences our quantitative experimental setup exposed.

**Summary of main changes**

- The abstract was rewritten to clarify the contribution and summarise the methodology.
- Section 3.1 explaining the general approach (p 12) has been largely rewritten to make the interplay between the different methods we use easier to follow for the reader.
- Parts detailing the offline experiments and their results, previously spread across Sections 4.3 and 4.4, have now been grouped in a new Section 4.5 (p 25). This new paragraph comes with a new Table 3 that gives the headline results that show how adding simple coordination mechanisms would have avoided erroneous interpretations of hydrological model results regarding the occurrence of water extremes.
- These clarifications are repeated and justified at different different locations throughout the manuscript (Introduction: p 2, lines 28-31, p 3 lines 1-5, p 5 lines 3-7; Methodology all of Section 3.1 p 15; Results with the addition of Section 4.5 starting p 25; Discussion p 28 lines 28-31).

- Note that there are also minor changes throughout the manuscript to announce these changes or clarify points of details raised by reviewers in this round of comments (see marked manuscript below).

[revised manuscript text omitted]

---

## Author Response (AR3)

**Editorial decision**

Dear Authors, I would thank you for your patience, as well as careful addressing the comments made by the reviewers. On the recommendation of the reviewers, I would like to recommend the paper for publication. However, some minor suggestions have been made to strengthen the link of the work presented to the large scale studies that consider reservoir operations and the challenges faced. I would think these are reasonably straightforward clarifications that contribute to underlining the novel contribution the paper makes. I am sure you will be able to provide these, following also the quite specific suggestions of the reviewer. In your response please indicate if and how these minor comments have been addressed. I will then review those, before the final recommendation for publication.

Regards and all the best wishes for 2021

Dear Editor,

Thank you for carefully assessing our paper. We have provided responses to the reviewer's suggestions and have added clarifications where appropriate to our manuscript.

We wish you all the best for 2021.

**Responses to comments from reviewer #3**

Authors have improved the paper with clear and direct statements about the contribution of the paper. Based on the response, I disagree with the authors that the lack of coordination was because researchers did not know any better, but because of a data availability and computational burden this was deemed an acceptable assumption for the time being. I agree with the authors that the quantification of such assumption is timely and of high interest. In view of this clarification, more specific recommendations are provided below to clarify i) why coordination was not ignored but rather seen as a computational tradeoff which is key for further recommendations (not solutions), ii) how the cost of this computational tradeoff is a timely scientific contribution, and iii) that the experimental approach needs to make the link with science questions typically addressed with such large studies rather than only watershed-scale flood and droughts.

Thank you for your time invested in re-assessing our work, and for providing new insightful comments. We really appreciate your time and effort.

Note that in what follows, we use page and line numbers from the new marked manuscript (track changes pdf file).

We agree that the lack of coordination in hydrological models' representations of multi-reservoir operations is non-trivial and addressing the issue does pose significant potential computational challenges. However, science (whatever the discipline) is an evolutionary learning process that must carefully contront when assumptions that were reasonable in previous studies need to be carefully re-examined. The trade-offs between computational demands and model accuracy are also themselves rapidly evolving with new emerging computational architectures, new software development services, and continual innovations in hydrological models' representations of coupled human-natural processes.. In fact, there is no evidence from our careful review of the literature that there has been a broad and consistent effort to revisit the assumption of coordination in multi-reservoir systems. Moreover, the evidence of computational demands precluding future engagement with the issue is not substantiated by our highlighted discussion of the rapidly changing rule-forms or embedded optimization strategies that have already emerged to date. The end of the second paragraph (p 2 L 20-24) cites the two studies that in our understanding, concern themselves with errors introduced by common (non-coordinated) representations of reservoir operations. None of these use the word "coordination", and only Masaki et al (2017) discusses trade-offs between computational efficiency and detailed representations of natural and social systems, before discussing how more accurate representations are needed.

As for the clarifications asked here, we would like to clarify that:

i) There is no substantive evidence in the literature on reservoir representations that the trade-offs between accuracy and computational costs have been carefully evaluated before deciding in any of the many papers that propose ever-more sophisticated release representations. Throughout the paper we prefer to remain agnostic as to why coordination has not been represented even as model sophistication increased. There are in fact many issues that can be considered on equal footing with computational demands such as

increased data requirements, challenges in choosing appropriate representations of state-aware adaptive operations, issues associated with conflicting operational objectives, challenges in abstracting the importance of exogenous information such market-based energy prices, etc. Instead, we focus on describing the unintended consequences from this approximation using the WBM illustrative example in collaboration with the model's lead developers.

ii) We appreciate the reviewer's focus on the trade-off between computational cost and model accuracy, and again, we fully agree there is one. However, we are not quantifying it in this paper and are wary of conjecture on the topic. To quantify this trade-off we would need to propose alternative reservoir rules with different levels of coordination, and run the full WBM with them to compare both model behavior and computational cost. Our core supported contribution is the quantitative diagnostic illustration of the importance and unintended consequences of not capturing coordination in complex multi-reservoir systems.

However, we agree that this trade-off will be important to carefully consider going forward, so we inserted the following sentence in the discussion (P 29 L 21-22)

"Approaches to address this need will have to contend with trade-offs between the quality of multi-reservoir operations modeling, computational costs, and data availability (Masaki et al., 2017)."

iii) Our introduction already addresses this via a detailed overview of the literature: how large-scale hydrological models have evolved (both in scope and through their representation of reservoir operations) and where that modeling is going (towards high-resolution models able to forecast and monitor water extremes and their consequences). In that sense, we explain how a detailed diagnostic of commonly used assumptions is timely, before we start seeing sophisticated models that make flawed predictions of the future (on this, please refer to our discussion, especially the paragraph starting P 28 L 31)

Since this paper is not a review, our survey of the literature is focused on the topic at hand. We see no specific evidence we should broaden the scope of our paper / literature review.

Overall:

[Approach to quantity the contribution of reservoir coordination needs improvement]; Authors have clarified the scope with clear statements.

[Conclusion is not novel and forward recommendations are not provided]; Scientific literature should include contributions that challenge previous approaches and results. From the present analytics, it is clear that high resolution model should invest in representing coordination between reservoirs, (alongside better operating rules as well), and evaluation approaches going beyond goodness-of-fit.

Thanks for this assessment.

Most of the analytics focus on how floods and droughts, while large scale studies typically focus more on energy-water-food nexus questions, flows of commodities and virtual flows across regions. From this paper, I would not conclude that previous large scale analyses on energy-water-food are completely wrong, and typically there are disclaimers around extreme events. And it is fair and supported by this analytics to say that coordination between reservoirs might be a strategic research area in large scale studies to address extreme events.

Our study does not pretend to falsify any previous large-scale hydrological modeling effort that dealt with water extremes in some way. As noted in (iii) above, we simply caution against using sophisticated but non-coordinated reservoir representations in studies that account for water extremes.

Accordingly, as noted in more detail below, most of the suggestions are around:
- clarification that operational water models exist and should be used for watershed-scale flood and drought studies

We agree that operational water models exist, but they are not commonly used in large-scale hydrological models and there are good reasons for this (e.g., data availability issues, refer in particular to discussion paragraph starting P 29 L 21). Therefore, these operational models are clearly out of scope here.

- the justification of the USRB as a case study would benefit from including links to energy-water-food questions and expand beyond flood and droughts.

We openly cite energy-food-water questions in our presentation of the USRB (see P 5  L 15). What is more, water scarcity is explicitly linked with water use for irrigation (see Section 4.3)

- a couple recommendations were provided in the discussion section however they represent the same computational and data challenges that led to "ignoring reservoir coordination as an accepted computational tradeoff" until more data are available and modeling platforms allow to run forward looking simulations with objective functions going beyond watershed-scale interest.

Thanks for these.

All pages and lines are based on the marked document which included the responses to reviewers. The actual marked change manuscript started on page 16 (out of 58).

1)       Conclusion and recommendations
L17, L26 and conclusion. "there remains opportunities for research to determine which aspects of human management are most urgent to integrate". I cannot agree more. The conclusion of the paper is relatively generic and a clear statement of the authors concluding that based on their analytics, they identified reservoir coordination as a next priority for research development would strengthen the way this paper can be cited. The conclusion already mentions "high resolution modeling" and "need support from more than water

resources modelers" but was not as direct on the emphasis on coordination between reservoirs. While not discussed in this paper, more accurate flows or and more accurate reservoir operations could also be needed.

Thanks for this insight.

2) Choice of the basin and choice of the models to support large scale science questions: P18L11-12: the USRB is presented as a good case study because of its 128 reservoirs and water management known to address both flood and drought events. This is contrasting with large scale studies that have often focused on overall water availability for different water users and regional flow of commodities (electricity, food). The paper might benefit from linking the USRB case study with known regional energy-water-food virtual flows which might be challenged by the way flood and droughts are represented. This would provide a more direct link with large scale studies.

Please note that the third paragraph of the introduction, discussed in your comment, comes after the framing from the two previous paragraphs.

3) Framing
P19, L26. "It is worth noting that all the reservoirs representations discussed above do not account for coordination […] to date there has been not carefully designed diagnostic". While authors also discuss it in the discussion, it would be good here to say why coordination had not been addressed before. The lack of data and the computational needs to represent coordination have been a roadblock. I do not think that this is fair to say that "coordination was assumed non-existent" rather coordination was ignored due to limited data availability and overall treated as a computational trade off. There has been to date no diagnostic quantifying the cost of this computational tradeoffs. It is timely to quantify it in order to inform research priorities as the community advances in high resolution modeling.

As noted in our response to the reviewer's first comment, the literature does not give reasons why coordination has not been addressed before, and we do not pretend to settle that matter. This is simply out of scope in our view.

Note we do not state that "coordination was assumed non-existent", nor do we intend to imply it at any point in the paper. We just point out that coordination is not accounted for, and explore potential consequences.

P21L9-12: "all of these characteristics … flood and drought ….dam failure ….climate change .. make the USRB basin a good case study". I found the paragraph misaligned with large scale studies. For dam failure and reservoir operations under extreme events for decision making at the watershed scale, there are a number of reservoir models for that purpose such as RiverWare and MODSIM (https://www.usbr.gov/research/projects/detail.cfm?id=3669). It seems a good basin for its link to energy-water-food research, with a decent number of reservoirs and a size that allows to carry out this computationally intensive diagnostic (as mentioned on P30L20-24 and could be moved earlier). It seems really important to clarify this point about large scale studies else one would still wonder why you did not consider a simplified version of Riverware with and with coordination to address watershed-scale flood and droughts risks.

We disagree with the idea that the paragraph is misaligned with large-scale studies (as already explained in Response 4 to reviewer 3's comments in the first round of revisions).

This being said, we thank the reviewer for their suggestion, and inserted at the end of Section 2.1 (P 5 L 32-33)

"Any unintended consequences from modeling non-coordinated operations would be a note of caution for large-scale studies featuring water infrastructure balancing protection against water extremes with other competing uses".

P34L20 "calibration of individual reservoirs in a cascade is an approximation modelers make at their own risk". Please substitute to "as a computational trade off" or equivalent.

We toned down by deleting "modelers make at their own risk".

We avoided to refer to a computational trade off because we do not quantify it (and nor has anybody else).

P36L30 ( difference in storage levels where 2013 could not recover after 2012). While it can be seen as a lack of coordination, the generic rules does not indicate carry-over storage which would drastically impact the way multi year droughts are simulated. I am not sure that this lack of carry over storage should be associated with a lack of coordination, but rather inaccurate reservoir operations.

Here the reviewer explicitly refers to a passage discussing historical levels (L30 and L31 of that version of the manuscript). The passage describes how historical storage levels did not recover at Palisades in the multi-year drought, meaning that the model simulations did not suffer from a non-representation of a carryover storage rule for which there is no evidence in the historical record at the end of the 2012 hydrological year.

To clarify things further in this revision we changed the first phrase of that paragraph (now starting P 22 L 13-16) into:

"Yet, in 2013 historical storage levels at Palisades (yellow line on panel (b)) had not recovered from the exceptional 2012 drawdown due to a combination of low carryover storage and insufficient snowmelt. Palisades reservoir could no longer supply extra water to the Snake River Plain."

Besides addressing the reviewer's question about carryover storage, this emphasises the sentence and those that follow describe historical observations.

Section 4.4 and 4.3 address flood and droughts. It is important to connect them with science question of large scale studies for water-energy-food questions as mentioned earlier.

Water scarcity in 4.3 is linked to irrigation. What is more, flood protection and water supply directly trade-off as reservoir management objectives.

P45L34-P46L1-3: use of watershed scale hydrology models to get more accurate rules. This statement could have been used earlier when presenting the USRB basin, clarifying that such models exist but you use the large scale representation to provide information to large scale studies. For flood and droughts specifically, and watershed -scale decision making for water users only, those more complex models should be used otherwise.

We feel mentioning these models earlier will cause confusion for the reader, since they are out of scope (as expressed earlier).

P46L11-20 (last paragraph in the discussion section): this entire paragraph is about reservoir operations optimization schemes potentially for large scale models – I do not suggest removing it but complementing it with statements about the fact that data availability and computational burden are already a challenge for large studies. And such complex representations would require even more research to develop objective functions that reflect purposes outside of the scale of the watershed and complex water-energy-food interaction with other regions.

Thanks, we agree and inserted a statement that (P 30 L 8-11): "Yet, this approach is also computationally expensive and needs to use offline water balance models to parameterize parsimonious reservoir rules that can be input into large-scale hydrological models."

Thanks again for your comments, which helped to improve the manuscript.